# Spectral Collapse Drives Loss of Plasticity in Deep Continual Learning

Arjun Prakash [* 1]  Naicheng He [* 1]  Kaicheng Guo [* 1]  Saket Tiwari [1]  Tyrone Serapio [1]  Ruo Yu Tao [1]
Amy Greenwald [1]  George Konidaris [1]

## Abstract

We investigate why deep neural networks suffer from loss of plasticity in continual learning, and thus fail to learn new tasks without reinitializing parameters. We show that this failure is preceded by Hessian spectral collapse at new-task initialization, where meaningful curvature directions vanish and gradient descent becomes ineffective. Analyzing a linearized ReLU network, we derive explicit $\epsilon$-rank conditions for successful training and prove that the loss-weighted Gram matrix is spectrally equivalent to the Generalized Gauss-Newton approximation, thereby relating NTK dynamics to Hessian curvature. Targeting spectral collapse directly, we then discuss the Kronecker factored approximation of the Hessian, which motivates two regularization enhancements: maintaining high effective feature rank and applying L2 penalties. Experiments on continual supervised and reinforcement learning tasks confirm that combining these two regularizers effectively preserves plasticity.

## 1. Introduction

Human intelligence is marked not by mastering a single task, but by the ability to continually adapt to new challenges, through the accumulation and refinement of skills and knowledge. Likewise, a central aspiration of artificial intelligence is to create systems that can learn throughout their lifetimes. Endowing artificial systems with this ability would ensure that they remain useful, relevant, and robust in an open world (Javed & Sutton, 2024).

Research on deep learning and deep reinforcement learning has demonstrated remarkable progress, reaching human and even superhuman performance in image recognition (Siméoni et al., 2026), game playing (Silver et al., 2017), and reasoning (Bakhtin et al., 2022). But these advances have largely been realized under stationary data distributions. *Continual learning* concerns a system's ability to perform well through a sequence of evolving tasks. Within this framework, two complementary objectives are often distinguished: (i) *maintaining plasticity*, meaning the capacity to acquire new knowledge (Dohare et al., 2024), and (ii) *preventing catastrophic forgetting*, where the learner forgets how to solve tasks it previously mastered (Kirkpatrick et al., 2017; Parisi et al., 2018; Baker et al., 2023).

Our work explores plasticity loss from an optimization perspective. Despite significant empirical progress in the area (Lyle et al., 2023; 2024; Abbas et al., 2023), there remains no unifying consensus on what drives loss of plasticity (Klein et al., 2024). Several disparate explanations have been investigated, including inactive neurons which output constants regardless of the inputs (Bjorck et al., 2022; Sokar et al., 2023), the reduction in feature expressiveness associated with large parameter norms (Lyle et al., 2024), and overfitting (Igl et al., 2021). Recent work has begun to connect plasticity loss to second-order properties, such as the stable rank of the Hessian or layer-wise spectral conditioning (Lewandowski et al., 2024b;a). Building on these insights, we argue that these seemingly disparate observations are in fact different facets of the same phenomenon, which we call *spectral collapse*: a degeneration of the Hessian eigenspectrum, indicating loss of curvature in most directions. An example comparison of Hessian eigenspectrum with and without the spectral collapse is shown in Figure 1: in the BP run, the task-50 spectrum has a broad positive bulk, indicating many usable curvature directions; but by task 300, most spectral mass has collapsed, leaving only a few isolated non-zero eigenvalue directions. In contrast, L2-ER preserves a broader spectral bulk across tasks.

Our main contributions can be summarized as follows:

1. We define Hessian spectral collapse, and identify it as a central mechanism underlying plasticity loss. To support this claim, we provide extensive empirical evidence across diverse continual learning benchmarks, demonstrating that existing plasticity-preserving inter-

[*]Equal contribution , order drawn from random [1]Department of Computer Science, Brown University, Providence, RI, USA. Correspondence to: Arjun Prakash <arjun_prakash.edu>, Naicheng He <naicheng_he@brown.edu>.

*Proceedings of the 43$^{rd}$ International Conference on Machine Learning*, Seoul, South Korea. PMLR 306, 2026. Copyright 2026 by the author(s).

ventions designed to target seemingly different phenomena all prevent spectral collapse.

2. We analyze a linearized ReLU network and derive explicit $\epsilon$-rank conditions for successful training. We establish that the loss-weighted Gram matrix is spectrally equivalent to the Generalized Gauss-Newton (GGN) matrix, thereby relating function-space analysis to parameter-space curvature.

3. Leveraging this theory, we propose $L2$-ER regularization, intended to target Hessian eigenspectrum collapse directly, and show that it matches or exceeds existing methods on various continual supervised learning and reinforcement learning benchmarks.[1]

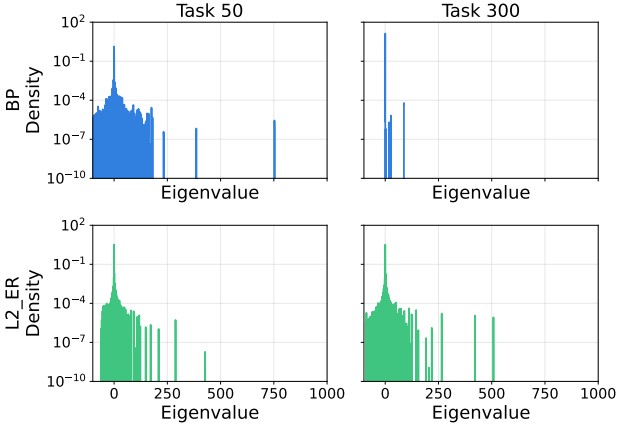

*Figure 1.* The Hessian eigenspectrum on continual Imagenet at task initialization. The top row shows the eigenspectrum on a network using standard backpropagation (BP). The bottom row shows the eigenspectrum using $L2$-ER regularization (ours).

## 2. Related Work

**Causes of Plasticity Loss**  Several pathologies have been proposed to explain loss of plasticity in continual learning, both supervised and reinforcement (Lyle et al., 2023; Klein et al., 2024). For example, *dormant neurons*, which output a positive value on all inputs, reduce a network's expressive power, while *dead neurons*, which output zero on all inputs, prevent gradient flow (Montufar et al., 2014; Sokar et al., 2023). Reductions in *effective feature rank*, which measures the intrinsic dimensionality of feature representations and directly relates to the persistence of dead neurons, has also been linked to loss of plasticity (Roy & Vetterli, 2007; Dohare et al., 2024), as has *parameter norm growth*, which leads to ill-conditioned Hessians that cause numerical instability (Obando-Ceron et al., 2024; Lyle et al., 2024). Lyle et al. (2024) propose a "swiss cheese" model that attempts

[1]Code is available at here.

to preserve plasticity by targeting all these pathologies simultaneously.

**Mitigating Plasticity Loss**  Existing methods for mitigating loss of plasticity target one or more of the aforementioned pathologies (Klein et al., 2024). Broadly speaking, these approaches fall into two categories: (i) continuous interventions that apply at every optimization step, and (ii) intermittent interventions that periodically reset or perturb parameters (Juliani & Ash, 2024). For example, Lyle et al. (2024) shows that combining $L2$ regularization to control parameter norm growth with layer normalization to stabilize pre-activation distributions is highly effective. Other approaches regularize the singular values of the layer-wise Jacobians (Lewandowski et al., 2024a) to maintain parameter distributions close to a uniform random initialization (He et al., 2015b; Hinton & Salakhutdinov, 2006). Architectural changes such as PELU (Godfrey, 2019), CRELU (Shang et al., 2016; Abbas et al., 2023), and adaptive rational activations (Delfosse et al., 2024) avoid the saturation of neurons. In contrast, periodic interventions, such as Shrink & Perturb (Ash & Adams, 2020), continual backpropagation (CBP) (Dohare et al., 2024), and ReDo (Sokar et al., 2023), selectively reset dormant neurons. Empirically, hybrid strategies—typically involving $L2$ regularization combined with either architectural or periodic interventions—exhibit the strongest performance (Lyle et al., 2024; Juliani & Ash, 2024), implying that plasticity loss can emerge from multiple mechanisms.

**Approximate Second-Order Methods**  Our work contributes to a recent line of inquiry that connects loss of plasticity to second-order information. Prior work by Lewandowski et al. (2024b) provides strong empirical evidence that loss of plasticity correlates with a decline in the stable rank of a partial Hessian, proposing a Wasserstein penalty to maintain characteristics of the initial parameter distribution. In a complementary approach, Lewandowski et al. (2024a) focus on the spectral properties of the layer-wise weight matrices, rather than the Hessian, using spectral regularization to keep the largest singular value near one, thereby maintaining favorable conditioning and gradient diversity. Similarly, Parseval regularization (Chung et al., 2024) improves weight-matrix conditioning by enforcing orthogonality in the weight matrices to preserve gradient norms.

**Neural Tangent Kernel in Continual Learning**  The neural tangent kernel (NTK) (Jacot et al., 2018) provides a tractable lens on continual learning by approximating training as kernel regression around initialization. Prior work analyzes continual learning in the NTK regime through transfer and forgetting (Doan et al., 2021) or uses empirical-NTK degeneracies as a diagnostic (Lyle et al., 2024). Our

analysis isolates a concrete failure mode: learnability is controlled by how much of the task error residual lies in the *fast* eigenspace, meaning the span of those Gram-matrix eigenvectors whose eigenvalues are large enough to contract substantially within the budgeted number of gradient steps. For permuted MNIST, we prove the NTK is invariant to input permutations, which implies that plasticity loss arises from progressive shrinkage of the fast subspace rather than task-induced kernel drift. While Tang et al. (2025) connect rank collapse to output variability, we show that once the fast eigenspace collapses, learning fails even with stable outputs—motivating L2-ER, which targets the underlying geometry directly. Our work is also related to Singular Learning Theory (SLT) (Watanabe, 2009), which is concerned with understanding singularities in the Fisher information matrix of neural networks. The empirical NTK and Fisher information matrices are equivalent up to reversal and scaling (Karakida et al., 2021) and share non-zero eigenvalues. SLT explains why models move toward singular regions, while our work studies the continual-learning consequence of this phenomenon.

## 3. Preliminaries

**Notation** We use $\tau$ to index tasks, $k$ for optimization iterates, $l$ for layers, $n$ and $m$ for samples, $o$ for output coordinates, and $q$ for eigenvalues. A layer-specific matrix at task $\tau$ and iterate $k$ is written $\mathbf{M}_\tau^{[l],(k)}$, with its $(a, b)$-entry written as $[\mathbf{M}_\tau^{[l],(k)}]_{ab}$. We write $\boldsymbol{x}_{\tau,n}$ and $\boldsymbol{y}_{\tau,n}$ for the $n$th data sample in dataset $\boldsymbol{X}$. We also write $[a]$ to denote the set of integers $\{1, \ldots, a\}$, for some $a \in \mathbb{Z}$.

**Deep Neural Networks** A multi-layered-perceptron (MLP), also called a feedforward neural network, is constructed by composing $L$ functions, i.e., $F^L \circ \cdots \circ F^1$, one per layer. Given an input (resp. output) dimension of $M^{l-1}$ (resp. $M^l$), the function at layer $l$, for $l \in [L]$, is a map $F^l : \mathbb{R}^{M^{l-1}} \to \mathbb{R}^{M^l}$. This function $F^l$ is the composition of an element-wise non-linear activation function $\sigma^l : \mathbb{R} \to \mathbb{R}$ and a weight matrix $\mathbf{W}^{[l]} \in \mathbb{R}^{M^l \times M^{l-1}}$, so that $F^l = \sigma^l \circ \mathbf{W}^{[l]}$. Our activation function of choice is ReLU , defined as $\max\{0, x\}$. The dimension at layer $l$ indicates the number of neurons at that layer. The total number of hidden neurons is $M \doteq \sum_{l=1}^{L-1} M^l$.

We collect all parameters into a single vector using the vec operator, so that the MLP parameterization is represented by $\boldsymbol{\theta} \doteq \text{vec}(\mathbf{W}^{[1]}, \ldots, \mathbf{W}^{[L]}) \in \mathbb{R}^P$, where $P = \sum_{l=1}^L M^l M^{l-1}$ is the total parameter count. Denoting the input (resp. output) dimension of the MLP by $I \doteq M^0$ (resp. $O \doteq M^L$), the MLP map is $F_{\boldsymbol{\theta}} : \mathbb{R}^I \to \mathbb{R}^O$.

A task $\tau$ is described by a loss function $\ell_\tau$ and a data distribution $\boldsymbol{d}_\tau$. Given such a task $\tau$, a learning algorithm seeks parameter values that minimize the expected loss $\mathcal{L}_\tau(\boldsymbol{\theta}) \doteq \mathbb{E}_{\boldsymbol{d}_\tau}[\ell_\tau(\boldsymbol{z}_\tau, \boldsymbol{y}_\tau)]$ on the task, where $\boldsymbol{z}_\tau = F_{\boldsymbol{\theta}_\tau}(\boldsymbol{x}_\tau)$.

The gradient of the network is the vector of first derivatives of the loss function w.r.t. the network parameters, i.e., $\nabla_{\boldsymbol{\theta}} \ell_\tau \in \mathbb{R}^P$, while the main object of our investigation, the *Hessian*, is the matrix of second derivatives, i.e., $\mathbf{H} \doteq \nabla_{\boldsymbol{\theta}}^2 \ell_\tau \in \mathbb{R}^{P \times P}$. Together, the Jacobian and the Hessian capture up to second-order changes in the loss, i.e., when the parameters change by a vector $\delta \in \mathbb{R}^P$, the loss can be approximated as $\mathcal{L}_\tau(\boldsymbol{\theta} + \delta) \approx \mathcal{L}_\tau(\boldsymbol{\theta}) + \nabla_{\boldsymbol{\theta}} \mathcal{L}_\tau^\top \delta + \frac{1}{2} \delta^\top \nabla_{\boldsymbol{\theta}}^2 \mathcal{L}_\tau \delta$.

**The Hessian** The Hessian offers valuable insight into the loss landscape of a neural network (Pouplin et al., 2023). For example, one line of research is concerned with large connected regions in weight space where the error remains approximately constant (Hochreiter & Schmidhuber, 1997); such regions may be less prone to overfitting and result in smaller generalization gaps (Cha et al., 2021). Since the Hessian $\mathbf{H}$ is symmetric, there exists an orthonormal basis of eigenvectors $\{\mathbf{v}_q\}$ and real eigenvalues $\{\lambda_q\}$ satisfying $\mathbf{H} \mathbf{v}_q = \lambda_q \mathbf{v}_q$, for all $q \in \{1, \ldots, P\}$. Each eigenvalue $\lambda_q$ measures the second-order curvature of the loss in the direction of eigenvector $\mathbf{v}_q$. A large eigenvalue indicates a high local curvature in the loss landscape in the direction of the corresponding eigenvector $\mathbf{v}_q$, while an eigenvalue near zero indicates low local curvature in that direction. Given the importance of the eigenvalues of the Hessian, the distribution of all eigenvalues, known as the *eigenspectrum*, provides more information than any single scalar summary, such as the largest eigenvalue or the trace. It is possible decompose the Hessian further using the chain rule, to obtain the Gauss-Newton decomposition (Zhao et al., 2024):

$$\nabla_{\boldsymbol{\theta}}^2 \ell = \mathbf{H} = \underbrace{\frac{1}{N} \sum_{n=1}^N \mathbf{J}_n^\top \left[ \nabla_{\boldsymbol{z}}^2 \ell(\boldsymbol{z}_n, \boldsymbol{y}_n) \right] \mathbf{J}_n}_{\mathbf{H}_{\text{GGN}}}$$

$$+ \underbrace{\frac{1}{N} \sum_{n=1}^N \sum_{o=1}^O \left[ \nabla_{\boldsymbol{z}} \ell(\boldsymbol{z}_n, \boldsymbol{y}_n) \right]_o \nabla_{\boldsymbol{\theta}}^2 [\boldsymbol{z}_n]_o,}_{\mathbf{R}} \quad (1)$$

where at the $n$th sample, $\mathbf{J}_n \in \mathbb{R}^{O \times P}$ is the Jacobian of the model w.r.t. its parameters (i.e., $\nabla_{\boldsymbol{\theta}} F_{\boldsymbol{\theta}}(\boldsymbol{x}_n)$), $\nabla_{\boldsymbol{z}}^2 \ell \in \mathbb{R}^{O \times O}$ is the Hessian of the loss w.r.t. the model outputs (i.e., the pre-softmax logits in classification or the scalar output in regression), $[\nabla_{\boldsymbol{z}} \ell]_o \in \mathbb{R}^{O \times 1}$ is the gradient in the $o$-th output dimension of the loss w.r.t. to the model's output, and $\nabla_{\boldsymbol{\theta}}^2 [\boldsymbol{z}_n]_o$ is the Hessian of the $o$-th output element w.r.t. the model parameters. Simply dropping the residual term $\mathbf{R}$, yields the Generalized Gauss-Newton (GGN) matrix $\mathbf{H}_{\text{GGN}}$, a faithful proxy for the Hessian (Dangel et al., 2025).

**Neural Tangent Kernel (NTK)** Given the the complexity of deep neural networks with non-linearities, the neural tangent kernel (NTK) has been proposed as an abstraction to characterize the training dynamics of (stochastic) gradient descent in the infinite-width limit (i.e., $M \to \infty$) (Jacot et al., 2018). The main principle behind NTK theory is to approximate a function by linearizing a neural network around its initialization. The seminal work by Jacot et al. shows that if $F$ is an appropriately scaled neural network, with $\boldsymbol{\theta}$ initialized i.i.d. from standard (or isotropic) Gaussian's (i.e. $\boldsymbol{\theta} \sim \mathcal{N}(0, \boldsymbol{I})$), then $F_{\boldsymbol{\theta}}(\boldsymbol{X}) \approx F_{\boldsymbol{\theta}^{(0)}}(\boldsymbol{X}) + \mathbf{J}(\boldsymbol{\theta} - \boldsymbol{\theta}^{(0)})$. The linearized model $F_{\boldsymbol{\theta}}$ can be interpreted as a kernel method, where the Jacobian is the feature map. The kernel induced by the feature map is the NTK and is dependent on the random initialization $\boldsymbol{\theta}^{(0)}$. The neural tangent kernel $\kappa_{\boldsymbol{\theta}} : \mathcal{X} \times \mathcal{X} \to \mathbb{R}^{O \times O}$ is given by

$\kappa_{\boldsymbol{\theta}}(\boldsymbol{x}, \boldsymbol{x}') \doteq \mathbb{E}_{\boldsymbol{\theta} \sim \mathcal{N}(0, \boldsymbol{I})}[\langle \frac{\partial F_{\boldsymbol{\theta}}(\boldsymbol{x})}{\partial \boldsymbol{\theta}}, \frac{\partial F_{\boldsymbol{\theta}}(\boldsymbol{x}')}{\partial \boldsymbol{\theta}} \rangle |_{\boldsymbol{\theta} = \boldsymbol{\theta}^{(0)}}]$. Evaluating this kernel function on a finite dataset of since $N$ (i.e., for all $\boldsymbol{x}_n, \boldsymbol{x}_m \in \boldsymbol{X}$) gives the empirical NTK matrix $\kappa_{\boldsymbol{\theta}}(\boldsymbol{x}_n, \boldsymbol{x}_m) = \mathbf{J}\mathbf{J}^\top$. This matrix, which captures all pairwise inner products among vectors in the set $\boldsymbol{X}$, is called the *Gram matrix* and is denoted by $\mathbf{G} \in \mathbb{R}^{NO \times NO}$. The celebrated result from NTK analysis is that as $M \to \infty$, the NTK $\kappa_{\boldsymbol{\theta}}$ converges in probability to an explicit deterministic limit $\kappa_{\boldsymbol{\theta}}^{\infty} = \mathbb{E}_{\boldsymbol{\theta} \sim \mathcal{N}(0, \boldsymbol{I})}[\mathbf{G}]$ (Jacot et al., 2018, Theorem 1). This deterministic limit only depends on the variance parameter of the Gaussian initialization and the choice of non-linearity. The NTK limit holds for ReLU activations (Arora et al., 2019) and LeCun initialization (Jacot et al., 2018).

# 4. Continual Learning and Plasticity Loss

Assume an agent facing a sequence of $T$ tasks whose decision-making capabilities are dictated by a neural network with parameters $\boldsymbol{\theta} \in \mathbb{R}^P$. Each task $\tau$ is characterized by an evaluation metric $f_\tau : \mathbb{R}^P \to \mathbb{R}$, such as accuracy for supervised learning or expected return for reinforcement learning. In continual learning, we seek a sequence of parameters $\{\boldsymbol{\theta}_\tau^*\}_\tau$ s.t. $\boldsymbol{\theta}_\tau^* \in \arg\max_{\boldsymbol{\theta}_\tau} f_\tau(\boldsymbol{\theta}_\tau)$, for all $\tau \in [T]$.

Since evaluation metrics may be non-differentiable, we further assume a (possibly regularized) surrogate expected loss function $\mathcal{L}_\tau : \mathbb{R}^P \to \mathbb{R}$, for each task $\tau$, which is intended to maximize the evaluation metric, but is generally better behaved. Additionally, we assume the learner has a budget of $K$ learning steps per task. A solution to the continual learning problem is generated by an algorithm $\mathcal{A} : \mathbb{R}^P \to \mathbb{R}^P$, which outputs parameters for the current task $\tau$ that optimize $\tau$'s surrogate loss function $\mathcal{L}_\tau$, given the parameters learned during the previous task, i.e., the parameters for task $\tau + 1$ are initialized as the learned parameters of task $\tau$: i.e., $\boldsymbol{\theta}_{\tau+1}^{(0)} \doteq \boldsymbol{\theta}_\tau^{(K)}$.

**Definition 4.1** (Successful Training). In a continual learning problem, given finite update budget $K$ and a small error tolerance $\upsilon$, we write $\mathbb{I}(\ell_\tau(\boldsymbol{\theta}_\tau^{(K)}) \leq \upsilon)$ to indicate *successful training* on task $\tau$. We say a neural network is *plastic* if it successfully trains on all tasks $\tau \in [T]$.

In the remainder of this section, we analyze how spectral information of the NTK Gram matrix governs successful training. This analysis motivates the design of our regularizer, which preserves neural network plasticity by controlling the eigenspectrum of the Hessian.

## 4.1. Linearized ReLU networks

We now instantiate the NTK framework for a linearized ReLU network. Assume input dimension $I$, width $M$, and output dimension $O$. Denote the first-layer weights by $\mathbf{W}^{[1]} = [\mathbf{w}_1^{[1]}, \ldots, \mathbf{w}_M^{[1]}] \in \mathbb{R}^{I \times M}$ where $\mathbf{w}_h^{[1]} = [\mathbf{W}_h^{[1]}]_{:,h} \in \mathbb{R}^I$ is the input weight vector of hidden unit $h$. We fix the output weights and the ReLU gates, so the only trainable parameters in this linearized model are the first-layer weights. Define the fixed output-layer weights by $\mathbf{W}^{[2]} = [\mathbf{w}_1^{[2]}, \ldots, \mathbf{w}_M^{[2]}]^\top \in \mathbb{R}^{M \times O}$ (i.e., $\mathbf{w}_h^{[2]} = [\mathbf{W}^{[2]}]_{h,:}^\top \in \mathbb{R}^O$) Then, with $\sigma(x) = \max\{0, x\}$, the output of a linearized ReLU network on input $\boldsymbol{x} \in \mathbb{R}^I$ is given by:

$$\hat{F}_{\boldsymbol{\theta}}(\boldsymbol{x}) \doteq \frac{1}{\sqrt{M}} \sum_{h=1}^M \mathbf{w}_h^{[2]} \sigma((\mathbf{w}_h^{[1]})^\top \boldsymbol{x}) \in \mathbb{R}^O.$$

We fix $\mathbf{W}^{[1](0)}$ to be a frozen reference copy of the initialized first-layer matrix $\mathbf{W}^{[1]}$. With this fixed initialization, we define the fixed ReLU gate function $g_h(\boldsymbol{x}) \doteq \mathbf{1}\{(\mathbf{w}_h^{[1](0)})^\top \boldsymbol{x} > 0\}, h \in 1, \ldots M$, and gate matrix: $\mathbf{D}_g(\boldsymbol{x}) \doteq \mathrm{diag}(g(\boldsymbol{x})) \in \mathbb{R}^{M \times M}$. By construction, the only trainable parameters are $\mathbf{W}^{[1]}$, initialized at $\mathbf{W}^{[1]} = \mathbf{W}^{[1](0)}$, and the model prediction is :

$$\hat{F}_{\boldsymbol{\theta}}(\boldsymbol{x}) = \frac{1}{\sqrt{M}} F_{\mathbf{W}^{[1]}}(\boldsymbol{x})^\top \mathbf{W}^{[2]} \tag{2}$$

$$= \frac{1}{\sqrt{M}} (\mathbf{W}^{[2]})^\top \mathbf{D}_g(\boldsymbol{x}) (\mathbf{W}^{[1]})^\top \boldsymbol{x} \in \mathbb{R}^O. \tag{3}$$

**Continual Leaning with NTKs** Given task $\tau$ with dataset $(\boldsymbol{X}_\tau, \boldsymbol{Y}_\tau) = \{\boldsymbol{x}_{\tau,n}, \boldsymbol{y}_{\tau,n}\}_{n=0}^{N_\tau}$, the model prediction is $\hat{F}_{\boldsymbol{\theta}}(\boldsymbol{X}_\tau) \in \mathbb{R}^{N_\tau \times O}$. We define the squared-error loss as $\ell_\tau(\boldsymbol{\theta}) \doteq \frac{1}{2} \left\| \hat{F}_{\boldsymbol{\theta}}(\boldsymbol{X}_\tau) - \boldsymbol{Y}_\tau \right\|^2$. We define the residual on task $\tau$ at iterate $k$ as $r_\tau^{(k)} \doteq \hat{F}_{\boldsymbol{\theta}_\tau^{(k)}}(\boldsymbol{X}_\tau) - \boldsymbol{Y}_\tau \in \mathbb{R}^n$. In continual learning, where each task is trained for a finite number of steps $K$, $\boldsymbol{\theta}_{\tau+1}^{(0)} \doteq \boldsymbol{\theta}_\tau^{(K)}$. Because the ReLU gates are frozen, the model is linear in $\mathbf{W}^{[1]}$. Thus the linearized expression, $\hat{F}_{\boldsymbol{\theta}} = F_{\boldsymbol{\theta}}$ is exact for all $\mathbf{W}^{[1]}$.

Next, we present two lemmas that enable us to define successful training based on $\epsilon$-rank. The proofs and extended explanations are deferred to Section B.

Lemma 4.2 shows that, under NTK dynamics with squared error loss and learning rate $\eta$, gradient descent acts directly on the residual by repeatedly multiplying the residual by $(\boldsymbol{I} - \eta\mathbf{G}_\tau)$. Lemma 4.3 reveals that by diagonalizing $\mathbf{G}_\tau$ each residual component contracts independently at a rate associated with the magnitude of eigenvalue.

**Lemma 4.2** (Residual dynamics in the linearized regime). *If $\hat{F}_{\boldsymbol{\theta}}(\boldsymbol{X}_\tau) \approx F_{\boldsymbol{\theta}_\tau^{(0)}}(\boldsymbol{X}_\tau) + \mathbf{J}_\tau(\boldsymbol{\theta} - \boldsymbol{\theta}_\tau^{(0)})$, then gradient descent on $\ell_\tau(\boldsymbol{\theta})$ with learning rate $\eta$ yields as the $k+1th$ iterate $r_\tau^{(k+1)} = (\boldsymbol{I} - \eta\mathbf{G}_\tau)r_\tau^{(k)}$, where $\mathbf{G}_\tau = \mathbf{J}_\tau\mathbf{J}_\tau^\top \in \mathbb{R}^{NO \times NO}$. Hence, the final value of the residual for task $\tau$ is $r_\tau^{(K)} = (\boldsymbol{I} - \eta\mathbf{G}_\tau)^K r_\tau^{(0)}$.*

**Lemma 4.3** (Eigenvalue Decomposition). *Assume $\mathbf{G}_\tau$ has eigenpairs $\{\lambda_{\tau,q}, \mathbf{v}_{\tau,q}\}_{q=1}^N$ with orthonormal $\{\mathbf{v}_{\tau,q}\}$ and $\lambda_{\tau,q} \geq 0$. Expand the initial residual:*

$$r_\tau^{(0)} = \sum_{q=1}^n \alpha_{\tau,q}\mathbf{v}_{\tau,q}, \quad \alpha_{\tau,q} \doteq (\mathbf{v}_{\tau,q})^\top r_\tau^{(0)}. \quad (4)$$

*Then, for all iterates $k$, $r_\tau^{(k)} = \sum_{q=1}^n (1 - \eta\lambda_{\tau,q})^k \alpha_{\tau,q}\mathbf{v}_{\tau,q}$.*

### 4.2. Eigenvalue contraction

We can now formalize a notion of *fast* and *slow eigenspaces* based on an explicit threshold derived from the number of per-task iterates $K$ and the learning rate $\eta$.

**Definition 4.4.** A *fast eigenvalue* must shrink by at least a factor of $\gamma \in (0,1)$ after $K$ steps: i.e., $(1 - \eta\lambda)K \leq \gamma$. In particular, $\lambda \geq \epsilon_{K,\gamma}$ where $\epsilon_{K,\gamma} \doteq \frac{1-\gamma^{1/K}}{\eta} \approx -\log\gamma/\eta K$. Given eigenvalues $\lambda_{\tau,q}$, define the fast set $\mathcal{F}_\tau \doteq \{q : \lambda_{\tau,q} \geq \epsilon_{K,\gamma}\}$ and slow set $\mathcal{S}_\tau \doteq \{q : \lambda_{\tau,q} < \epsilon_{K,\gamma}\}$. The $\epsilon$-rank can now be defined as $|\mathcal{F}_\tau|$.[2]

**Definition 4.5** (Projectors). Given eigenvectors $\mathbf{v}$ of $\mathbf{G}_\tau$, we define the fast (resp. slow) projection matrix $\mathbf{P}_\tau^\mathcal{F} \doteq \sum_{q\in\mathcal{F}_\tau} \mathbf{v}_{\tau,q}\mathbf{v}_{\tau,q}^\top$ (resp. $\mathbf{P}_\tau^\mathcal{S} \doteq \boldsymbol{I} - \mathbf{P}_\tau^\mathcal{F}$).

**Lemma 4.6.** *For any task $\tau$,*

$$\left\|r_\tau^{(K)}\right\| \geq \left\|\mathbf{P}_\tau^\mathcal{S} r_\tau^{(K)}\right\| \geq \gamma \left\|\mathbf{P}_\tau^\mathcal{S} r_\tau^{(0)}\right\|. \quad (5)$$

The consequence of Lemma 4.6 is that slow eigenvalues are sticky: If a component of the initial residual lies in the slow eigenspace, then after $K$ gradient steps at least a $\gamma$ fraction of that component remains in that slow subspace. By Definition 4.1, successful training on task $\tau + 1$ requires $\{\ell_\tau(\boldsymbol{\theta}_{\tau+1}^{(K)}) \leq \upsilon\}$, which, by Lemma 4.6, can be equivalently stated as $\{\|r_{\tau+1}^{(K)}\| \leq \sqrt{2\upsilon}\}$.

---

[2]In our notation, we omit dependence of $\mathcal{F}_\tau, \mathcal{S}_\tau$ on $K, \gamma$.

**Lemma 4.7** (Necessary fast-projection condition). *Using the fast and slow projections, $\epsilon_{K,\gamma}$, and the initial loss residual $r_\tau^{(0)}$, a necessary condition to achieve successful training on task $\tau$ is $\gamma \left\|\mathbf{P}_\tau^\mathcal{S} r_\tau^{(0)}\right\| \leq \sqrt{2\upsilon}$. Equivalently,*

$$\left\|\mathbf{P}_\tau^\mathcal{F} \frac{r_\tau^{(0)}}{\|r_\tau^{(0)}\|}\right\|^2 \geq 1 - \left(\frac{\sqrt{2\upsilon}}{\left\|\gamma r_\tau^{(0)}\right\|^2}\right) \quad (6)$$

To succeed on a new task, the eigenvalues of the initial Gram matrix must lie overwhelmingly in the fast subspace, unless the initial residual magnitude $\left\|r_\tau^{(0)}\right\|^2$ is already tiny. When too many eigenvalues fall below the threshold $\epsilon_{K,\gamma}$, the fast subspace shrinks and the necessary condition in Lemma 4.7 becomes impossible to satisfy. We refer to this phenomenon as *spectral collapse*. Next, we demonstrate how spectral collapse governs successful training in a linearized ReLU network.

## 5. Permuted MNIST

We now analyze Permuted MNIST to show that, under the frozen-gate NTK model, fresh pixel permutations do not change the infinite-width kernel geometry. Thus, when spectral collapse is observed in finite-width continual training, it is attributable to the curvature of the learned representation having degenerated across previous tasks, rather than to the permutations themselves.

The MNIST dataset (LeCun, 1998) is a collection of $N$ handwritten digits with input dimension $I = 28 \times 28 = 784$ and labels $\boldsymbol{y} \in \{0, \ldots, 9\}$. Let $\boldsymbol{\Pi} \in \mathbb{R}^{I \times I}$ be a permutation matrix with a single 1 in each row and and a of 1 in each column, but not diagonal. A *permuted MNIST task* is obtained by sampling a fresh permutation $\boldsymbol{\Pi}_\tau$ and transforming every input by $\boldsymbol{x} \mapsto \boldsymbol{\Pi}_\tau\boldsymbol{x}$, leaving labels unchanged. Concretely, from an underlying base dataset $\{(\bar{\boldsymbol{x}}_n, \bar{\boldsymbol{y}}_n)\}_{n=1}^N$ we define the task-$\tau$ dataset $\boldsymbol{X}_\tau \doteq \{\boldsymbol{\Pi}_\tau\bar{\boldsymbol{x}}_n\}_{n=1}^N$ and $\boldsymbol{Y}_\tau \doteq \{\bar{\boldsymbol{y}}_n\}_{n=1}^N$. Informally, Lemma C.6 states that $[\kappa^\infty]_{o,o'}(\boldsymbol{\Pi}\boldsymbol{x}_n, \boldsymbol{\Pi}\boldsymbol{x}_m) = [\kappa^\infty]_{o,o'}(\boldsymbol{x}_n, \boldsymbol{x}_m)$ for $m, n \in [N]$. This implies the expected MSE loss and frozen-gates Gram matrix is task-invariant under fresh pixel permutations, up to finite-width fluctuations, which means that loss of plasticity is *not* caused by permutations. Instead, spectral collapse is caused by the accumulation of low-rank features and the shrinking of the fast eigenspace during training of prior tasks, which leaves the network ill-conditioned for subsequent permutations.

**Softmax Cross-Entropy Loss** For multi-class cross-entropy with logits $\boldsymbol{z} = (\boldsymbol{z}_1, \ldots, \boldsymbol{z}_N) \in \mathbb{R}^{NO}$ and block-wise softmax $\mathrm{softmax}(\boldsymbol{z})$, the empirical risk $\ell(\boldsymbol{z}) = \sum_{n=1}^N \ell(\boldsymbol{z}_n, \boldsymbol{y}_n)$ satisfies $\nabla_{\boldsymbol{z}}\ell(\boldsymbol{z}) = \mathrm{softmax}(\boldsymbol{z}) - \boldsymbol{Y}$. In

the NTK regime, with the Jacobian frozen at initialization, gradient descent induces a closed-form update on logits of the form:

$$z^{(k+1)} = z^{(k)} - \eta G^{(0)} \nabla_z \ell(z^{(k)}), \qquad (7)$$

$$G^{(0)} = J^{(0)}(J^{(0)})^\top, \qquad (8)$$

where $J^{(0)}$ is the initial feature map induced by the inputs and $G^{(0)} \in \mathbb{R}^{(NO) \times (NO)}$ is the corresponding multi-output Gram matrix with blocks

$$[G^{(0)}]_{(n,o),(m,o')} = \langle \nabla_\theta [F_\theta(x_n)]_o, \nabla_\theta [F_\theta(x_m)]_{o'} \rangle.$$

This is exactly the discrete-time counterpart of the standard NTK function-space dynamics for general losses (Lee et al., 2019) (i.e., $[\kappa_\theta(x_n, x_m)]_{o,o'}$). We show in Section C.3 that the linearized weighted error $\mathbf{e}^{(k)}$ evolves according to

$$\mathbf{e}^{(k+1)} \approx (I - \eta C^{1/2} G^{(0)} C^{1/2}) \mathbf{e}^{(k)}, \qquad (9)$$

where $C = \nabla_z^2 \ell(z)$. Consequently, for cross-entropy loss, the eigenvalues that determine collapse are those of the weighted Gram matrix, $G_{\mathrm{LW}} = C^{1/2} G^{(0)} C^{1/2}$. Since by Lemma C.6, the infinite-width limit of $G^{(0)}$ remains task-invariant, the geometry of the optimization landscape is not inherently altered by the permutations themselves. The fast set is therefore defined as $\mathcal{F} = \{q : \lambda_q(G_{\mathrm{LW}}) > \epsilon_{K,\gamma}\}$. Lemma 4.7 now implies a necessary condition for success on a new task: the initial residual must project negligibly onto the *slow* subspace. However, as the size of the fast set ($|\mathcal{F}|$) shrinks due to spectral collapse, this condition becomes statistically impossible to satisfy. This yields a direct mechanistic interpretation of Figure 2: across tasks, training accuracy is positively correlated with the $\epsilon$-rank ($|\{q : \lambda_q(G_{\mathrm{LW}}) > \epsilon_{K,\gamma}\}|$). $L2$ regularization acts as a counter-force by shifting $\lambda_q(G_{\mathrm{LW}})$ by a positive constant, ensuring more eigenvalues remain in $\mathcal{F}$. Conversely, low-rank feature initialization exacerbates the loss of plasticity by starting with fewer effective directions, thereby accelerating the reduction of $|\mathcal{F}|$.

## 6. Mitigating Spectral Collapse

Our NTK-inspired experiment suggests a practical alternative to directly analyzing the complicated Hessian of a nonlinear neural network. Rather than working with the full Hessian, we analyze training in the linearized regime through the Jacobian of model outputs and a function-space Gram operator. For general losses, the relevant function-space operator is the loss-weighted Gram

$$G_{\mathrm{LW}} \doteq C^{1/2} G^0 C^{1/2}, \quad G^0 \doteq JJ^\top. \qquad (10)$$

This operator has two crucial properties for our purposes: It is positive semidefinite for standard convex-in-logits losses,

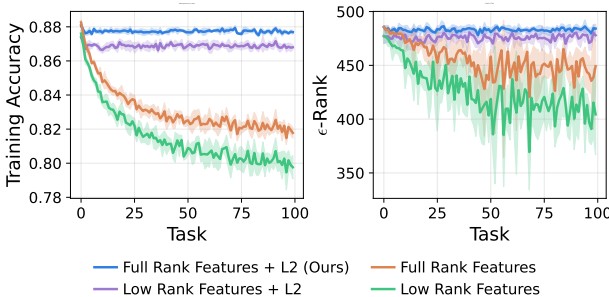

*Figure 2.* The left plot shows the training accuracy of the width 1000 linearized ReLU network. The right plot shows the $\epsilon_{K,\gamma}$-rank of $G_{\mathrm{LW}}$ with $K = |\text{train set}|$ and $\gamma = 1e^{-3}$. Accuracy and $\epsilon_{K,\gamma}$-rank are correlated.

and it is spectrally equivalent to the generalized Gauss Newton matrix $H_{\mathrm{GGN}} \doteq J^\top CJ$. Indeed, letting $A \doteq C^{1/2} J$ yields $G_{\mathrm{LW}} = AA^\top$, while $H_{\mathrm{GGN}} = A^\top A$, so they share the same nonzero eigenvalues. As a result, the $\epsilon$-rank of $G_{\mathrm{LW}}$ equals that of $H_{\mathrm{GGN}}$, but now expressed in a parameter space where it connects directly to curvature, optimization stability, and regularization. In Figure 3, we demonstrate the correspondence between successful training and the $\epsilon$-rank of the Hessian matrix in Continual ImageNet. Additional empirical findings appear in Sections H to J.

We therefore treat the $H_{\mathrm{GGN}}$ as the go-between linking the NTK function-space analysis to a parameter-space curvature target. This choice is also computationally motivated: unlike the true Hessian, $H_{\mathrm{GGN}}$ can be approximated efficiently using structured second-order methods (e.g., layerwise block and Kronecker-factor approximations), enabling us to define an explicit regularizer that increases the effective rank of the approximate curvature without ever materializing a the full Hessian, a $P \times P$ matrix.

In this section, we operationalize this idea: we build a scalable approximation of $H_{\mathrm{GGN}}$, identify which factors control its rank in MLPs, and introduce L2-ER as a simple objective that prevents curvature collapse. To do so, we first obtain the Gauss-Newton decomposition of the Hessian from Equation (1).

A neural network has parameters corresponding to each layer. Given a specific parameter, $[W^{[l]}]_{i,j}$ in layer $l$ and another parameter, $[W^{[l']}]_{i',j'}$, in $l' \neq l$), in practice $\partial^2 \mathcal{L} / \partial [W^{[l]}]_{i,j} \partial [W^{[l']}]_{i',j'}$ is approximately 0 (Becker et al., 1988), where $\mathcal{L}$ is the loss. Therefore, Hessian admits a natural "layer-wise block" structure.

Kronecker-factored approximate curvature (KFAC) is a method to approximate a matrix using only block-diagonal $\{M^l\}_{l \in L}$ elements. In particular, KFAC is frequently used

---

[3]The $x$-axis is *normalized $\epsilon$-rank($H$)*, meaning $\epsilon$-rank($H$) divided by $P$, for $\epsilon = 0.03$.

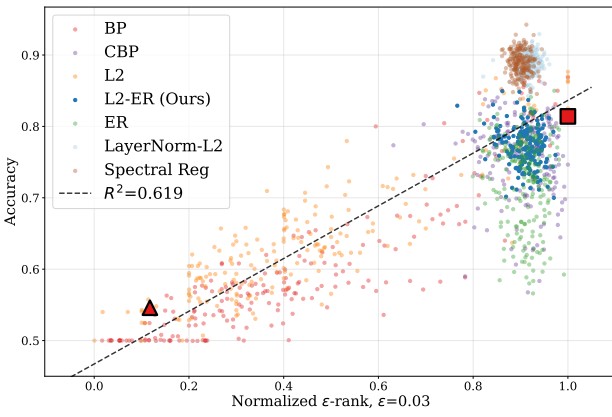

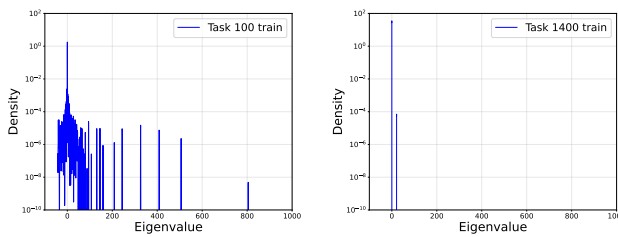

*Figure 3.* (Top) Final test accuracy vs. Curvature (i.e., normalized $\epsilon$-rank($\mathbf{H}$) at task initialization)[3] on Continual ImageNet. A linear fit (dotted line) highlights the positive association between curvature and accuracy. (Bottom Left) ■: BP Hessian eigenspectrum *before spectral collapse* at task 100. (Bottom Right) ▲: BP Hessian eigenspectrum *after spectral collapse* at task 1400.

for approximating the numerical quantities related to the Hessian like $\mathbf{H}_{\mathrm{GGN}}$ or Fisher Information matrices (Dangel et al., 2025; Martens, 2016). We therefore justify our effective-rank-based regularizer using a KFAC approximation of the Hessian. We then argue that increasing the rank of the KFAC-approximate $\mathbf{H}_{\mathrm{GGN}}$ implies increasing the rank of the true Hessian for mean-square-error (MSE) and softmax-cross-entropy losses (SCE).

As an analogue of a multi-layer MLP, we use a multi-layer linear neural network without bias to justify our regularizer. The layer $l$ in a neural network has an associated weight matrix $\mathbf{W}^{[l]}$. For a dataset of size $N$, let $\mathbf{a}_n^{[l-1]}$ be the input activations to the layer $l$ for the $n$th datum, producing pre-activation feature $\mathbf{h}_n^{[l]} = \mathbf{W}^{[l]}\mathbf{a}_n^{[l-1]}$. Let $\boldsymbol{\delta}_{n,o}^{[l]} = \nabla_{\mathbf{z}_n^{[l]}}[\ell(\mathbf{z}_n, \mathbf{y}_n)]_o$ be the gradient of the loss with respect to the layer's output $o$. We can now define $\mathbf{H}_{\mathrm{GGN}}$ as a sum of Kroneker products :

$$\mathbf{H}_{\mathrm{GGN}} = A \sum_{n=1}^{N} \sum_{o=1}^{O} (\mathbf{a}_n^{[l-1]}\mathbf{a}_n^{[l-1]\top}) \otimes (\boldsymbol{\delta}_{n,o}^{[l]}\boldsymbol{\delta}_{n,o}^{[l]\top}),$$

where $A$ is an accumulation factor[4] and $\otimes$ denotes the Kro-

---

[4]Typical choices are $1/n$, $1$, etc. (Dangel et al., 2025).

necker product operator. The term $\mathbf{a}_n^{[l-1]}\mathbf{a}_n^{[l-1]\top}$ captures the input feature statistics, while $\boldsymbol{\delta}_{n,o}^{[l]}\boldsymbol{\delta}_{n,o}^{[l]\top}$ captures the backpropagated output gradient statistics. We can now apply the computationally tractable KFAC approximation by decoupling the summations over the inputs and gradients (Dangel et al., 2025): $\mathbf{H}_{\mathrm{GGN}} \approx \widehat{\mathbf{H}}_{\mathrm{GGN}}$, where $\widehat{\mathbf{H}}_{\mathrm{GGN}}$ is

$$\left( A \sum_{n=1}^{N} \mathbf{a}_n^{[l-1]}\mathbf{a}_n^{[l-1]\top} \right) \otimes \left( \frac{1}{N} \sum_{n=1}^{N} \sum_{o=1}^{O} \boldsymbol{\delta}_{n,o}^{[l]}\boldsymbol{\delta}_{n,o}^{[l]\top} \right).$$

In our sample-based setting, the approximation error can be expressed as the difference between the average values of the Kronecker products and the Kronecker product of average values (see Section D):

$$\text{KFAC Error} = \left| \mathbf{H}_{\mathrm{GGN}} - \widehat{\mathbf{H}}_{\mathrm{GGN}} \right|.$$

The approximation error is small when the joint distribution over $\mathbf{a}^{[l]}$, $\mathbf{a}^{[l]\top}$, $\boldsymbol{\delta}^{[l]}$, and $\boldsymbol{\delta}^{[l]\top}$ is a multivariate Gaussian, which appears to hold in practice (Martens & Grosse, 2015). Alternatively, the approximation can be viewed as an assumption of statistical independence between $\mathbf{a}_n^{[l-1]}\mathbf{a}_n^{[l-1]\top}$ and $\boldsymbol{\delta}_n^{[l]}\boldsymbol{\delta}_n^{[l]\top}$, which holds for deep linear networks with squared-error loss (Bernacchia et al., 2018). Nonetheless, KFAC has been found to capture useful curvature information for convolutional neural networks (Grosse & Martens, 2016), recurrent neural networks (Martens et al., 2018) and vision-transformers (Eschenhagen et al., 2023).

**Maximizing the Rank of the Input Covariance Matrix** For each layer $l$, the KFAC approximation $\widehat{\mathbf{H}}_{\mathrm{GGN}}$ has the block form $\mathbb{E}_n[\mathbf{a}_n^{[l-1]}\mathbf{a}_n^{[l-1]\top}] \otimes \mathbb{E}_n[\boldsymbol{\delta}_n^{[l]}\boldsymbol{\delta}_{n\top}^{[l]}]$. Since $\mathrm{rank}(\boldsymbol{A} \otimes \boldsymbol{B}) = \mathrm{rank}\boldsymbol{A} \cdot \mathrm{rank}\boldsymbol{B}$, the rank of $\widehat{\mathbf{H}}_{\mathrm{GGN}}$ is monotone in the rank of both the input covariance matrix $\mathbb{E}_n[\mathbf{a}_n^{[l-1]}\mathbf{a}_n^{[l-1]\top}]$ and the gradient covariance matrix $\mathbb{E}_n[\boldsymbol{\delta}_n^{[l]}\boldsymbol{\delta}_{n\top}^{[l]}]$. The rank of $\mathbb{E}_n[\mathbf{a}_n^{[l-1]}\mathbf{a}_n^{[l-1]\top}]$ coincides with rank of the input representation of layer $l$ itself. This latter quantity, known as the *effective feature rank (ER)*, has been shown to be a useful indicator of plasticity (Dohare et al., 2024; Lyle et al., 2023), and standard (i.e., single-task) RL (Kumar et al., 2021).

**Maximizing the Rank of the Hessian** Adding $L2$ regularization is required to ensure the rank of the Hessian is sufficiently high, since $\mathbf{H}_{\mathrm{GGN}}$ does not take into account the contribution of the residual matrix $\mathbf{R}$. Thus, preserving high (effective) rank of the activation covariances (and avoiding degenerate gradient covariances) directly preserves the dimensionality of the non-collapsed curvature subspace captured by $\mathbf{H}_{\mathrm{GGN}}$, motivating an explicit effective-rank penalty on $\mathbb{E}_n[\mathbf{a}_n^{[l-1]}\mathbf{a}_n^{[l-1]\top}]$.

**L2-ER Regularization** Given task $\tau$, with data distribution $\mathbf{d}_\tau$ and loss function $\ell_\tau$, standard gradient descent mini-

mizes $\mathbb{E}_{\boldsymbol{d}_\tau}[\ell_\tau(F_{\boldsymbol{\theta}}(\boldsymbol{x}), \boldsymbol{y})]$. We propose the $L2$-ER regularizer, which combines effective rank and $L2$ penalties, leading to a theoretically grounded and practical objective:

$$\min_{\boldsymbol{\theta}} \mathcal{L}_\tau(\boldsymbol{\theta}) = \min_{\boldsymbol{\theta}} \underbrace{\mathbb{E}_{\boldsymbol{d}_\tau}\Big[\ell_\tau(F_{\boldsymbol{\theta}}(\boldsymbol{x}), \boldsymbol{y})\Big]}_{\text{Loss}} + \underbrace{\lambda \|\boldsymbol{\theta}\|_2^2}_{L2 \text{ regularization}}$$
$$- \beta \underbrace{\mathrm{erank}\left(\sum_{l \in L} \mathbb{E}_n\left[\mathbf{a}_n^{[l-1]} \mathbf{a}_n^{[l-1]\top}\right]\right)}_{\text{Effective rank penalty}}$$

In summary, we add an empirically computable effective rank penalty to the objective that increases the $\epsilon$-rank of the approximate GGN Hessian, $\widehat{\mathbf{H}}_{\mathrm{GGN}}$, which in turn affects the bulk of the eigenspectrum of the true Hessian.

## 7. Experiments

We conduct experiments across a range of network architectures and continual learning tasks adapted from supervised and reinforcement learning environments.

1. **Permuted MNIST** (Goodfellow et al., 2013): a variant of the MNIST dataset (LeCun, 1998) where new tasks are created by permuting the pixels in all images in the same way. Each permutation represents a new task. We model the classifier with a ReLU MLP.

2. **Continual Imagenet** (Dohare et al., 2024): an adaptation of Imagenet (Krizhevsky et al., 2012) where pairs of classes are randomly sampled for binary classification. Distinguishing the two classes in each new class pair is a new task. We use a convolutional neural network.

3. **Incremental CIFAR** (Dohare et al., 2024): an adaptation of the CIFAR-100 dataset (Krizhevsky et al., 2009) where classes are added incrementally. Starting with 5 classes, each task adds 5 new classes until all 100 classes are included. We use a ResNet architecture (He et al., 2015a).

4. **Slippery Ant** (Dohare et al., 2024): a continual reinforcement learning environment where the friction between the ant and the ground changes every $T$ timesteps, forcing the agent to continually adapt its policy.

We compare L2-ER against six baselines: vanilla backpropagation (BP), L2 regularization (L2), effective rank regularization (ER), continual backpropagation (Dohare et al., 2024) (CBP), layer-normalization with L2 (Lyle et al., 2024) (LayerNorm-L2), and spectral regularization (Lewandowski et al., 2024a) (Spectral). For Incremental CIFAR, where task difficulty increases with the number

of classes, we include a RESET baseline that reinitializes parameters at each task change; algorithms outperforming RESET are considered to maintain plasticity. We perform extensive hyperparameter sweeps and report accuracy for supervised learning and online returns for RL. Implementation details appear in Sections H to K.

**Performance** Figure 4 depicts results across all four environments. L2-ER maintains plasticity in all, and provides an early performance boost on Permuted MNIST before plasticity loss would typically occur. L2 preserves plasticity in most cases but fails on Continual ImageNet; CBP succeeds except on Incremental CIFAR. LayerNorm-L2 performs well except on Permuted MNIST. Spectral Regularization succeeds except on Incremental CIFAR but requires roughly twice the training time (Figure 15). ER and BP exhibit plasticity loss across all environments.

## 8. Discussion

Our goal in this work is to analyze continual learning with a simplified model of a neural network, to find and isolate the mechanisms that contribute to loss of plasticity. Our model of choice is the NTK, which approximates neural network functions by linearizing around initialization. We show that gradient descent applied to the linearized model under the squared error loss progressively reduces the components of the error residual along directions associated with large eigenvalues of the Gram matrix $\mathbf{G} = \mathbf{J}\mathbf{J}^\top$. As a result, components aligned with small eigenvalue directions decay slowly over a finite training horizon. This leads to a simple constraint on learnability: if the new-task residual has substantial mass in these slow directions, then it cannot be reduced sufficiently within the allotted $K$ gradient steps, unless it is already small or happens to align with the fast directions. In this sense, successful learning requires that most of the residuals lie in directions where the model can make rapid progress.

However, in the NTK regime under squared loss, the kernel matrix, $\mathbf{G} = \mathbf{J}\mathbf{J}^\top$ remains invariant over training. As a result, we shift our attention to classification, where the weighted Gram matrix $\mathbf{G}_{\mathrm{LW}} = \mathbf{C}^{1/2}\mathbf{G}^{(0)}\mathbf{C}^{1/2}$ governs training dynamics. This distinction is crucial on Permuted MNIST. We prove that, in the infinite-width frozen-gate model, the raw kernel $\mathbf{G}^{(0)}$ remains invariant to pixel permutations. Thus, the permutations themselves alone are not enough to explain spectral collapse. Empirically, however, the loss-weighted Gram matrix, $\mathbf{G}_{\mathrm{LW}}$ does undergo spectral collapse, and its $\epsilon$-rank tracks the loss of plasticity due to the loss-weighted geometry that governs classification dynamics.

Having built an understanding of the mechanisms behind plasticity loss using the NTK, we apply our findings to

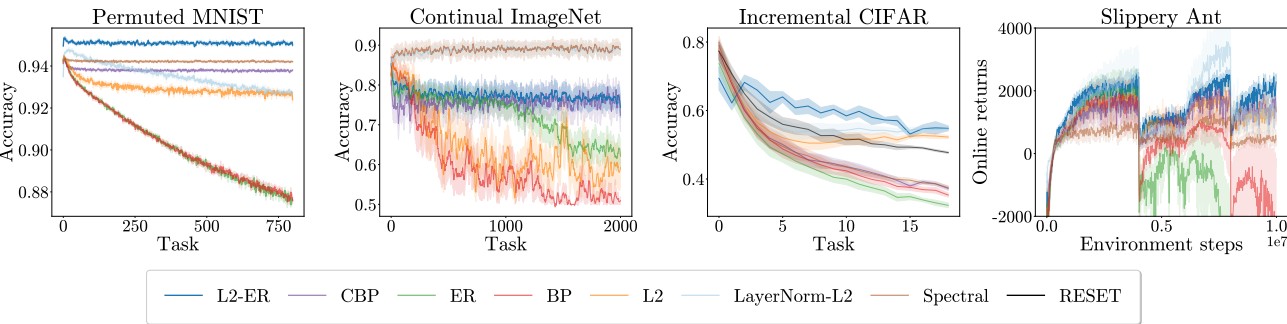

*Figure 4.* Performance across all four environments. Classification accuracy is reported for Permuted MNIST, Continual ImageNet, and Incremental CIFAR, while online returns are reported for Slippery Ant. Results are averaged across 5 seeds for all environments, except Continual ImageNet, where we used 10 seeds due to higher variance. Shaded regions denote a 95% confidence interval.

conventional deep non-linear networks. In particular, we show that the loss-weighted Gram matrix $\mathbf{G}_{\mathrm{LW}}$, is spectrally equivalent to the generalized Gauss Newton matrix, $\mathbf{H}_{\mathrm{GGN}}$. Consequently, we propose the $L2$-ER regularizer, which aims to force a wide-bulk to the Hessian eigenspectrum to maintain plasticity. We measure the Hessian spectrum at new-task initialization and observe that eigenspectrum collapse precedes failures to learn. When we apply our intervention, the spectrum of the Hessian retains a wide bulk and maintains plasticity. When we remove the intervention, the bulk collapses and so does plasticity. Moreover, we find that other plasticity preserving methods, like CBP also maintain a wide bulk.

$L2$-ER is much cheaper than explicitly estimating the full Hessian spectrum, and the eigenspectrum diagnostics are not used inside the regularizer. However, the effective-rank penalty does introduce a nontrivial SVD cost, which we amortize over $U$ steps. For each layer $l \in L_{ER}$ in the network to be regularized, we collect the activations $\mathbf{a}^{[l]} \in \mathbb{R}^{d^l}$, and stack them row-wise over a batch of data $B$ as $\mathbf{A}_{\mathrm{ER}}^{[l]} = \left[ \mathbf{a}_1^{[l]} \cdots, \mathbf{a}_B^{[l]} \right]^{\top} \in \mathbb{R}^{B \times d^l}$. The amortized SVD cost of $\mathbf{A}_{\mathrm{ER}}^{[l]}$ is $\sum_{l \in L_{\mathrm{ER}}} O\left(\frac{1}{U} \min\{B(d^l)^2, \ B^2 d^l\}\right)$ with memory $O(B d^l)$.

In our experiments, small feature buffers were sufficient, and the ER learning rate was the most sensitive ER hyperparameter; the ER batch/window mainly stabilized the singular-value estimate. We recommend tuning the task optimizer first, then the L2 coefficient, and finally the ER learning rate, typically below the task learning rate. The ER buffer should be the smallest one that gives a stable singular-value estimate, and ER updates should be applied as infrequently as possible while retaining their benefit. For larger models, selected-layer regularization, bottleneck-layer regularization, randomized or truncated SVD, and sketching methods may be needed. We do not yet have language-model results; extending $L2$-ER to such models will likely require applying the penalty only to selected rep-

resentations, adapters, or other low-dimensional modules, and attention layers may require a more specialized curvature approximation.

We believe our work opens several directions for future work. Having identified when a new task becomes unlearnable, our causal claim that spectral collapse drives loss of plasticity is empirical. Future work could use tools from singular learning theory (Watanabe, 2009) or stochastic gradient dynamics (Wu et al., 2018; Wu & Su, 2023; Xu et al., 2026) to understand the evolution of the Hessian eigenspectrum in continual learning.

## 9. Conclusion

We identify Hessian spectral collapse as the central mechanism for plasticity loss in continual learning. Analyzing a linearized ReLU network, we derive explicit $\epsilon$-rank conditions for successful training and establish that the loss-weighted Gram matrix is spectrally equivalent to the Generalized Gauss-Newton approximation, bridging NTK dynamics to parameter-space curvature. Using the KFAC approximation of the Hessian, we propose $L2$-ER, a simple regularizer that empirically prevents spectral collapse and preserves plasticity across diverse benchmarks. These results position spectral collapse as both a unifying explanation and a practical diagnostic for plasticity loss. We leave adapting our theoretical framework to the unique dynamics of RL, where the Hessian is influenced by non-stationary policy and value functions, for future work. The implications of this work extend to Silver & Sutton's (2025) long-term vision for adaptive AI systems. Consider a large language model or robot acting as an assistant on a multi-year long project. By providing a stable foundation for continual learning, our findings represent a crucial step towards building more capable and continuously adapting AI systems of the future.

## Acknowledgments

Arjun Prakash and Ruo Yu Tao were supported by the Office of Naval Research (ONR) grant N00014-22-1-2592 and N00014-24-1-2657. Saket Tiwari was supported by ONR REPRISM MURI N00014-24-1-2603, ONR grant 00014-22-1-2592 and partial funding was also provided by the Robotics and AI Institute.

## Impact Statement

This paper presents work whose goal is to advance the field of Machine Learning. There are many potential societal consequences of our work, none which we feel must be specifically highlighted here.

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

# A. Summary of Contributions

We describe the contributions of each author following the CRediT Taxonomy (Hosseini et al., 2026).

**Arjun** contributed to the conceptualization and the investigation of the Hessian. He was the primary contributor to the NTK analysis and the KFAC interpretation of the the $L2$-ER regularizer. He contributed to the original draft, and the final revision by adapting the NTK analysis and responding to reviewers. He also led the general project administration.

**Naicheng** was the primary contributor to the investigation and conceptualization of spectral collapse. He was the primary contributor to the $L2$-ER regularizer and confirmed its practicality together with Kaicheng. He led the project administration for the NeurIPS ARLET workshop submission and contributed to the original draft.

**Kaicheng** was a contributor to the $L2$-ER regularizer and to confirming its practical effectiveness across continual learning settings. He led the development of the entire codebase and was responsible for the main experimental pipeline. He led the empirical evaluation of the paper, including the performance experiments across all benchmarks and the Hessian spectrum analyses used to diagnose spectral collapse. His works provide the main experimental evidence supporting the paper's claims.

# B. Neural Tangent Kernel Analysis

## B.1. Introduction on NTKs

For completeness we provide a short introduction to NTKs. The neural tangent kernel (NTK) has been a popular method to characterize the training dynamics of (stochastic) gradient descent with a infinitesimally small learning rates and the infinite-width limit (i.e., $M \to \infty$). The main principle behind NTK theory is to approximate the neural network function by linearizing it around it's parameters around the initialization. The seminal work (Jacot et al., 2018) show that if $F$ is an appropriately scaled neural network, with $\boldsymbol{\theta}$ initialized i.i.d. from standard (or isotropic) Gaussian's (i.e. $\boldsymbol{\theta} \sim \mathcal{N}(0, \boldsymbol{I})$) then:

$$\hat{F}_{\boldsymbol{\theta}}(\boldsymbol{X}_\tau) \approx F_{\boldsymbol{\theta}_\tau^{(0)}}(\boldsymbol{X}_\tau) + \mathbf{J}_\tau(\boldsymbol{\theta} - \boldsymbol{\theta}_\tau^{(0)}).$$

The linearized model $\hat{F}_{\boldsymbol{\theta}}$ can be interpreted as a kernel method where the Jacobian is the feature map:

$$\boldsymbol{\phi}_{\boldsymbol{\theta}^{(0)}}(\boldsymbol{X}) \doteq \left. \frac{\partial F_{\boldsymbol{\theta}}(\boldsymbol{X})}{\partial \boldsymbol{\theta}} \right|_{\boldsymbol{\theta} = \boldsymbol{\theta}^{(0)}} = \mathbb{R}^{NO \times P}. \tag{11}$$

The kernel induced by the feature map is the NTK and is dependent on the random initialization $\boldsymbol{\theta}^{(0)}$. The neural tangent kernel $\kappa : \mathcal{X} \times \mathcal{X} \to \mathbb{R}^{O \times O}$ is :

$$\kappa_{\boldsymbol{\theta}}(\boldsymbol{x}, \boldsymbol{x}') \doteq \mathbb{E}_{\boldsymbol{\theta} \sim \mathcal{N}(0, \boldsymbol{I})} \left[ \left\langle \frac{\partial F_{\boldsymbol{\theta}}(\boldsymbol{x})}{\partial \boldsymbol{\theta}}, \frac{\partial F_{\boldsymbol{\theta}}(\boldsymbol{x}')}{\partial \boldsymbol{\theta}} \right\rangle \Big|_{\boldsymbol{\theta} = \boldsymbol{\theta}^{(0)}} \right] \tag{12}$$

$$= \mathbb{E}_{\boldsymbol{\theta}^{(0)} \sim \mathcal{N}(0, \boldsymbol{I})} \left[ \langle \nabla_{\boldsymbol{\theta}} F_{\boldsymbol{\theta}^{(0)}}(\boldsymbol{x}), \nabla_{\boldsymbol{\theta}} F_{\boldsymbol{\theta}^{(0)}}(\boldsymbol{x}') \rangle \right]. \tag{13}$$

The evaluating the kernel on a finite dataset (i.e. for all pairs $(\boldsymbol{x}, \boldsymbol{x}') \in \boldsymbol{X}$) gives the empirical NTK matrix:

$$\kappa_{\boldsymbol{\theta}}(\boldsymbol{x}, \boldsymbol{x}') = \mathbf{J}\mathbf{J}^\top = \mathbf{G} \in \mathbb{R}^{NO \times NO}. \tag{14}$$

The celebrated result from NTK analysis is that as $M \to \infty$, the NTK, $\kappa_{\boldsymbol{\theta}}$ converges in probability to an explicit deterministic limit $\kappa_{\boldsymbol{\theta}}^\infty$ (Jacot et al., 2018, Theorem 1) where:

$$\kappa_{\boldsymbol{\theta}}^\infty \doteq \mathbb{E}_{\boldsymbol{\theta} \sim \mathcal{N}(0, \boldsymbol{I})} [\mathbf{G}]. \tag{15}$$

This deterministic limit only depends on the variance parameter of the Gaussian initialization distribution and the choice of non-linearity. The NTK limit holds for ReLU activations (Arora et al., 2019) and LeCun initialization (Jacot et al., 2018).

### B.2. Linearized ReLU networks

We now instantiate the NTK framework for the exact experimental model used in our continual learning experiments: a linearized ReLU network (Dwaraknath et al., 2023). Let the input dimension be $I$, width be $M$, and output dimension be $O$. Denote the first-layer weights with $\mathbf{W}^{[1]} = [\mathbf{w}_1^{[1]}, \ldots, \mathbf{w}_M^{[1]}] \in \mathbb{R}^{I \times M}$ and the fixed output-layer weights with $\mathbf{W}^{[2]} = [\mathbf{w}_1^{[2]}, \ldots, \mathbf{w}_M^{[2]}]^\top \in \mathbb{R}^{M \times O}$ (so each hidden unit $h$ has output row $(\mathbf{w}_h^{[2]})^\top \in \mathbb{R}^O$). A standard linearized ReLU network is:

$$\hat{F}^{\boldsymbol{\theta}}(\boldsymbol{x}) \doteq \frac{1}{\sqrt{M}} \sum_{h=1}^M \mathbf{w}_h^{[2]} \sigma((\mathbf{w}_h^{[1]})^\top \boldsymbol{x}), \qquad \sigma(x) = \max\{0, x\}. \tag{16}$$

**Linearized ReLU Network**   Let $\mathbf{W}^{[1](0)} \in \mathbb{R}^{I \times M}$ be a frozen reference copy of the initialized first-layer matrix $\mathbf{W}^{[1]}$. With this fixed initialization, we define the fixed ReLU gate function and gate matrix respectively:

$$g(\boldsymbol{x}) \doteq \mathbf{1}\{(\mathbf{w}_h^{[1](0)})^\top \boldsymbol{x} > 0\} \text{ for } h \in 1, .., M \qquad \mathbf{D}_g(\boldsymbol{x}) \doteq \mathrm{diag}(g(\boldsymbol{x})). \tag{17}$$

Note that the gates $g(\boldsymbol{x})$ (equivalently, $\mathbf{D}_g(\boldsymbol{x})$) and the output matrix $\mathbf{W}^{[2]}$ remain frozen throughout training. The only trainable parameters are $\mathbf{W}^{[1]} \in \mathbb{R}^{d \times M}$, initialized at $\mathbf{W}^{[1]} = \mathbf{W}^{[1](0)}$. Thus, the linearized activations are:

$$F_{\mathbf{W}^{[1]}}(\boldsymbol{x}) \doteq \sigma((\mathbf{W}^{[1](0)})^\top \boldsymbol{x}) + g(\boldsymbol{x}) \odot ((\mathbf{W}^{[1]})^\top \boldsymbol{x} - (\mathbf{W}^{[1](0)})^\top \boldsymbol{x}) \tag{18}$$

$$= g(\boldsymbol{x}) \odot ((\mathbf{W}^{[1]})^\top \boldsymbol{x}) \tag{19}$$

$$= \mathbf{D}_g(\boldsymbol{x})(\mathbf{W}^{[1]})^\top \boldsymbol{x}, \tag{20}$$

where we used $\sigma((\mathbf{W}^{[1](0)})^\top \boldsymbol{x}) = g(\boldsymbol{x}) \odot (\mathbf{W}^{[1](0)})^\top \boldsymbol{x}$. The full linearized network the model prediction is

$$F_{\boldsymbol{\theta}}(\boldsymbol{x}) = F_{\mathbf{W}^{[1]}}(\boldsymbol{x})^\top \mathbf{W}^{[2]} = \mathbf{W}^{[2]} \mathbf{D}_g(\boldsymbol{x})(\mathbf{W}^{[1]})^\top \boldsymbol{x} \in \mathbb{R}^O. \tag{21}$$

We denote this specific instantiation of the gated-linear tangent kernel (GLTK) with $\chi : \boldsymbol{x} \times \boldsymbol{x} \to \mathbb{R}^{O \times O}$. Specifically, for output coordinates $o, o' \in [O]$:

$$\chi(\boldsymbol{x}, \boldsymbol{x}')_{(o, o')} \doteq \left\langle \frac{\partial F_o(\boldsymbol{x})}{\partial \mathbf{W}^{[1]}}, \frac{\partial F_{o'}(\boldsymbol{x}')}{\partial \mathbf{W}^{[1]}} \right\rangle_{\mathrm{F}}, \tag{22}$$

Given the dataset $\{\boldsymbol{x}^1, .., \boldsymbol{x}^n\}$, using $\partial F_o(\boldsymbol{x})/\partial \mathbf{w}_h^{[1]} = [\mathbf{w}_h^{[2]}]_o \, g_h(\boldsymbol{x}) \, \boldsymbol{x}$, yields the Gram matrix:

$$[\chi(\boldsymbol{x}^n, \boldsymbol{x}^m)]_{o, o'} \doteq [\mathbf{G}]_{(n,o),(m,o')} \doteq = \langle \boldsymbol{x}^n, \boldsymbol{x}^m \rangle \frac{1}{M} \sum_{h=1}^M \left( [\mathbf{w}_h^{[2]}]_o [\mathbf{w}_h^{[2]}]_{o'} \, g_h(\boldsymbol{x}^n) \, g_h(\boldsymbol{x}^m) \right). \tag{23}$$

Furthermore, the taking $M \to \infty$, the infinite width limit is:

$$[\chi_\infty]_{o, o'}(\boldsymbol{x}_n, \boldsymbol{x}_m) \doteq \mathbf{E}_{\mathbf{W}^{[1](0)}, \mathbf{W}^{[2]}}[\mathbf{G}] = \mathbf{E}_{\mathbf{W}^{[1](0)}, \mathbf{W}^{[2]}} \Big[ \langle \boldsymbol{x}^n, \boldsymbol{x}^m \rangle \sum_{h=1}^M [\mathbf{w}_h^{[2]}]_o [\mathbf{w}_h^{[2]}]_{o'} \, g_h(\boldsymbol{x}^n) \, g_h(\boldsymbol{x}^m) \Big]. \tag{24}$$

**Relation to NTK linearization.**   Consider the full two-layer network in Equation (16) with frozen $V$ and linearize in $W$ around $W^{(0)}$:

$$\hat{F}_{\boldsymbol{\theta}}(\boldsymbol{X}_\tau) \approx F_{\boldsymbol{\theta}_\tau^{(0)}}(\boldsymbol{X}_\tau) + \mathbf{J}_\tau(\boldsymbol{\theta} - \boldsymbol{\theta}_\tau^{(0)}).$$

In particular, for output coordinate $o \in [O]$.

$$\frac{\partial F_{\boldsymbol{\theta}}(\boldsymbol{x})}{\partial \mathbf{W}^{[1]}} = \frac{1}{\sqrt{M}} \boldsymbol{x}(g \odot [\mathbf{W}^{[2]}]_{:,o})^\top. \tag{25}$$

In our construction Equation (20)–Equation (21), this linearization is *exact globally*: once gates are frozen, $F_{\boldsymbol{\theta}}(\boldsymbol{x})$ is linear in $\mathbf{W}^{[1]}$ for all $\mathbf{W}^{[1]}$, not just locally around $\mathbf{W}^{[1](0)}$. Therefore, Assumption C.1 holds with equality for this experimental model.

# C. NTK Mathematical Results

Our detailed mathmatical results rely on the following assumptions:

**Assumption C.1** (Local Linearization). For task $\tau$, during the next $K$ gradient descent steps starting at $\boldsymbol{\theta}_\tau^{(0)}$, we approximate:

$$\hat{F}_{\boldsymbol{\theta}}(\boldsymbol{X}_\tau) \approx F_{\boldsymbol{\theta}_\tau^{(0)}}(\boldsymbol{X}_\tau) + \mathbf{J}_\tau(\boldsymbol{\theta} - \boldsymbol{\theta}_\tau^{(0)}), \tag{26}$$

where the Jacobian evaluated on the dataset/batch is:

$$\mathbf{J} \doteq \frac{\partial F_{\boldsymbol{\theta}}(\boldsymbol{X}_\tau)}{\partial \boldsymbol{\theta}}\Bigg|_{\boldsymbol{\theta}=\boldsymbol{\theta}_\tau^{(0)}} \in \mathbb{R}^{N\times P}, \tag{27}$$

is treated as constant over the window.

For the continual NTK, $\mathbf{G} = \mathbf{J}\mathbf{J}^\top \in \mathbb{R}^{N\times N}$, we will linearize around $\boldsymbol{\theta}_0 = \boldsymbol{\theta}_{\tau+1}^{(0)}$.

**Assumption C.2** (Step size stability). The learning rate $\eta$ is small enough such that $\boldsymbol{I} - \eta\mathbf{G}_\tau$ is non-expansive (i.e., $\eta\lambda_{\max}(\mathbf{G}_\tau) \leq 1$).

In addition, we also assume that our model linear model is initialized from i.i.d. standard Gaussian distributions as an NTK.

**Lemma 4.2** (Residual dynamics in the linearized regime). *If $\hat{F}_{\boldsymbol{\theta}}(\boldsymbol{X}_\tau) \approx F_{\boldsymbol{\theta}_\tau^{(0)}}(\boldsymbol{X}_\tau) + \mathbf{J}_\tau(\boldsymbol{\theta} - \boldsymbol{\theta}_\tau^{(0)})$, then gradient descent on $\ell_\tau(\boldsymbol{\theta})$ with learning rate $\eta$ yields as the $k+1$th iterate $r_\tau^{(k+1)} = (\boldsymbol{I} - \eta\mathbf{G}_\tau)r_\tau^{(k)}$, where $\mathbf{G}_\tau = \mathbf{J}_\tau\mathbf{J}_\tau^\top \in \mathbb{R}^{NO\times NO}$. Hence, the final value of the residual for task $\tau$ is $r_\tau^{(K)} = (\boldsymbol{I} - \eta\mathbf{G}_\tau)^K r_\tau^{(0)}$.*

*Proof.* Using *Assumption C*.1, $\hat{F}_{\boldsymbol{\theta}}(\boldsymbol{X}_\tau) \approx F_{\boldsymbol{\theta}_\tau^{(0)}}(\boldsymbol{X}_\tau) + \mathbf{J}_\tau(\boldsymbol{\theta} - \boldsymbol{\theta}_\tau^{(0)})$, so we can approximate the residual with the following:

$$r_\tau(\boldsymbol{\theta}) = F_{\boldsymbol{\theta}}(\boldsymbol{X}_\tau) - \boldsymbol{Y}_\tau \tag{28}$$

$$\approx F_{\boldsymbol{\theta}_\tau^{(0)}}(\boldsymbol{X}_\tau) - \boldsymbol{Y}_\tau + \mathbf{J}(\boldsymbol{\theta} - \boldsymbol{\theta}_\tau^{(0)}) \tag{29}$$

$$= r_\tau^{(0)} + \mathbf{J}(\boldsymbol{\theta} - \boldsymbol{\theta}_\tau^{(0)}). \tag{30}$$

For the squared error loss $\ell = \frac{1}{2}\|r(\boldsymbol{\theta})\|^2$, with linear model $r(\boldsymbol{\theta}) = r_\tau^{(0)} + \mathbf{J}(\boldsymbol{\theta} - \boldsymbol{\theta}_\tau^{(0)})$ the gradient is:

$$\nabla_{\boldsymbol{\theta}}\ell_\tau(\boldsymbol{\theta}) = \mathbf{J}_\tau^\top r(\boldsymbol{\theta}), \tag{31}$$

which means the gradient descent update is:

$$\boldsymbol{\theta}_\tau^{(k+1)} = \boldsymbol{\theta}_\tau^{(k)} - \eta\mathbf{J}_\tau^\top r_\tau^{(k)}. \tag{32}$$

Now using the fact that $r_\tau^{(k+1)} \approx r_\tau^{(k)} + \mathbf{J}_\tau(\boldsymbol{\theta}^{(k+1)} - \boldsymbol{\theta}^{(k)})$, we obtain :

$$r_\tau^{(k+1)} \approx r_\tau^{(k)} + \mathbf{J}_\tau(\boldsymbol{\theta}^{(k+1)} - \boldsymbol{\theta}^{(k)}) \tag{33}$$

$$= r_\tau^{(k)} + \mathbf{J}_\tau(-\eta\mathbf{J}_\tau^\top r_\tau^{(k)}) \tag{34}$$

$$= (\boldsymbol{I} - \eta\mathbf{J}_\tau\mathbf{J}_\tau^\top)r_\tau^{(k)} \tag{35}$$

$$= (\boldsymbol{I} - \eta\mathbf{G}_\tau)r_\tau^{(k)} \tag{36}$$

Which we can repeatedly apply using *assumption C*.1 to obtain the final result. $\qquad\square$

**Lemma 4.3** (Eigenvalue Decomposition). *Assume $\mathbf{G}_\tau$ has eigenpairs $\{\lambda_{\tau,q}, \mathbf{v}_{\tau,q}\}_{q=1}^N$ with orthonormal $\{\mathbf{v}_{\tau,q}\}$ and $\lambda_{\tau,q} \geq 0$. Expand the initial residual:*

$$r_\tau^{(0)} = \sum_{q=1}^n \alpha_{\tau,q} \mathbf{v}_{\tau,q}, \quad \alpha_{\tau,q} \doteq (\mathbf{v}_{\tau,q})^\top r_\tau^{(0)}. \tag{4}$$

*Then, for all iterates $k, r_\tau^{(k)} = \sum_{q=1}^n (1 - \eta\lambda_{\tau,q})^k \alpha_{\tau,q} \mathbf{v}_{\tau,q}$.*

*Proof.* First, we expand $r_\tau^{(0)}$ in its eigenbasis:

$$r_\tau^{(0)} = \sum_{i=1}^n \alpha_{\tau,q} \mathbf{v}_{\tau,q}, \quad \alpha_{\tau,q} \doteq (\mathbf{v}_{\tau,q})^\top r_\tau^0, \tag{37}$$

and

$$r_\tau^{(k)} = (\boldsymbol{I} - \eta\mathbf{G}_\tau)^k \left( \sum_{q=1}^n \alpha_{\tau,q} \mathbf{v}_{\tau,q} \right) \tag{38}$$

Using the identity of eigenpairs: $\mathbf{G}_\tau \mathbf{v}_{\tau,q} = \lambda_{\tau,q} \mathbf{v}_{\tau,q}$ which implies that:

$$(\boldsymbol{I} - \eta\mathbf{G}_\tau)\mathbf{v}_{\tau,q} = \boldsymbol{I}\mathbf{v}_{\tau,q} - \eta(\mathbf{G}_\tau \mathbf{v}_{\tau,q}) \tag{39}$$
$$= 1 \cdot \mathbf{v}_{\tau,q} - \eta(\lambda_{\tau,q} \mathbf{v}_{\tau,q}) \tag{40}$$
$$= (1 - \eta\lambda_{\tau,q})\mathbf{v}_{\tau,q}. \tag{41}$$

For any power $k$

$$(\boldsymbol{I} - \eta\mathbf{G}_\tau)^k \mathbf{v}_{\tau,q} = (1 - \eta\lambda_{\tau,q})^k \mathbf{v}_{\tau,q}, \tag{42}$$

since $\lambda_{\tau,q}$ is an eigenvector of $\mathbf{G}_\tau$, it is also an eigenvalue of the polynomial $(\boldsymbol{I} - \eta\mathbf{G}_\tau)$, and by induction of $(\boldsymbol{I} - \eta\mathbf{G}_\tau)^k$ with eigenvalue $(\boldsymbol{I} - \eta\lambda_{\tau,q})^k$.

We can now plugging this back into *eq.* (38):

$$r_\tau^{(k)} = \sum_{q=1}^n (1 - \eta\lambda_{\tau,q})^k \alpha_{\tau,q} \mathbf{v}_{\tau,q}. \tag{43}$$

and squaring yields the desired result. □

### C.1. Fast and slow modes

We can now formalize *fast* and *slow* with an explicit threshold derived from $K$ and $\eta$. Pick $\gamma \in (0,1)$.

**Definition C.3.** A *fast mode* should shrink by at least a factor of $\gamma$ after $K$ steps. That is: if a mode is fast $(1 - \eta\lambda)K \leq \gamma$.

Solve for $\lambda$:

$$(1 - \eta\lambda)^K \leq \gamma \iff 1 - \eta\lambda \leq \gamma^{1/K} \iff \lambda \geq \epsilon_{K,\gamma}, \tag{44}$$

where

$$\epsilon_{K,\gamma} \doteq \frac{1 - \gamma^{1/K}}{\eta} \approx \frac{-\log\gamma}{\eta K}. \tag{45}$$

Given eigenvalues $\lambda_\tau^i$, define the fast set:

$$\mathcal{F}_\tau \doteq \{q : \lambda_{\tau,q} \geq \epsilon_{K,\gamma}\}, \tag{46}$$

and slow set:

$$\mathcal{S}_\tau \doteq \{q : \lambda_{\tau,q} < \epsilon_{K,\gamma}\}. \tag{47}$$

The $\epsilon$-rank can now be defined as $|\mathcal{F}|$.

Finally, we define the fast projector:

$$\mathbf{P}_\tau^{\mathcal{F}} \doteq \sum_{i \in \mathcal{F}_\tau} \mathbf{v}_{\tau,q} \mathbf{v}_{\tau,q}^\top, \tag{48}$$

and slow projector:

$$\mathbf{P}_\tau^{\mathcal{S}} \doteq \boldsymbol{I} - \mathbf{P}_\tau^{\mathcal{F}} = \sum_{i \in \mathcal{S}_\tau} \mathbf{v}_{\tau,q} \mathbf{v}_{\tau,q}^\top. \tag{49}$$

**Lemma 4.6.** *For any task* $\tau$,

$$\left\| r_\tau^{(K)} \right\| \geq \left\| \mathbf{P}_\tau^{\mathcal{S}} r_\tau^{(K)} \right\| \geq \gamma \left\| \mathbf{P}_\tau^{\mathcal{S}} r_\tau^{(0)} \right\|. \tag{5}$$

*Proof.* From Lemma 4.3,

$$\mathbf{P}^{\mathcal{S}} = \sum_{i \in \mathcal{S}} (1 - \eta\lambda_i)^K \alpha^i \mathbf{v}^i. \tag{50}$$

If $i \in \mathcal{S}_\tau$, then $\lambda^i < \epsilon_{K,\gamma}$, hence

$$1 - \eta\lambda^i > 1 - \eta\epsilon_{K,\gamma} = \gamma^{1/K} \implies (1 - \eta\lambda^i)^K > \gamma. \tag{51}$$

Therefore, each slow coefficient satisfies $|(1 - \eta\lambda^i)^K \alpha^i \geq \gamma|\alpha^i|$. Squaring and summing gives:

$$\|\mathbf{P}_\tau^{\mathcal{S}} r^{(K)}\|^2 = \sum_{i \in \mathcal{S}} (1 - \eta\lambda^i)^{2K} (\alpha^i)^2 \geq \sum_{i \in \mathcal{S}} \gamma^2 (\alpha^i)^2 = \gamma^2 \|\mathbf{P}^{\mathcal{S}} r^{(0)}\|^2. \tag{52}$$

Taking square roots closes the proof. $\square$

*Remark* C.4. Slow modes are sticky. If the eigenvalues are in the slow-regime, after $K$ steps, the NTK still retains at least $\gamma$ fraction of the eigenvalues less than $\epsilon_{K,\gamma}$.

### C.2. Continual Case

Now we will connect slow-stickiness to the success criterion. For the squared loss on task $\tau + 1$,

$$\ell_{\tau+1}(\boldsymbol{\theta}_{\tau+1}^{(K)}) = \frac{1}{2} \|r_{\tau+1}^{(K)}\|^2. \tag{53}$$

.

Now, we can define a success criterion if the final loss is at most $\upsilon$:

$$\text{Success}_{\tau+1} \doteq \left\{ \ell_\tau(\boldsymbol{\theta}_{\tau+1}^{(K)}) \leq \upsilon \right\} = \left\{ \|r_{\tau+1}^{(K)}\| \leq \sqrt{2\upsilon} \right\}. \tag{54}$$

**Lemma 4.7** (Necessary fast-projection condition). *Using the fast and slow projections, $\epsilon_{K,\gamma}$, and the initial loss residual $r_\tau^{(0)}$, a necessary condition to achieve successful training on task $\tau$ is $\gamma \left\| \mathbf{P}_\tau^{\mathcal{S}} r_\tau^{(0)} \right\| \leq \sqrt{2v}$. Equivalently,*

$$\left\| \mathbf{P}_\tau^{\mathcal{F}} \frac{r_\tau^{(0)}}{\|r_\tau^{(0)}\|} \right\|^2 \geq 1 - \left( \frac{\sqrt{2v}}{\left\| \gamma r_\tau^{(0)} \right\|^2} \right) \tag{6}$$

*Proof.* From Lemma 4.6 applied to task $\tau + 1$:

$$\|r_{\tau+1}^{(K)}\| \geq \gamma \|\mathbf{P}_{\tau+1}^{\mathcal{S}} r_{\tau+1}^{(0)}\| \tag{55}$$

If success holds, then $\|\mathbf{P}_{\tau+1}^{\mathcal{S}} r_\tau^{(0)}\| \leq \sqrt{2v}$, so combining:

$$\sqrt{2v} \geq \|r_{\tau+1}^{(K)}\| \geq \gamma \|\mathbf{P}_{\tau+1}^{\mathcal{S}} r_{\tau+1}^{(0)}\| \implies \|\mathbf{P}_{\tau+1}^{\mathcal{S}} r_{\tau+1}^{(0)}\| \leq \frac{\sqrt{2v}}{\gamma}. \tag{56}$$

Now define $V = \frac{r_{\tau+1}^{(0)}}{\|r_{\tau+1}^{(0)}\|_2}$,

$$\|\mathbf{P}_{\tau+1}^{(\mathcal{S})} r_{\tau+1}^{(0)}\|^2 = \|r_{\tau+1}^{(0)}\|^2 \cdot \|\mathbf{P}_{\tau+1}^{(\mathcal{S})} V\|^2 \leq \left( \frac{\sqrt{2v}}{\gamma} \right)^2 \implies \|\mathbf{P}_{\tau+1}^{(\mathcal{S})} V\|^2 \leq \left( \frac{\sqrt{2v}}{\gamma \|r_{\tau+1}^{(0)}\|^2} \right)^2. \tag{57}$$

Finally, since $\mathbf{P}_{\tau+1}^{\mathcal{F}} = \mathbf{I} - \mathbf{P}_\tau^{\mathcal{S}}$,

$$\|\mathbf{P}_{\tau+1}^{\mathcal{F}} V\|^2 = 1 - \|\mathbf{P}_{\tau+1}^{\mathcal{S}} V\|^2 \geq 1 - \left( \frac{\sqrt{2v}}{\gamma \|r_{\tau+1}^{(0)}\|^2} \right)^2 \tag{58}$$

$\square$

*Remark* C.5. To succeed on the new task, the eigenvalues of of the initial Gram matrix must lie overwhelmingly in the fast subspace, unless the initial residual magnitude $\|r_{\tau+1}^{(0)}\|^2$ is already tiny. This means that if the model is successful on the current task, and the new task is similar, we can expect successful training. However, if the model eigenvalues are small and so lie in the slow-subspace, it will not be able to converge to success in $K$ gradient steps.

**Lemma C.6** (Permutation invariance of the gated-linear Gram). *Consider a linearized ReLU network with weights from Equation* (16) *initialized with i.i.d Gaussian entries as NTK. Let the input dimension be $I$, width be $M$, and output dimension be $O$. Given inputs indices $n, m$, (i.e., $\boldsymbol{x}_n, \boldsymbol{x}_m \in \{\boldsymbol{x}^1, .., \boldsymbol{x}^n\}$), and output indices $o, o'$, the GLTK $\chi_\infty = \mathbb{E}_{\mathbf{W}^{[1](0)}, \mathbf{W}^{[2]}}[\mathbf{G}]$ is given by:*

$$[\chi^\infty]_{o,o'}(\boldsymbol{x}_n, \boldsymbol{x}_m) = \mathbf{E}_{\mathbf{W}^{[1](0)}, \mathbf{W}^{[2]}} \left[ \langle \boldsymbol{x}_n, \boldsymbol{x}_m \rangle \sum_{h=1}^M [\mathbf{w}_h^{[2]}]_o [\mathbf{w}_h^{[2]}]_{o'} \, g_h(\boldsymbol{x}_n) \, g_h(\boldsymbol{x}_m) \right]. \tag{59}$$

*Let $\mathbf{\Pi} \in \mathbb{R}^{I \times I}$ be a orthogonal permutation matrix [5] (i.e., $\mathbf{\Pi}^\top \mathbf{\Pi} = \mathbf{I}$). Then, the infinite width kernel $[\chi]_{(o,o')}[\infty](\boldsymbol{x}_n, \boldsymbol{x}_m)$ is invariant to the transformation $\mathbf{P}$ applied to it's inputs. Specifically for any pair of data $\boldsymbol{x}_n, \boldsymbol{x}_m$ and output indices $o, o'$:*

$$[\kappa^\infty]_{o,o'}(\mathbf{\Pi}\boldsymbol{x}_n, \mathbf{\Pi}\boldsymbol{x}_m) = [\kappa]_{o,o'}[\infty](\boldsymbol{x}_n, \boldsymbol{x}_m). \tag{60}$$

*Proof.* First, note that inner products are invariant under orthogonal transforms: $\langle \mathbf{\Pi}\boldsymbol{x}, \mathbf{\Pi}\boldsymbol{x}' \rangle = \langle \boldsymbol{x}, \boldsymbol{x}' \rangle$. Next, for each hidden unit $h$,

$$g_h(\mathbf{\Pi}\boldsymbol{x}) = \mathbf{1}\{(\mathbf{w}_h^{[1](0)})^\top \mathbf{\Pi}\boldsymbol{x} > 0\} = \mathbf{1}\{(\mathbf{\Pi}^\top \mathbf{w}_h^{[1](0)})^\top \boldsymbol{x} > 0\}. \tag{61}$$

---

[5] A matrix with exactly one entry of 1 in each row and column, but not diagonal.

Since $\mathbf{w}_h^{[1](0)}$ is initialized from an isotropic distribution and $\mathbf{\Pi}$ is orthogonal, we have $\mathbf{\Pi}^\top \mathbf{w}_h^{[1](0)} \equiv \mathbf{w}_h^{[1](0)}$. Therefore, for any fixed $\boldsymbol{x}, \boldsymbol{x}'$, the pair $(g_h(\mathbf{\Pi}\boldsymbol{x}), g_h(\mathbf{\Pi}\boldsymbol{x}'))$ has the same distribution as $(g_h(\boldsymbol{x}), g_h(\boldsymbol{x}'))$, and in particular $\mathbf{E}[g_h(\mathbf{\Pi}\boldsymbol{x})g_h(\mathbf{\Pi}\boldsymbol{x}')] = \mathbf{E}[g_h(\boldsymbol{x})g_h(\boldsymbol{x}')]$, where the expectation is over $\mathbf{w}_h^{[1](0)}$.

Finally, using the explicit kernel form (23) and independence of $V$ from $\mathbf{W}^{[1](0)}$,

$$\mathbf{E}_{\mathbf{W}^{[1](0)},\mathbf{W}^{[2]}}\Big[[\chi]_{o,o'}(\mathbf{\Pi}\boldsymbol{x}_n, \mathbf{\Pi}\boldsymbol{x}_m)\Big] = \mathbf{E}_{\mathbf{W}^{[1](0)},\mathbf{W}^{[2]}}\Big[\langle \mathbf{\Pi}\boldsymbol{x}_n, \mathbf{\Pi}\boldsymbol{x}_m\rangle \sum_{h=1}^{M}[\mathbf{w}_h^{[2]}]_o[\mathbf{w}_h^{[2]}]_{o'} \, g_h(\mathbf{\Pi}\boldsymbol{x}_n)\, g_h(\mathbf{\Pi}\boldsymbol{x}_m)\Big] \quad (62)$$

$$= \langle \boldsymbol{x}_n, \boldsymbol{x}_m\rangle \sum_{h=1}^{M} \mathbf{E}_{\mathbf{W}^{[2]}}[[\mathbf{w}_h^{[2]}]_o[\mathbf{w}_h^{[2]}]_{o'}]\, \mathbf{E}_{\mathbf{W}^{[1](0)}}[g_h(\mathbf{\Pi}\boldsymbol{x}_n)g_h(\mathbf{\Pi}\boldsymbol{x}_m)] \quad (63)$$

$$= \langle \boldsymbol{x}_n, \boldsymbol{x}_m\rangle \sum_{h=1}^{M} \mathbf{E}_{\mathbf{W}^{[2]}}[[\mathbf{w}_h^{[2]}]_o[\mathbf{w}_h^{[2]}]_{o'}]\, \mathbf{E}_{\mathbf{W}^{[2]}}[g_h(\boldsymbol{x}_n)g_h(\boldsymbol{x}_m)] \quad (64)$$

$$= \mathbf{E}_{\mathbf{W}^{[1](0)},\mathbf{W}^{[2]}}\Big[[\chi]_{(o,o')}(\boldsymbol{x}_n, \boldsymbol{x}_m)\Big], \quad (65)$$

which proves the claim. $\qquad\square$

*Remark* C.7. For permuted MNIST, each task applies some permutation $\mathbf{\Pi}_\tau$ to all inputs. Lemma C.6 implies the *expected* frozen-gates Gram (and hence its expected spectrum / $\epsilon$-rank) is task-invariant under fresh pixel permutations, up to finite-width fluctuations.

## C.3. Extension to Multi-class Cross-Entropy

For multi-class cross-entropy with logits $\boldsymbol{z} = (\boldsymbol{z}_1, \ldots, \boldsymbol{z}_n) \in \mathbb{R}^{nO}$ and blockwise softmax $\mathrm{softmax}(\boldsymbol{z})$, the empirical risk $\ell(\boldsymbol{z}) = \sum_{i=1}^{n} \ell(\boldsymbol{z}_i, \boldsymbol{y}_i)$ satisfies $\nabla_{\boldsymbol{z}}\ell(\boldsymbol{z}) = \mathrm{softmax}(\boldsymbol{z}) - \boldsymbol{Y}$.

In the NTK/linearized regime (Jacobian frozen at initialization), discrete gradient descent on parameters induces a closed-form update on training logits of the form:

$$\boldsymbol{z}^{(k+1)} = \boldsymbol{z}^{(k)} - \eta \mathbf{G}^{(0)}\nabla_{\boldsymbol{z}}\ell(\boldsymbol{z}^{(k)}), \quad (66)$$

$$\mathbf{G}^{(0)} = \mathbf{J}^{(0)}(\mathbf{J}^{(0)})^\top, \quad (67)$$

where $\mathbf{J}_0$ is the initialization-time feature map induced by the training inputs and $\mathbf{G}^{(0)} \in \mathbb{R}^{(nO)\times(nO)}$ is the corresponding multi-output NTK/Gram matrix with blocks $\mathbf{G}^{(0)}_{(i,o),(j,o')} = \langle \nabla_{\boldsymbol{\theta}}F_o(\boldsymbol{x}_i), \nabla_{\boldsymbol{\theta}}F_{o'}(\boldsymbol{x}_j)\rangle$. This is exactly the discrete-time counterpart of the standard NTK function-space dynamics for general losses (Lee et al., 2019). To recover the same fast/slow eigenmode story as in squared loss, we linearize the cross-entropy gradient in logit space around a reference $\boldsymbol{z}^\star$:

$$\nabla_{\boldsymbol{z}}\ell(\boldsymbol{z}^\star + \delta\boldsymbol{z}) = \nabla_{\boldsymbol{z}}\ell(\boldsymbol{z}^\star) + \mathbf{H}^\star \delta\boldsymbol{z} + \text{h.o.t.}, \quad (68)$$

where $\mathbf{C} = \nabla_{\boldsymbol{z}}^2\ell(\boldsymbol{z}^\star) = \mathrm{blockdiag}(\mathbf{C}_1, \ldots, \mathbf{C}_n)$ and $\mathbf{C}_i = \mathrm{diag}(\mathbf{p}_i) - \mathbf{p}_i\mathbf{p}_i^\top \succeq 0$ with $\mathbf{p}_i = \mathrm{softmax}(\boldsymbol{z}_\star^i)$ (Wu et al., 2022). Then,

$$\boldsymbol{z}^{(k+1)} = \boldsymbol{z}^{(*)} + \delta\boldsymbol{z}^{(k)} - \eta\mathbf{G}^{(0)}\nabla_{\boldsymbol{z}}\ell(\boldsymbol{z}^{(*)} + \delta\boldsymbol{z}^{(k)}) \quad (69)$$

$$\approx \boldsymbol{z}^{(*)} + \delta\boldsymbol{z}^{(k)} - \eta\mathbf{G}^{(0)}\nabla_{\boldsymbol{z}}\ell(\boldsymbol{z}^{(*)} + \mathbf{C}^*\delta\boldsymbol{z}^{(k)}), \quad (70)$$

which implies

$$\delta\boldsymbol{z}^{(k+1)} \approx (\boldsymbol{I} - \eta\mathbf{G}^{(0)}\mathbf{C})\delta\boldsymbol{z}^{(k)} - \eta\mathbf{G}^{(0)}\nabla_{\boldsymbol{z}}\ell(\boldsymbol{z}^*). \quad (71)$$

If $\boldsymbol{z}^*$ is a stationary point, so that $\nabla_{\boldsymbol{z}}\ell(\boldsymbol{z}^*) = 0$, the additive term vanishes; otherwise, the local linearization is affine with respect to the linear same linear part.

Define the weighted error $\mathbf{e}^{(k)} \doteq \mathbf{C}^{1/2}\delta\boldsymbol{z}^{(k)}$, so that the local cross-entropy quadratic satisfies $\delta\boldsymbol{z}^\top\mathbf{C}\delta\boldsymbol{z} = \|\mathbf{e}\|_2^2$; in these coordinates the linearized update becomes

$$\mathbf{e}^{(k+1)} \approx (\boldsymbol{I} - \eta\mathbf{C}^{1/2}\mathbf{G}^{(0)}\mathbf{C}^{1/2})\mathbf{e}^{(k)}. \quad (72)$$

Hence, the relevant "fast eigenvalues" for cross-entropy are those of the GGN $\mathbf{G}^{\star}_{\mathrm{LW}} \doteq \mathbf{C}^{1/2}\mathbf{G}^{(0)}\mathbf{C}^{1/2}$, not the raw $\mathbf{G}^{(0)}$. By Lemma C.6 the infinite width limit of $\mathbf{G}^{(0)}$ remains task-invariant.

This matches the generalized Gauss–Newton viewpoint: in the linearized model, the parameter-space curvature is $\mathbf{J}^{\top(0)}\mathbf{C}\mathbf{J}^{(0)}$, whose nonzero spectrum agrees with $\mathbf{C}^{1/2}\mathbf{J}^{(0)}(\mathbf{J}^{(0)})^{\top}\mathbf{C}^{1/2}$ (Martens & Grosse, 2015).

## C.4. Empirical Results

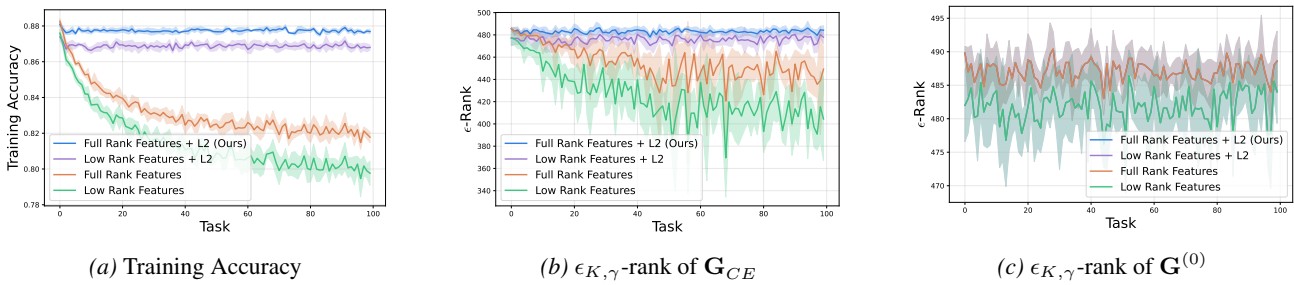

*(a)* Training Accuracy      *(b)* $\epsilon_{K,\gamma}$-rank of $\mathbf{G}_{CE}$      *(c)* $\epsilon_{K,\gamma}$-rank of $\mathbf{G}^{(0)}$

*Figure 5.* Metics tracked in our linearized ReLU network experiment. As expected, the training accuracy correlates with the $\epsilon_{K,\gamma}$-rank of the $\mathbf{G}_{CE}$, while the $\epsilon_{K,\gamma}$-rank of $\mathbf{G}^{(0)}$ remains fairly flat on average over training.

## C.5. Proofs For Section 6

**Theorem C.8.** *Suppose an $L$-layered MLP $F_{\boldsymbol{\theta}} : \mathbb{R}^I \to \mathbb{R}^O$, and all input data lies in $\mathcal{B}_r(0)$, and either means-squared error or softmax-cross-entropy losses $\ell$. Let the network be $\boldsymbol{x} \mapsto F_{\boldsymbol{\theta}}(\boldsymbol{x})$ be $L_f$-Lipschitz on $\mathcal{B}_r(0)$ and let the logit-gradient, $\boldsymbol{z} \mapsto \nabla_{\boldsymbol{z}} \ell(\boldsymbol{z}, \boldsymbol{y})$ be $L_g$-Lipschitz. Define: $\Gamma(\boldsymbol{\theta}) := \sup_{\boldsymbol{x} \in \mathcal{B}_r(0), 1 \leq i \leq O} \| \nabla_{\boldsymbol{\theta}}^2 \boldsymbol{z}_i(\boldsymbol{x}; \boldsymbol{\theta}) \|_2$ and $\boldsymbol{z} = F_{\boldsymbol{\theta}}(\boldsymbol{x})$ Then, the residual $\mathbf{R}$ of the Hessian on task $\tau + 1$ obeys the following operator-norm bound:*

$$\| \mathbf{R}(\boldsymbol{\theta}_{\tau+1}^0) \|_2 \leq \sqrt{O} \Gamma(\boldsymbol{\theta}_\tau) \left( \sup_{\boldsymbol{x} \in \mathrm{supp}(\mathcal{X}_\tau)} [\nabla_{\boldsymbol{z}} \ell(\boldsymbol{z}_i, \boldsymbol{y})] + L_f L_g d(\mathcal{X}, \mathcal{X}') \right) \doteq \boldsymbol{B}_\tau. \tag{73}$$

*Proof.* We start with the definition of the residual term of the Hessian:

$$\mathbf{R}(\boldsymbol{\theta}_{\tau+1}^0) = \mathop{\mathbb{E}}_{\boldsymbol{x} \sim \mathcal{X}_{\tau+1}} \left[ \sum_i^O \nabla_{\boldsymbol{z}} \ell(\boldsymbol{z}_i, \boldsymbol{y}) \nabla_{\boldsymbol{\theta}}^2 \boldsymbol{z}_i(\boldsymbol{x}; \boldsymbol{\theta}_\tau) \right] . \tag{74}$$

Next we apply linearity and operator norms:

$$\| \mathbf{R}(\boldsymbol{\theta}_{\tau+1}^0) \|_2 = \left\| \mathop{\mathbb{E}}_{\boldsymbol{x} \sim \mathcal{X}_{\tau+1}} \left[ \sum_i^O \nabla_{\boldsymbol{z}} \ell(\boldsymbol{z}_i, \boldsymbol{y}) \nabla_{\boldsymbol{\theta}}^2 \boldsymbol{z}_i(\boldsymbol{x}; \boldsymbol{\theta}_\tau) \right] \right\| \tag{75}$$

$$\leq \mathop{\mathbb{E}}_{\boldsymbol{x} \sim \mathcal{X}_{\tau+1}} \sum_i^O \| \nabla_{\boldsymbol{z}} \ell(\boldsymbol{z}_i, \boldsymbol{y}) \|_2 \| \nabla_{\boldsymbol{\theta}}^2 \boldsymbol{z}_i(\boldsymbol{x}; \boldsymbol{\theta}_\tau) \|_2 \tag{76}$$

and apply a deterministic worst-case bound followed followed by the Cauchy-Schwartz inequality to obtain:

$$\| \mathbf{R}(\boldsymbol{\theta}_{\tau+1}^0) \|_2 \leq \sup_{\boldsymbol{x} \in \mathrm{supp}(\mathcal{X}')} \| \nabla_{\boldsymbol{z}} \ell(\boldsymbol{z}_i, \boldsymbol{y}) \|_2 \sqrt{O} \Gamma(\boldsymbol{\theta}) \tag{77}$$

Let $\pi$ be a coupling measure on $\mathcal{X}$ and $\mathcal{X}'$, and denote $h(\boldsymbol{x}; \boldsymbol{\theta}) = \nabla_{\boldsymbol{z}} \ell(F_{\boldsymbol{\theta}}(\boldsymbol{x}), \boldsymbol{y})$. First operating inside the supp, we apply the triangle inequality:

$$\| \nabla_{\boldsymbol{z}} h(\boldsymbol{x}; \boldsymbol{\theta}) \|_2 \leq \| h(\boldsymbol{x}'; \boldsymbol{\theta}) \|_2 + \| h(\boldsymbol{x}; \boldsymbol{\theta}) - h(\boldsymbol{x}'; \boldsymbol{\theta}) \|_2 \tag{78}$$

$$\leq \| h(\boldsymbol{x}'; \boldsymbol{\theta}) \|_2 + L_f L_g d(\mathcal{X}, \mathcal{X}') \quad \text{Applying Lipschitz assumptions.} \tag{79}$$

Now, operating inside the supp w.r.t. $\pi$ we obtain:

$$\sup_{\boldsymbol{x} \in \mathrm{supp}(\mathcal{X})} \| h(\boldsymbol{x}; \boldsymbol{\theta}) \|_2 = \sup_{(\boldsymbol{x}, \boldsymbol{x}') \in \mathrm{supp}(\pi)} \| h(\boldsymbol{x}'; \boldsymbol{\theta}) \|_2 \leq \sup_{\boldsymbol{x}' \in \mathrm{supp}(\mathcal{X}')} \| h(\boldsymbol{x}'; \boldsymbol{\theta}) \|_2 + L_f L_g d(\mathcal{X}, \mathcal{X}') \tag{80}$$

Finally, bounding the RHS of the inequality from above by an arbitrary constant $B$ yields the desired result. $\square$

**Corollary C.9.** *We can use Weyl's inequality to show the existence of an appropriate L2 regularizer to guarantee that increasing $\mathrm{rank}(\mathbf{H}_{GGN})$ implies increasing the $\mathrm{rank}(\mathbf{H})$. Suppose ordered eigenvalues of $\mathbf{H}_{GGN}$ (i.e., $\lambda_1(\cdot) \geq ... \lambda_n(\cdot)$), then by applying Wely's inequality we obtain*

$$|\lambda_i(\mathbf{H}_{GGN} + \mathbf{R} -) \lambda_i(\mathbf{R})| \leq \| \mathbf{R} \|_2 \tag{81}$$

$$\lambda_i(\mathbf{H}_{GGN} + \mathbf{R}) \geq \lambda_i(\mathbf{H}_{GGN}) - \| \mathbf{R} \|_2 \tag{82}$$

$$\lambda_i(\mathbf{H}) \geq \lambda_i(\mathbf{H}_{GGN}) - \| \mathbf{R} \|_2 \tag{83}$$

$$\lambda_{\min}^+(\mathbf{H}) \geq \lambda_{\min}^+(\mathbf{H}_{GGN}) - \| \mathbf{R} \|_2. \tag{84}$$

*The L2-regularized Hessian becomes:*

$$\lambda_{\min}^+(\mathbf{H} + 2\boldsymbol{I}q) \geq \lambda_{\min}^+(\mathbf{H}_{GGN}) - \| \mathbf{R} \|_2 + 2q \tag{85}$$

$$\geq \lambda_{\min}^+(\mathbf{H}_{GGN}) - \| \boldsymbol{B} \|_2 + 2q, \tag{86}$$

*so if $2q > \| \mathbf{R} \|_2 - \lambda_{\min}^+(\mathbf{H}_{GGN})$, then every positive eigenvalue of $\mathbf{H}_{GGN}$ remains positive in the q-regularized $\mathbf{H}$.*

*Remark* C.10. Theorem C.8 and Corollary C.9 justify why $L2$ is needed alongside $ER$ regularization. Under softmax-cross-entropy loss and mean-squared-error loss with bounded targets, $h(\boldsymbol{x}; \boldsymbol{\theta})$ is also bounded which ensures the theorem is well-posed. The theorem also requires $\Gamma(\boldsymbol{\theta})$ to be uniformly bounded which can be achieved by techniques like spectral regularization (although we do use it in our experiments) (Yoshida & Miyato, 2017). In practice, we find that small choices of $L2$ regularization are needed which we report in table 2, table 4, and table 6.

## D. KFAC Error

For a given layer, $\mathbf{W}^{[l]} \in \mathbb{R}^{d_{in} \times d_{out}}$, with activations into the layer $\mathbf{a}^{[l]} \in \mathbb{R}^{d_{in}}$ and gradient outputs $\boldsymbol{\delta}^{[l]} \in \mathbb{R}^{d_{out}}$. The respective covariances are $\mathbf{a}^{[l]}\mathbf{a}^{[l]\top} \in \mathbb{R}^{d_{in} \times d_{in}}$ and $\boldsymbol{\delta}^{[l]}\boldsymbol{\delta}^{[l]\top} \in \mathbb{R}^{d_{out} \times d_{out}}$. The resulting KFAC block is of size $(d_{out}d_{in}) \times (d_{out}d_{in})$. The per-entry error, on the $n$th sample, can be obtained by indexing over Kronecker matrix pairs with row-index $(i, \alpha)$ and column index $(j, \beta)$. We use $\cdot_{(i,*)}^{[l]}$ to denote the $i$th row of $(\cdot)$ in layer $l$, and $\cdot_{(*,j)}^{[l]}$ for the $j$th column. Flattening over the Kronecker product, we obtain the following for a given entry in layer $l$:

$$(\mathbf{H}_{GGN})^{[l]}_{([i,\alpha],[j,\beta])} = \frac{1}{N} \sum_{n=1}^{N} \sum_{o=1}^{O} [\mathbf{a}_n^{[l]}]_{(i,*)}[\mathbf{a}_n^{[l]}]_{(*,j)}[\boldsymbol{\delta}_{n,o}^{[l]}]_{(\alpha,*)}[\boldsymbol{\delta}_{n,o}^{[l]}]_{(*,\beta)} = \overline{[\mathbf{a}^{[l]}]_{(i,*)}[\mathbf{a}^{[l]}]_{(*,j)}[\boldsymbol{\delta}^{[l]}]_{(\alpha,*)}[\boldsymbol{\delta}^{[l]}]_{(*,\beta)}} \tag{87}$$

$$(\widehat{\mathbf{H}}_{GGN})^{[l]}_{([i,\alpha],[j,\beta])} = \left( \frac{1}{N} \sum_{m=1}^{N} [\mathbf{a}_n^{[l]}]_{(i,*)}[\mathbf{a}_n^{[l]}]_{(*,j)} \right) \cdot \left( \frac{1}{N} \sum_{n=1}^{N} \sum_{o=1}^{O} [\boldsymbol{\delta}_{m,o}^{[l]}]_{(\alpha,*)}[\boldsymbol{\delta}_{m,o}^{[l]}]_{(*,\beta)} \right) = \overline{[\mathbf{a}^{[l]}]_{(i,*)}[\mathbf{a}^{[l]}]_{(*,j)}} \cdot \overline{[\boldsymbol{\delta}^{[l]}]_{(\alpha,*)}[\boldsymbol{\delta}^{[l]}]_{(*,\beta)}}, \tag{88}$$

where we use $\overline{\cdot}$ to denote the empirical sample average. In this way, we can compute the error as the difference between the average of products and the product of averages; for a fixed index quadruple $(i, j, \alpha, \beta)$, the KFAC error of the $((i, \alpha), (j, \beta))$th-entry is:

$$\Delta_{[i,\alpha],[j,\beta]} = (\mathbf{H}_{GGN} - \widehat{\mathbf{H}}_{GGN})_{[i,\alpha],[j,\beta]} = \overline{[\mathbf{a}^{[l]}]_i[\mathbf{a}^{[l]}]_j[\boldsymbol{\delta}^{[l]}]_\alpha[\boldsymbol{\delta}^{[l]}]_\beta} - \overline{[\mathbf{a}^{[l]}]_i[\mathbf{a}^{[l]}]_j} \cdot \overline{[\boldsymbol{\delta}^{[l]}]_\alpha[\boldsymbol{\delta}^{[l]}]_\beta} \tag{89}$$

# E. Algorithm

## E.1. Baselines

**Backpropagation (BP)**: We optimize the objective with stochastic gradient descent (SGD) on cross-entropy loss over the current task. Let $\theta$ denote all trainable parameters and $\mathcal{D}_t$ the data for task $\tau$. The objective is

$$\min_{\theta} \; \mathbb{E}_{(x,y)\sim\mathcal{D}_t}[L(F_\theta(x), y)].$$

**Vanilla Effective Rank (ER)**: In addition to BP, we collect the output features of each dense layer (excluding convolutional layers) over a fixed number of steps (er-batch in Table 1). From these stacked features, we compute the effective rank (Roy & Vetterli, 2007) for each layer and sum across all layers. Then we maximize the effective rank of the network representations.

$$\mathcal{L}_{\text{ER}} = -\frac{1}{L}\sum_{\ell=1}^{L}\text{ER}(\mathbf{h}^{[l]}), \qquad \text{ER}(\mathbf{h}^{[l]}) = \exp\Big(-\sum_{i=1}^{d_\ell} p_i^{[\ell]} \log p_i^{[\ell]}\Big),$$

where $\mathbf{h}^{[l]} \in \mathbb{R}^{N \times d_\ell}$ is the stacked feature matrix for layer $\ell$, $s_i(\mathbf{h}^{[l]})$ are its singular values, and

$$p_i^{[\ell]} = \frac{s_i(\mathbf{h}^{[l]})}{\sum_j s_j(\mathbf{h}^{[l]})}.$$

**L2 normalization (L2)**: We add weight decay to BP. Now the objective becomes:

$$\min_{\theta} \; \mathbb{E}_{(x,y)\sim\mathcal{D}_t}[L(F_\theta(x), y)] \; + \; \lambda_w\|\theta\|_2^2,$$

where $\lambda_w$ is the weight-decay coefficient (weight-decay in Table 1).
**Continual Backpropagation (CBP)**: We follow the architecture in (Dohare et al., 2024).
**Layer-normalization (LayerNorm-L2)**: We follow the architecture in (Lyle et al., 2024).
**Spectral**: We follow the architecture in (Lewandowski et al., 2024a).

## E.2. Effective Rank with $L2$ normalization ($L2$-ER)

The pseudocode for our implementation of $L2$-ER is shown in Algorithm 1. $L2$-ER augments the standard task objective with two additional regularizers: (1) an effective rank penalty and (2) an $L2$ weight decay term. At each step, the task loss $L_{\text{task}}$ together with the weight decay term is minimized via standard backpropagation. Simultaneously, the hidden features $\mathbf{h}^{[l]}$ are collected over a fixed window L of updates. Every L steps, these stacked features are used to compute the effective rank (Roy & Vetterli, 2007) of the representation at each layer. The effective rank losses are averaged across layers. Note that a gradient descent step is taken on this loss only every L steps.

---

**Algorithm 1** Continual Learning with EffectiveRank Regularization ($L$2-ER)

---

**Input:** Task datasets $\{\{(\boldsymbol{x}_{\tau,i}, \boldsymbol{y}_{\tau,i})\}_{i=0}^{N-1}\}_{\tau=0}^{T-1}$; model $F_{\boldsymbol{\theta}}$; learning rates $\alpha$ (task), $\beta$ (ER); weight decay $\lambda$; ER update interval $U$ (in steps)

**State:** Per-layer feature buffers $\{\mathcal{B}^{\ell}\}_{\ell=1}^{L}$ with capacity $U$

**for** $\tau = 0$ **to** $T - 1$ **do**

    **for** $i = 0, \ldots, N - 1$ **do**

        $(\hat{\boldsymbol{y}}, \{\mathbf{h}^{[l]}\}_{\ell=1}^{L}) \leftarrow F_{\boldsymbol{\theta}}(\boldsymbol{x}_{\tau,i})$ {$\hat{\boldsymbol{y}}$ is the model prediction}

        $\forall \ell: \ \mathcal{B}^{\ell} \leftarrow \mathrm{enqueue}(\mathcal{B}^{\ell}, \mathbf{h}^{[l]});$ if $|\mathcal{B}^{\ell}| > U$ then drop oldest

        $L_{\mathrm{task}} \leftarrow \mathrm{Loss}(\hat{\boldsymbol{y}}, \boldsymbol{y}_{\tau,i}) + \lambda \|\boldsymbol{\theta}\|_2^2$

        $\boldsymbol{\theta} \leftarrow \boldsymbol{\theta} - \alpha \nabla_{\boldsymbol{\theta}} L_{\mathrm{task}}$

        **if** $(i + 1) \bmod U = 0$ **then**

            $L_{\mathrm{erank}} \leftarrow -\frac{1}{L} \sum_{\ell=1}^{L} \mathrm{ER}(\mathrm{SVD}(\mathrm{Stack}(\mathcal{B}^{\ell})))$ {(Roy & Vetterli, 2007)}

            $\boldsymbol{\theta} \leftarrow \boldsymbol{\theta} - \beta \nabla_{\boldsymbol{\theta}} L_{\mathrm{erank}}$

            $\mathcal{B}^{\ell} \leftarrow \emptyset$

        **end if**

    **end for**

**end for**

**Return:** $\boldsymbol{\theta}$

---

## F. Dead Neurons and Effective Rank

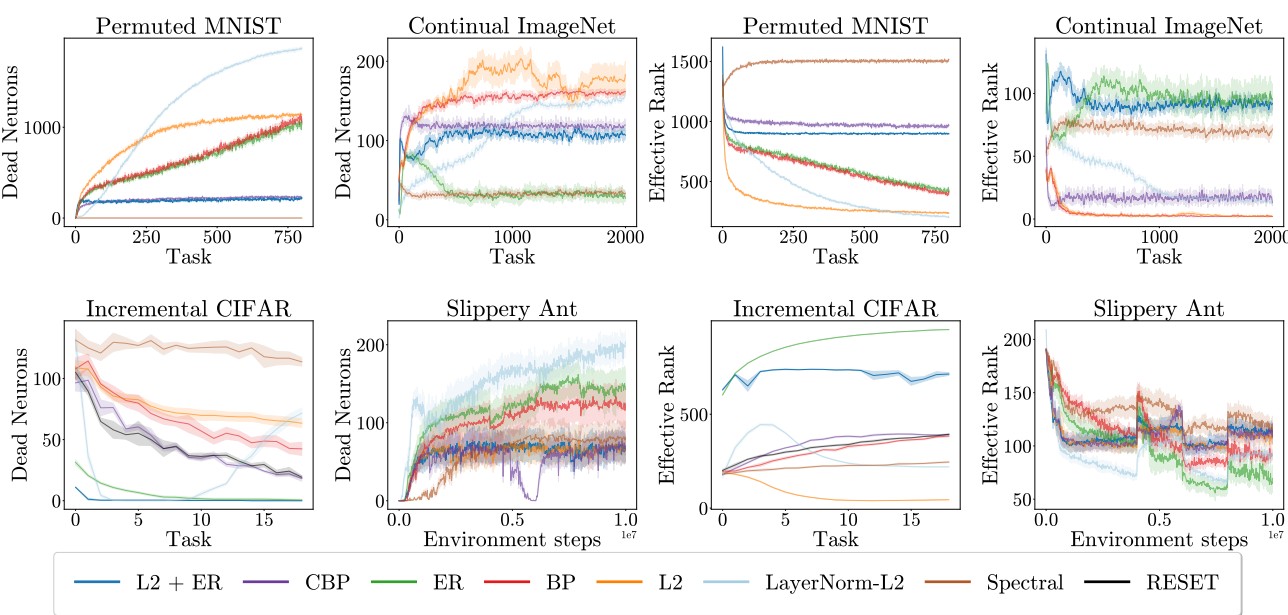

*Figure 6.* Number of dead neurons (left) and effective rank (right) corresponding to Figure 4.

Here, we present number of dead neurons and effective rank corresponding to Figure 4. An important observation to note here is that dead neurons in Incremental Cifar is decreasing rather than increasing. This effect arises due to the following two reasons: First, at the beginning of training, the environment contains only 5 classes, which gradually increase to 100. As more classes are introduced, the evaluation set becomes much larger, increasing the likelihood of encountering samples that activate a given neuron (i.e., produce nonzero outputs). Second, to accommodate the changing number of classes, we mask the outputs of the final layer to match the number of active classes in each task. In the early tasks, this masking leads to a sharp rise in the number of dead neurons, since the network tends to overfit to the small set of available classes. As more classes are added, this effect diminishes. Therefore, we also measured the number of dead neurons and effective rank in RESET for comparison. Learning curves that lie above RESET indicate an increase in dead neurons due to loss of plasticity, whereas curves below suggest relatively fewer dead neurons.

## G. Epsilon Hessian Rank

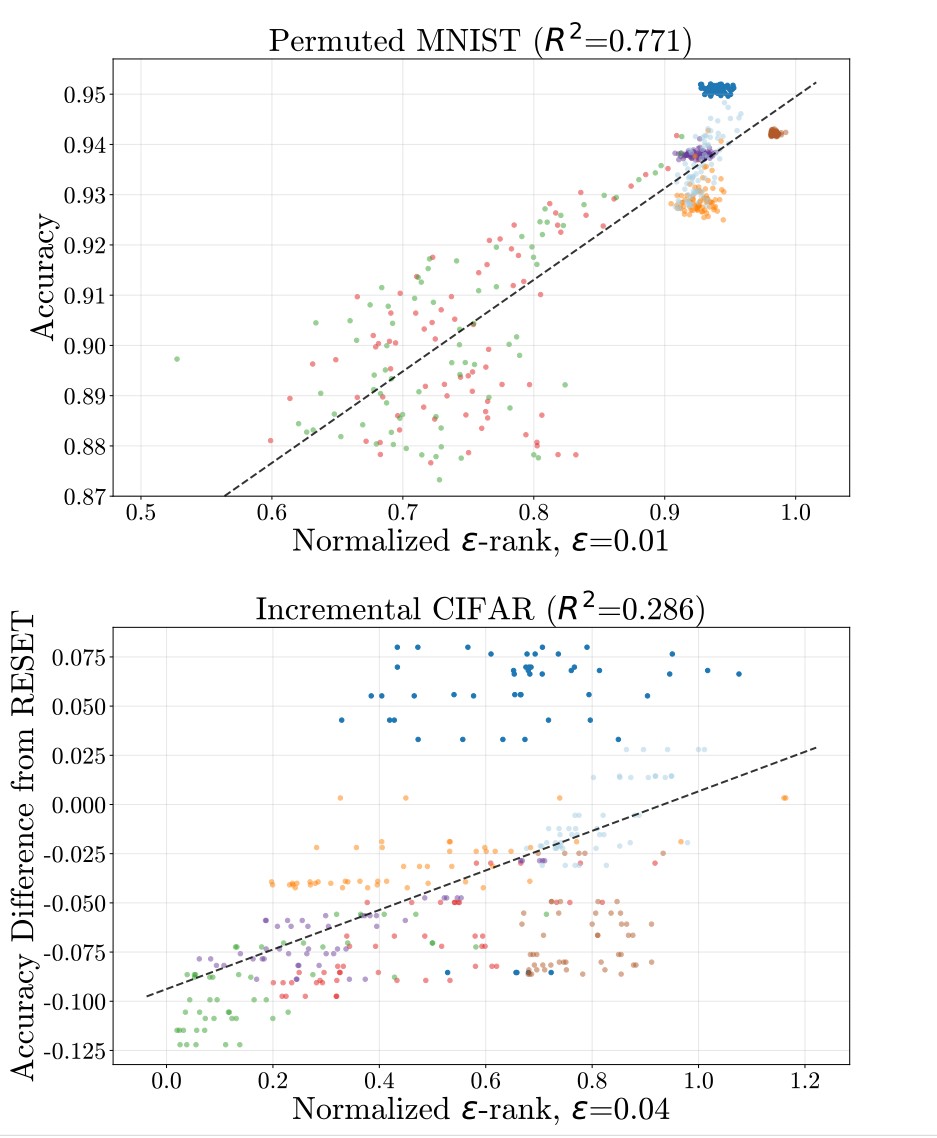

*Figure 7.* Curvature vs. Accuracy on Permuted MNIST (Left) and First 10 Task Incremental CIFAR (Right). A linear fit (dotted line) highlights the positive association between curvature and accuracy.

### G.1. Measuring the Hessian Spectral Density

To empirically demonstrate the relationship between dead ReLUs and the rank of the Hessian, we estimate the Hessian spectrum over continual learning tasks (Ghorbani et al., 2019). The spectral density is defined as $\psi(u) = \frac{1}{P} \sum_{i=1}^{P} \delta(u - \lambda_i)$ where $\delta$ is a Dirac delta operator. Since the naive approach to estimating the density would involve calculating every eigenvalue, we turn to a Gaussian relaxation (Ghorbani et al., 2019):

$$\psi_\sigma(u) = \frac{1}{P} \sum_{i=1}^{P} f(\lambda_i; u, \sigma^2)$$

where $f(\lambda_i; u, \sigma^2) = \frac{1}{\sigma\sqrt{2\pi}} \exp(-\frac{(u-\lambda)^2}{2\sigma^2})$ and $\sigma^2$ is the variance. When $\sigma^2$ is small, $\psi_\sigma$ provides a tight estimate of the spectral density. Since materializing the full Hessian is prohibitively expensive, we estimate the density with the

stochastic Lanczos quadrature algorithm (Golub & Welsch, 1969; Ghorbani et al., 2019), which takes advantage of the fact that accessing Hessian-vector-products (HVP) is a computationally efficient operation (Pearlmutter, 1994). Given $\mathbf{H}$ is diagonalized and $f$ has a closed-form equation, we can define $f(\mathbf{H}) = \mathbf{Q}f(\mathbf{\Lambda})\mathbf{Q}^\top$ where $f(\mathbf{\Lambda})$ acts on the diagonal. Now we estimate the spectrum of Hessian through the HVP with $\mathbf{v} \sim N(0, \frac{1}{P}\boldsymbol{I}_{P \times P})$ which gives:

$$\psi_\sigma = \frac{1}{P}\text{tr}\bigg(f(\mathbf{H}, u, \sigma^2)\bigg) = \mathbb{E}[\mathbf{v}^\top f(\mathbf{H}, u, \sigma^2)\mathbf{v}]$$

In practice, each $\psi_\sigma^{\mathbf{v}}$ is approximated by $m$-steps of the Lanczos algorithm resulting in a $m \times m$ tridiagonal matrix with $m$ locations-weight pairs $(\ell_j, \omega_j)$ so that:

$$\psi_\sigma^{\mathbf{v}}(u) \approx \sum_{j=1}^{m} \omega_j f(\ell_j; u, \sigma^2) \doteq \hat{\psi}^{\mathbf{v}}(u)$$

Moreover, $\psi_\sigma^{\mathbf{v}}(\cdot)$ is an unbiased estimator of $\psi_\sigma(\cdot)$ and $\hat{\psi}^{\mathbf{v}}(u)$ converges exponentially fast around its expectation over samples of $\mathbf{v}$ (Ghorbani et al., 2019)[Claim 2.3] resulting in a computationally efficient and accurate estimation of the spectral density, even for large neural networks.

Throughout our experiments, we measure the Hessian eigen-spectrum at the beginning and at the end of each new task. Our results show that the spectrum of standard back-propagation narrows as the task number grows. In fact, a complete loss of learning corresponds to a complete collapse of the spectrum. Furthermore, we show that loss of plasticity mitigation like continual backpropagation and our own $L2$-ER method preserve the spectrum.

In the following sections, we present the details of each environment, their corresponding hyperparameter sweeps and selected best hyperparameters, followed by the analysis of the Hessian spectrum.

### G.2. Epsilon Hessian Rank vs Accuracy

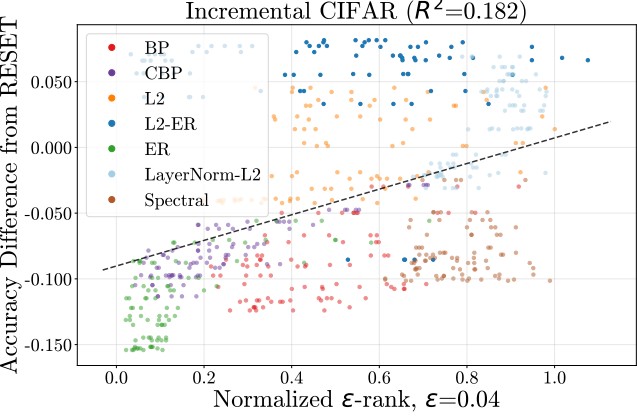

*Figure 8.* Curvature vs. Accuracy on Full 20 Task Incremental CIFAR. A linear fit (dotted line) highlights the positive association between curvature and accuracy.

In addition to Figure 3, we provide the epsilon rank of the other two supervised learning environments in Figure 7. Continual ImageNet and Permuted MNIST's data points on the graph are calculated by an average across 5 seeds.

Putting all the Incremental Cifar tasks into one plot does not yield as clear a positive relationship as Permuted MNIST or Continual Imagenet. All the tasks in Permuted MNIST or Continual Imagenet are about the same level of difficulties for $\tau \in \{1, \cdots, T\}$, which gives us the ability to easily measure successful training through task accuracies. In Incremental Cifar, each task $\tau$ is composed of classifying $5 * \tau$ classes of images using the same computational budget, which makes it a challenge to isolate unsuccessful training due to loss of plasticity from increasing task difficulties. We report the performance difference between algorithms and a freshly initialized network as a measure of successful training. Furthermore,

neural network tends to find a lower-rank solution regardless of initialization (Hu et al., 2022). Since we always report the $\epsilon$-Hessian rank of the Full databatch, the percentage of data the neural network has already been trained on becomes larger and larger as $\tau$ grows. For example, at task 2 initialization, we trained on 5 classes and acquired a low rank representation, then 5 classes are added and the $\epsilon$-Hessian rank is high; while at task 20 initialization, we trained on 19 classes and are evaluating on 20, this is basically the same as the eigenspectrum at task 19 convergence, which is a low rank solution. In short, due to the non-uniform property of the tasks, $\epsilon$-Hessian rank is not an accurate measure of Spectral Collapse in Incremental Cifar. When we group the first ten tasks (fig. 7), we can see that the positive relationship is clearer since the task difficulties are more similar.

# H. Permuted MNIST

We now detail environments and architectural details in all experiments. All algorithms are written in JAX (Bradbury et al., 2018). Although each environment is independent, our training procedure is designed to be easily generalizable. In the following sections, we provide detailed descriptions of each environment.

Permuted MNIST (Dohare et al., 2024) is a variant of the original MNIST dataset (LeCun, 1998) where the pixels are permuted randomly in the same way for each image in the original dataset. Each permutation defines a new task for the learner. In total, we evaluate on 800 tasks, each of which is a 10-class classification problem.

## H.1. Network Architecture

We employ a standard multilayer perceptron (MLP) with three hidden layers of hidden size 1000 each followed by a ReLU activation. All weights are initialized with Kaiming uniform. The architecture can be summarized as follows:

```
MLP(
  Sequential(
    (0): Dense(init=kaiming_uniform, out_dims=1000, bias=True)
    (1): ReLU()
    (2): Dense(init=kaiming_uniform, out_dims=1000, bias=True)
    (3): ReLU()
    (4): Dense(init=kaiming_uniform, out_dims=1000, bias=True)
    (5): ReLU()
    (6): Dense(init=kaiming_uniform, out_dims=10, bias=True)
  )
)
```

## H.2. Hyperparameters

In Table 1, we present the default hyperparam of our experiments. Unless otherwise specified, experiments use the default hyperparameters.

## H.3. Experiments

Prior work (Dohare et al., 2024), along with our own experiments, shows that the learning rate is a critical factor in continual learning: using a smaller learning rate consistently yields only marginal differences in performance. To better highlight the phenomenon of loss of plasticity, we fix the learning rate to $1 \times 10^{-2}$ and sweep over the remaining hyperparameters in Table 2. We report the results in Figure 6 and the selected best hyperparameters in Table 2.

## H.4. Permuted MNIST Hessian Spectrum

We use the best hyperparameters in Table 2 and rerun it with 5 seeds to plot the hessian spectrum during our training. We calculate the approximated hessian matrix every fixed number of tasks (compute-hessian-interval in Table 1). The corresponding performance is shown in Figure 4 and Hessian spectrum of seed 2025 is shown in Figure 9, where the orange curve corresponds to the Hessian spectrum on the test set and the blue curve corresponds to the training set. Since plotting all tasks is impractical, we present only a subset of representative tasks for BP, L2, ER, CBP, L2-ER.

Comparing Figure 4 with Figure 9, we observe that algorithms which eventually lose plasticity exhibit a sparse Hessian spectrum. In contrast, algorithms that maintain plasticity preserve a rich and dense spectrum, proving our claim that spectral collapse is strongly correlated with the loss of plasticity. Furthermore, even subtle reductions in plasticity are reflected in the spectrum: for instance, $L2$ shows a slight decline of about 1% in accuracy over training. This small change is captured by the eigenspectrum, as shown in Figure 9, where the range of eigenvalues for $L2$ is narrower than that of the other two algorithms that preserve plasticity.

*Table 1.* permuted MNIST default hyperparameters

| Hyperparam Name | Default | Description |
|---|---|---|
| study_name | test | experiment name |
| seed | 2024 | base random seed |
| debug | False | true to enable debug mode |
| platform | gpu | {cpu, gpu} |
| n_seeds | 1 | number of seeds |
| env | permuted_mnist | environment name |
| agent | l2_er | {er, bp, l2, cbp, l2_er, layernorm_l2, spectral} |
| activation | relu | activation options: {relu, tanh} |
| lr | [0.01] | learning rate(s) |
| optimizer | sgd | {adam, sgd} |
| weight_decay | 0.001 | $L2$ weight decay |
| num_features | 1000 | hidden size for the mlp |
| change_after | $10 \times 6000$ | steps between task switches |
| to_perturb | False | whether to perturb the input data |
| perturb_scale | $1 \times 10^{-5}$ | magnitude of input perturbation |
| num_hidden_layers | 3 | number of hidden layers in the mlp |
| mini_batch_size | 1 | minibatch size |
| no_anneal_lr | True | if true, do not anneal the learning rate |
| max_grad_norm | 0.5 | gradient clipping threshold |
| num_tasks | 800 | number of tasks used in training/eval |
| **effective rank** | | |
| er_lr | [0.01] | ER learning rate |
| er_batch | 100 | batch size for er computation |
| er_step | 1 | ER update frequency |
| **evaluation** | | |
| evaluate | True | evaluate after each task |
| evaluate_previous | False | evaluate on previous task |
| eval_size | 2000 | number of evaluate samples per task |
| compute_hessian | False | whether to compute hessian spectrum |
| compute_hessian_size | 2000 | samples used for hessian computation |
| compute_hessian_interval | 1 | interval in tasks between hessian runs |
| **continual backpropagation** | | |
| cont_backprop | False | enable CBP |
| replacement_rate | $1 \times 10^{-6}$ | CBP replacement probability per step |
| decay_rate | 0.99 | exponential decay for CBP statistics |
| maturity_threshold | 100 | steps before a unit is considered mature |
| **Spectral Regularizer** | | |
| k | 2 | power used in $(\sigma_{\max}^k - \texttt{target})^2$ |
| target | 2.0 | target value for the largest singular value |
| spectral_strength | 0.1 | coefficient multiplying the spectral penalty |
| num_iter | 10 | number of power-iteration steps to estimate $\sigma_{\max}$ |

**Algorithms that *lose plasticity***

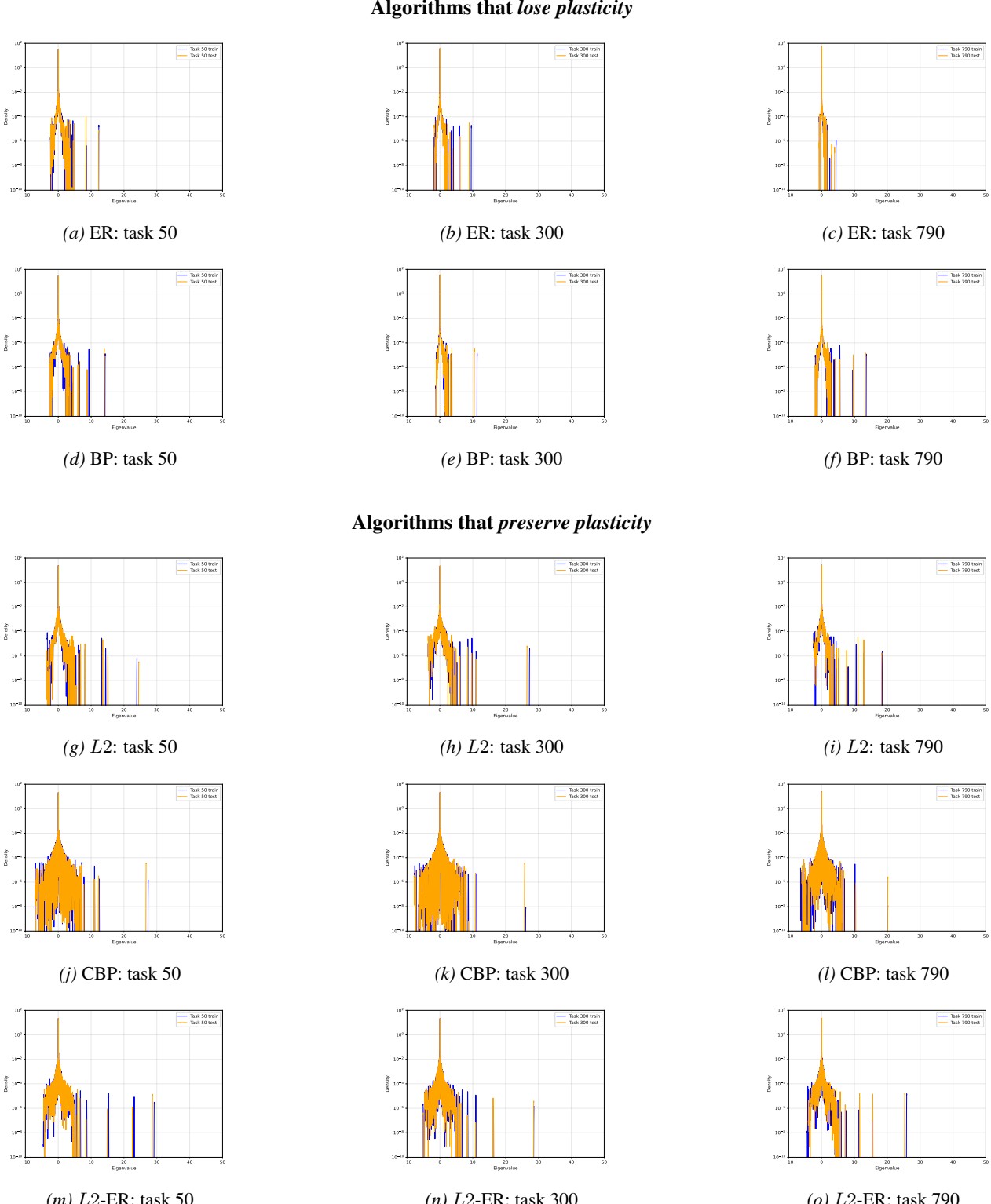

*Figure 9.* Comparison of Hessian eigenspectra at **task init** across permuted MNIST. From top to bottom, the algorithms are ordered by increasing accuracy. **Top:** Algorithms that lose plasticity (ER, BP). **Bottom:** Algorithms that preserve plasticity ($L2$, CBP, $L2$-ER).

*Table 2.* Hyperparameter sweeps and best values for Permuted MNIST across all algorithms under a fixed learning rate of $1 \times 10^{-2}$.

| Algorithm | Hyperparameter | Sweep Hyperparameters | Best |
|---|---|---|---|
| BP | Learning rate | $\{1 \times 10^{-2}\}$ | $1 \times 10^{-2}$ |
| ER | Learning rate | $\{1 \times 10^{-2}\}$ | $1 \times 10^{-2}$ |
| | Effective rank lr | $\{1 \times 10^{-2}, 1 \times 10^{-3}, 1 \times 10^{-4}\}$ | $1 \times 10^{-3}$ |
| $L2$-ER | Learning rate | $\{1 \times 10^{-2}\}$ | $1 \times 10^{-2}$ |
| | Effective rank lr | $\{1 \times 10^{-2}, 1 \times 10^{-3}, 1 \times 10^{-4}\}$ | $1 \times 10^{-3}$ |
| | Weight decay | $\{1 \times 10^{-3}, 1 \times 10^{-4}, 1 \times 10^{-5}\}$ | $1 \times 10^{-3}$ |
| CBP | Learning rate | $\{1 \times 10^{-2}\}$ | $1 \times 10^{-2}$ |
| | Replacement rate | $\{1 \times 10^{-4}, 1 \times 10^{-5}, 1 \times 10^{-6}\}$ | $1 \times 10^{-6}$ |
| $L2$ | Learning rate | $\{1 \times 10^{-2}\}$ | $1 \times 10^{-2}$ |
| | Weight decay | $\{1 \times 10^{-3}, 1 \times 10^{-4}, 1 \times 10^{-5}\}$ | $1 \times 10^{-3}$ |
| LayerNorm-L2 | Learning rate | $\{1 \times 10^{-2}\}$ | $1 \times 10^{-2}$ |
| | Weight decay | $\{1 \times 10^{-2}, 1 \times 10^{-3}, 1 \times 10^{-4}\}$ | $1 \times 10^{-5}$ |
| Spectral Regularizer | Learning rate | $\{1 \times 10^{-2}\}$ | $1 \times 10^{-2}$ |
| | Spectral Strength | $\{1 \times 10^{-2}, 1 \times 10^{-3}, 1 \times 10^{-4}\}$ | $1 \times 10^{-3}$ |

# I. Continual ImageNet

Continual ImageNet (Dohare et al., 2024) is an adaptation of ImageNet (Krizhevsky et al., 2012) in which pairs of classes are randomly sampled to form binary classification tasks. We evaluate performance on a total of 2000 tasks.

## I.1. Network Architecture

We adopt the same architecture as (Dohare et al., 2024) with one key modification. In their setup, the final layer of the network is reinitialized at the beginning of every task. To ensure fairness in comparison, we remove this feature and keep the final layer fixed across tasks. We use a threeblock convolutional network followed by two dense layers and a dense classifier. Each convolution is followed by a ReLU nonlinearity and a $2 \times 2$ maxpool with stride 2. The architecture is summarized as follows:

```
ConvNet(
  Sequential(
    (0): Conv2d(out_channels=32, kernel_size=5x5, bias=True)
    (1): ReLU()
    (2): MaxPool(window=2x2, stride=2)

    (3): Conv2d(out_channels=64, kernel_size=3x3, bias=True)
    (4): ReLU()
    (5): MaxPool(window=2x2, stride=2)

    (6): Conv2d(out_channels=128, kernel_size=3x3, bias=True)
    (7): ReLU()
    (8): MaxPool(window=2x2, stride=2)
    (9): Flatten()

    (10): Dense(out_dims=128, bias=True)
    (11): ReLU()
    (12): Dense(out_dims=128, bias=True)
    (13): ReLU()
    (14): Dense(out_dims=2, bias=True)
  )
)
```

## I.2. Hyperparameters

In Table 3, we present the default hyperparameters for our ImageNet experiments. Unless otherwise specified, experiments use these defaults.

## I.3. Experiments

Due to high variance, we conducted full hyperparameter sweeps from Table 4 across 10 seeds and apply Savitzky-Golay filter (Savitzky & Golay, 1964) to the results. The best hyperparameters selected from these sweeps are summarized in Table 4.

## I.4. Continual ImageNet Hessian Spectrum

We again use the best hyperparameters from Table 4 to run over 10 seeds to plot the hessian spectrum. We presents our results on BP, ER, L2-ER, CBP, L2 in Figure 11 and its corresponding performance in Figure 4. Note that we categorize ER as preserving plasticity in this case because, in this single run, ER successfully maintained plasticity rather than losing it. We present this single run hessian spectrum accuracy in Figure 10. From Figure 11, we observe that all algorithms that lose plasticity experience spectral collapse, whereas those that preserve plasticity maintain a wide and dense spectrum throughout training.

*Table 3.* ImageNet default hyperparameters

| Hyperparam Name | Default | Description |
| --- | --- | --- |
| study_name | test | experiment name |
| seed | 2024 | base random seed |
| debug | False | true to enable debug mode |
| platform | gpu | {cpu, gpu} |
| n_seeds | 1 | number of seeds |
| env | imagenet | environment name |
| agent | l2_er | agent options: {er, bp, l2, snp_l2, snp, cbp, l2_er, laynorm_l2, spectral_reg} |
| alg | ppo | algorithm type: {actor_critic, ppo} |
| activation | relu | activation options: {relu, tanh} |
| lr | [0.01] | learning rate(s) |
| optimizer | sgd | {adam, sgd} |
| weight_decay | 0.001 | $L2$ weight decay |
| to_perturb | False | whether to perturb the input data |
| perturb_scale | $1 \times 10^{-5}$ | magnitude of input perturbation |
| mini_batch_size | 100 | minibatch size |
| no_anneal_lr | True | if true, do not anneal the learning rate |
| max_grad_norm | $1 \times 10^{9}$ | gradient clipping threshold |
| num_tasks | 2000 | number of tasks used in training/eval |
| num_epochs | 20 | number of epochs per task |
| momentum | 0.9 | SGD momentum coefficient |
| **effective rank** | | |
| er_lr | [0.01] | ER learning rate(s) |
| er_batch | 12 | batch size for ER computation |
| er_step | 1 | ER update frequency |
| **model architecture** | | |
| use_layernorm | False | enable LayerNorm in the model |
| **spectral regularization** | | |
| use_spectral_reg | False | enable spectral regularization |
| spectral_reg_strength | 0.1 | regularization strength |
| spectral_k | 2 | number of singular values/eigs (or rank) used |
| spectral_target | 2.0 | target value for spectral objective |
| spectral_power_iter | 10 | power-iteration steps (if used) |
| **evaluation** | | |
| evaluate | True | evaluate after each task |
| evaluate_previous | False | also evaluate on previous tasks |
| eval_size | 2000 | number of evaluation samples per task |
| compute_hessian | False | whether to compute Hessian spectrum |
| compute_hessian_size | 2000 | samples used for Hessian computation |
| compute_hessian_interval | 1 | interval (in tasks) between Hessian runs |
| **continual backpropagation** | | |
| cont_backprop | False | enable CBP |
| replacement_rate | $1 \times 10^{-6}$ | CBP replacement probability per step |
| decay_rate | 0.99 | exponential decay for CBP statistics |
| maturity_threshold | 100 | steps before a unit is considered mature |

*Table 4.* Hyperparameter sweeps and selected best values for Continual ImageNet across all algorithms.

| Algorithm | Hyperparameter | Sweep Hyperparameters | Best Hyperparam |
|---|---|---|---|
| BP | Learning rate | $10^{-2}$, $10^{-3}$, $10^{-4}$ | $10^{-4}$ |
| ER | Learning rate | $10^{-2}$, $10^{-3}$, $10^{-4}$ | $10^{-4}$ |
| | Effective rank lr | $10^{-3}$, $10^{-4}$, $10^{-5}$ | $10^{-5}$ |
| $L2$-ER | Learning rate | $10^{-3}$ | $10^{-3}$ |
| | Effective rank lr | $10^{-3}$, $10^{-4}$, $10^{-5}$ | $10^{-4}$ |
| | Weight decay | $10^{-3}$, $10^{-4}$, $10^{-5}$ | $10^{-3}$ |
| LayerNorm-L2 | Learning rate | $10^{-2}$, $10^{-3}$, $10^{-4}$ | $10^{-3}$ |
| | Weight decay | $10^{-3}$, $10^{-4}$, $10^{-5}$ | $10^{-5}$ |
| Spectral | Learning rate | $10^{-2}$, $10^{-3}$, $10^{-4}$ | $10^{-2}$ |
| | spectral_strengths | $10^{-2}$, $10^{-3}$, $10^{-4}$ | $10^{-2}$ |
| CBP | Learning rate | $10^{-2}$, $10^{-3}$, $10^{-4}$ | $10^{-4}$ |
| | Replacement rate | $10^{-4}$, $10^{-5}$, $10^{-6}$ | $10^{-5}$ |
| $L2$ | Learning rate | $10^{-2}$, $10^{-3}$, $10^{-4}$ | $10^{-4}$ |
| | Weight decay | $10^{-3}$, $10^{-4}$, $10^{-5}$ | $10^{-3}$ |

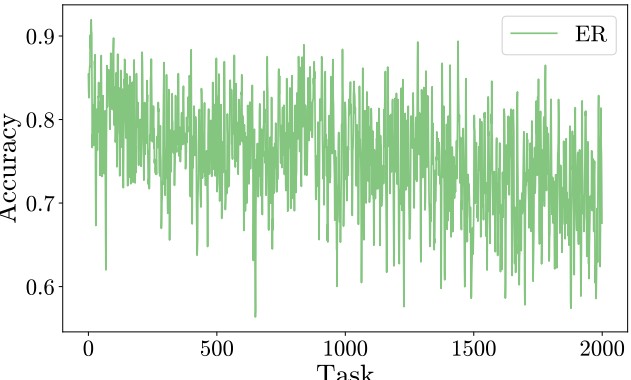

*Figure 10.* ER Accuracy correspond to Hessian Spectrum on Continual ImageNet.

**Algorithms that *lose plasticity***

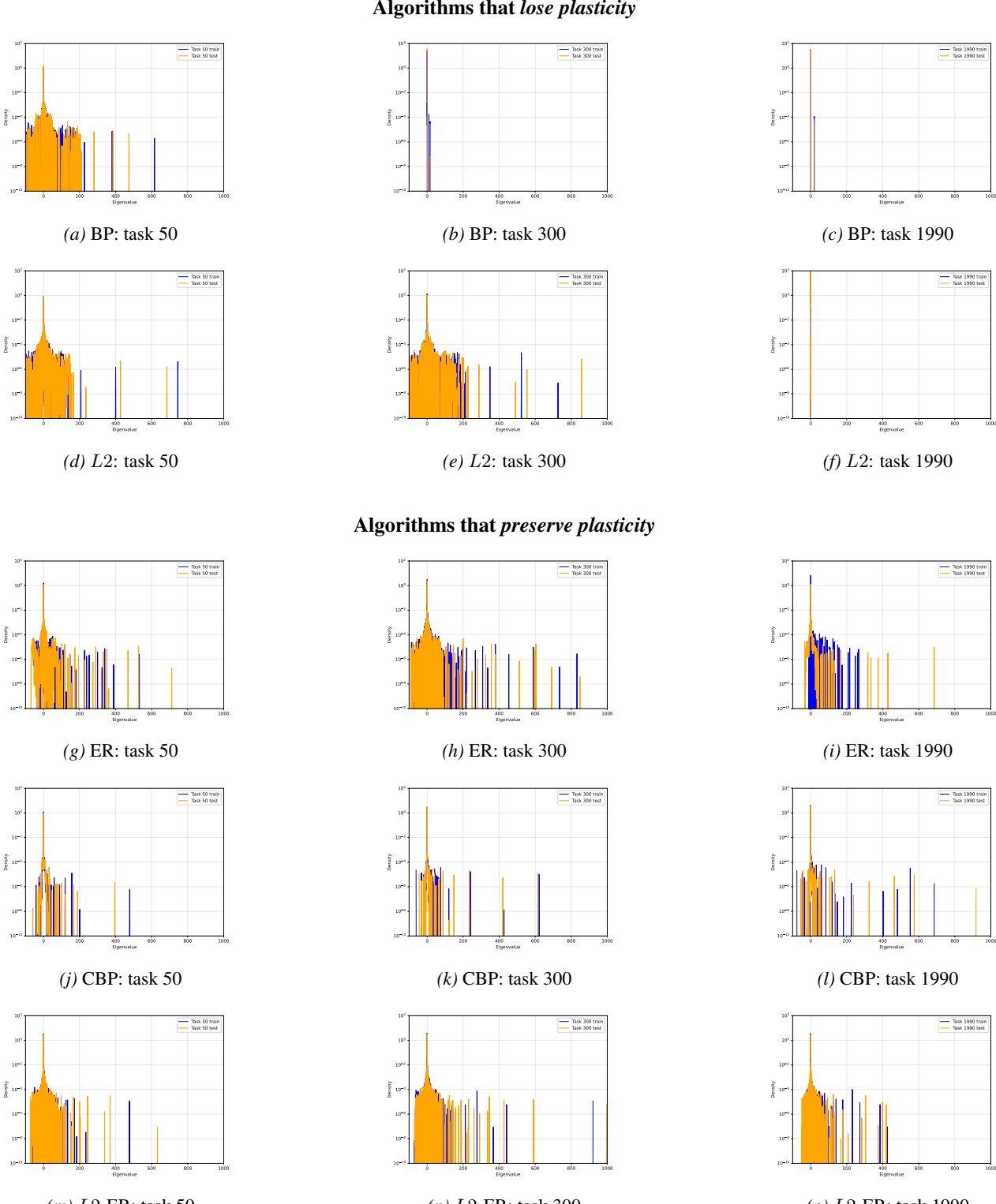

*Figure 11.* Comparison of Hessian eigenspectra at task init on Continual ImageNet. From top to bottom, the algorithms are ordered by increasing accuracy. **Top:** Algorithms that lose plasticity (ER, BP). **Bottom:** Algorithms that preserve plasticity (*L*2, CBP, *L*2-ER).

# J. Incremental CIFAR

Incremental CIFAR (Dohare et al., 2024) is an adaptation of the original CIFAR-100 dataset (Krizhevsky et al., 2009) where classes are incrementally added. Starting with 5 classes, each task adds 5 new classes for classification until all 100 classes are included. Our setup largely follows (Dohare et al., 2024), but we remove random data transformations and learning rate annealing from their design.

## J.1. Network Architecture

We employ a standard ResNet-18 architecture adapted from (Dohare et al., 2024). The network begins with a $3 \times 3$ convolutional stem with 64 channels, followed by four sequential residual stages. Each stage contains two BasicBlocks: Layer1 keeps the width at 64 channels, while Layers 24 progressively double the number of channels to 128, 256, and 512, with the first block of each stage performing downsampling via stride-2 convolutions and $1 \times 1$ skip connections. After the residual stages, a global average pooling layer reduces the spatial dimension, and then pass the resulting feature vector through two fully connected layers with ReLU activations. Finally, a dense classification head outputs logits for the number of classes in the current task.

```
ResNet18(
  Stem(
    Conv2d(out_channels=64, kernel_size=3x3, stride=1, padding=1, bias=True)
    BatchNorm()
    ReLU()
  )
  Layer1: Sequential(
    BasicBlock(64 -> 64, stride=1)(
      Conv2d(64, 3x3, stride=1, padding=1, bias=True), BatchNorm(), ReLU,
      Conv2d(64, 3x3, stride=1, padding=1, bias=True), BatchNorm(),
      Skip: Identity,
      Add, ReLU
    )
    BasicBlock(64 -> 64, stride=1)(
      Conv2d(64, 3x3, stride=1, padding=1, bias=True), BatchNorm(), ReLU,
      Conv2d(64, 3x3, stride=1, padding=1, bias=True), BatchNorm(),
      Skip: Identity,
      Add, ReLU
    )
  )
  Layer2: Sequential(
    BasicBlock(64 -> 128, stride=2)(
      Conv2d(128, 3x3, stride=2, padding=1, bias=True), BatchNorm(), ReLU,
      Conv2d(128, 3x3, stride=1, padding=1, bias=True), BatchNorm(),
      Skip: Conv2d(128, 1x1, stride=2, bias=True) + BatchNorm(),
      Add, ReLU
    )
    BasicBlock(128 -> 128, stride=1)(
      Conv2d(128, 3x3, stride=1, padding=1, bias=True), BatchNorm(), ReLU,
      Conv2d(128, 3x3, stride=1, padding=1, bias=True), BatchNorm(),
      Skip: Identity,
      Add, ReLU
    )
  )
  Layer3: Sequential(
    BasicBlock(128 -> 256, stride=2)(
      Conv2d(256, 3x3, stride=2, padding=1, bias=True), BatchNorm(), ReLU,
      Conv2d(256, 3x3, stride=1, padding=1, bias=True), BatchNorm(),
```

```
      Skip: Conv2d(256, 1x1, stride=2, bias=True) + BatchNorm(),
      Add, ReLU
    )
    BasicBlock(256 -> 256, stride=1)(
      Conv2d(256, 3x3, stride=1, padding=1, bias=True), BatchNorm(), ReLU,
      Conv2d(256, 3x3, stride=1, padding=1, bias=True), BatchNorm(),
      Skip: Identity,
      Add, ReLU
    )
  )
  Layer4: Sequential(
    BasicBlock(256 -> 512, stride=2)(
      Conv2d(512, 3x3, stride=2, padding=1, bias=True), BatchNorm(), ReLU,
      Conv2d(512, 3x3, stride=1, padding=1, bias=True), BatchNorm(),
      Skip: Conv2d(512, 1x1, stride=2, bias=True) + BatchNorm(),
      Add, ReLU
    )
    BasicBlock(512 -> 512, stride=1)(
      Conv2d(512, 3x3, stride=1, padding=1, bias=True), BatchNorm(), ReLU,
      Conv2d(512, 3x3, stride=1, padding=1, bias=True), BatchNorm(),
      Skip: Identity,
      Add, ReLU
    )
  )
  GlobalAveragePool()
  Dense(out_dims=512, bias=True), ReLU()
  Dense(out_dims=512, bias=True), ReLU()
  Dense(out_dims=num_classes, bias=True)
)
```

### J.2. Hyperparameters

In Table 5, we present the default hyperparameters for our ImageNet experiments. Unless otherwise specified, experiments use these defaults.

### J.3. Experiments

We conduct our hyperparameter sweep experiments over 5 seeds for all hyperparameters listed in Table 6, and report the selected best hyperparameters in Table 6.

### J.4. Incremental CIFAR hessian spectrum

We use the best hyperparameters in Table 6 to plot the Hessian spectrum over 5 seeds, with results shown in Figure 12. Again consistent with our previous experiments, algorithms that preserve plasticity maintain a dense Hessian spectrum, whereas those that lose plasticity exhibit a sparse spectrum. This demonstrates our claim that spectral collapse is a clear indicator of plasticity loss.

| Hyperparam Name | Default | Description |
|---|---|---|
| study_name | test | experiment name |
| seed | 2024 | base random seed |
| debug | False | true to enable debug mode |
| platform | gpu | {cpu, gpu} |
| n_seeds | 1 | number of seeds |
| env | incremental_cifar | environment name |
| agent | l2 | agent options: {er, bp, l2, snp_l2, snp, cbp, l2_er} |
| alg | ppo | algorithm type: {actor_critic, ppo} |
| activation | relu | activation options: {relu, tanh} |
| lr | [0.1] | learning rate(s) |
| optimizer | sgd | {adam, sgd} |
| weight_decay | 0.0005 | $L2$ weight decay |
| to_perturb | False | whether to perturb the input data |
| perturb_scale | $1 \times 10^{-5}$ | magnitude of input perturbation |
| mini_batch_size | 100 | minibatch size |
| no_anneal_lr | False | if true, do not anneal the learning rate |
| max_grad_norm | $1 \times 10^{9}$ | gradient clipping threshold |
| num_tasks | 20 | number of tasks used in training/eval |
| num_epochs | 200 | number of epochs per task |
| momentum | 0.9 | SGD momentum coefficient |
| **effective rank** | | |
| er_lr | [0.01] | ER learning rate |
| er_batch | 5 | batch size for ER computation |
| er_step | 1 | ER update frequency |
| **resetting** | | |
| reset | False | reset the network after each task |
| **evaluation** | | |
| evaluate | True | evaluate after each task |
| evaluate_previous | False | evaluate on previous tasks |
| eval_size | 2000 | number of evaluation samples per task |
| compute_hessian | False | whether to compute Hessian spectrum |
| compute_hessian_size | 2000 | samples used for Hessian computation |
| compute_hessian_interval | 1 | interval in tasks between Hessian runs |
| **continual backpropagation** | | |
| cont_backprop | False | enable CBP |
| replacement_rate | $1 \times 10^{-6}$ | CBP replacement probability per step |
| decay_rate | 0.99 | exponential decay for CBP statistics |
| maturity_threshold | 100 | steps before a unit is considered mature |

*Table 5.* Incremental CIFAR default hyperparameters

**Algorithms that *lose plasticity***

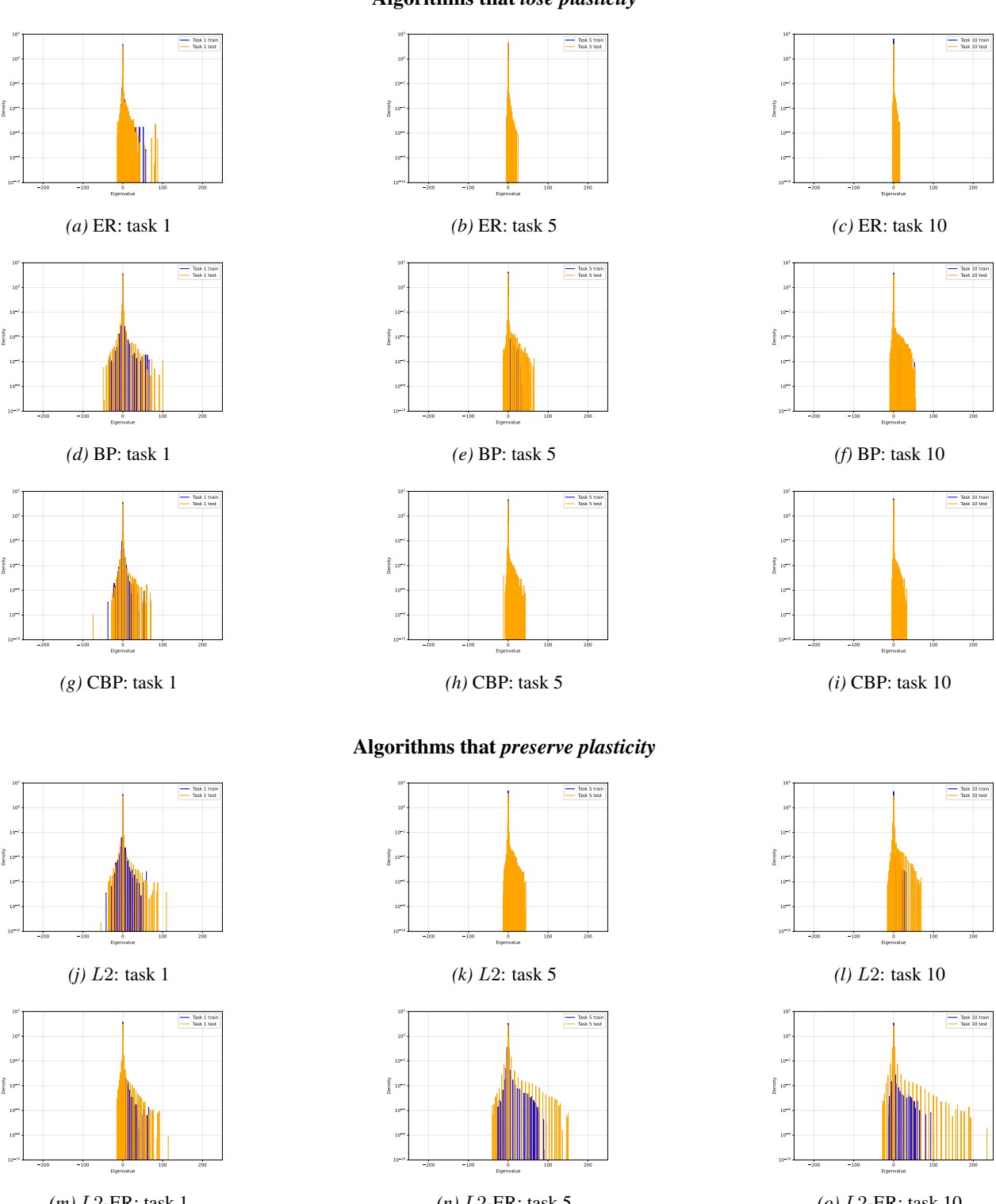

*(a)* ER: task 1

*(b)* ER: task 5

*(c)* ER: task 10

*(d)* BP: task 1

*(e)* BP: task 5

*(f)* BP: task 10

*(g)* CBP: task 1

*(h)* CBP: task 5

*(i)* CBP: task 10

**Algorithms that *preserve plasticity***

*(j)* $L2$: task 1

*(k)* $L2$: task 5

*(l)* $L2$: task 10

*(m)* $L2$-ER: task 1

*(n)* $L2$-ER: task 5

*(o)* $L2$-ER: task 10

*Figure 12.* Comparison of Hessian eigenspectra on Incremental CIFAR. Algorithms are ordered by increasing accuracy. **Top:** Algorithms that lose plasticity (ER, BP, CBP). **Bottom:** Algorithms that preserve plasticity ($L2$, $L2$-ER).

| Algorithm | Hyperparameter | Sweep Values | Best Value |
|---|---|---|---|
| BP | Learning rate | $\{1 \times 10^{-2}, 1 \times 10^{-3}, 1 \times 10^{-4}\}$ | $1 \times 10^{-2}$ |
| ER | Learning rate | $\{1 \times 10^{-2}, 1 \times 10^{-3}, 1 \times 10^{-4}\}$ | $1 \times 10^{-2}$ |
| | Effective rank lr | $\{1 \times 10^{-3}, 1 \times 10^{-4}, 1 \times 10^{-5}\}$ | $1 \times 10^{-4}$ |
| $L2$-ER | Learning rate | $\{1 \times 10^{-2}\}$ | $1 \times 10^{-2}$ |
| | Effective rank lr | $\{1 \times 10^{-2}, 1 \times 10^{-3}, 1 \times 10^{-4}\}$ | $1 \times 10^{-3}$ |
| | Weight decay | $\{1 \times 10^{-3}, 1 \times 10^{-4}, 1 \times 10^{-5}\}$ | $1 \times 10^{-3}$ |
| CBP | Learning rate | $\{1 \times 10^{-2}, 1 \times 10^{-3}, 1 \times 10^{-4}\}$ | $1 \times 10^{-2}$ |
| | Replacement rate | $\{1 \times 10^{-4}, 1 \times 10^{-5}, 1 \times 10^{-6}\}$ | $1 \times 10^{-6}$ |
| $L2$ | Learning rate | $\{1 \times 10^{-2}, 1 \times 10^{-3}, 1 \times 10^{-4}\}$ | $1 \times 10^{-2}$ |
| | Weight decay | $\{1 \times 10^{-3}, 1 \times 10^{-4}, 1 \times 10^{-5}\}$ | $1 \times 10^{-4}$ |
| LayerNorm-L2 | Learning rate | $\{1 \times 10^{-2}, 1 \times 10^{-3}, 1 \times 10^{-4}\}$ | $1 \times 10^{-2}$ |
| | Weight decay | $\{1 \times 10^{-3}, 1 \times 10^{-4}, 1 \times 10^{-5}\}$ | $1 \times 10^{-4}$ |
| Spectral | Learning rate | $\{1 \times 10^{-2}, 1 \times 10^{-3}, 1 \times 10^{-4}\}$ | $1 \times 10^{-2}$ |
| | Spectral Strengths | $\{1 \times 10^{-3}, 1 \times 10^{-4}, 1 \times 10^{-5}\}$ | $1 \times 10^{-2}$ |

*Table 6.* Hyperparameter sweeps and selected best values for Incremental CIFAR across all algorithms.

# K. Slippery Ant

Slippery Ant is a continual reinforcement learning environment where the friction between the ant and the ground changes every 2 million steps. The agent has to adapt its policy to this new friction. In our experiments, we use PPO (Schulman et al., 2017) as our base algorithm, which is an online learning method designed for training on vectorized environments. It is parallelized using the JAX library (Bradbury et al., 2018) based on a batch experimentation library written in JAX (Lu et al., 2022).

## K.1. Network Architecture

```
ActorCritic(
  Actor(
    Sequential(
      (0): Dense(in_dims=hidden_size, out_dims=hidden_size, bias=True)
      (1): ReLU()
      (2): Dense(in_dims=hidden_size, out_dims=action_dims, bias=True)
      (3): Categorical() or MultivariateNormalDiag()
    )
  )
  Critic(
    Sequential(
      (0): Dense(in_dims=hidden_size, out_dims=hidden_size, bias=True)
      (1): ReLU()
      (2): Dense(in_dims=hidden_size, out_dims=1, bias=True)
    )
  )
)
```

## K.2. Hyperparameters

The default hyperparameters is in Table 8. Unless otherwise specified, experiments use the default hyperparameters.

| Algorithm | Hyperparameter | Sweep Values | Best Value |
|---|---|---|---|
| BP | Learning rate | $\{1 \times 10^{-2}, 1 \times 10^{-3}, 1 \times 10^{-4}\}$ | $1 \times 10^{-4}$ |
| ER | Learning rate | $\{1 \times 10^{-2}, 1 \times 10^{-2}, 1 \times 10^{-4}\}$ | $1 \times 10^{-4}$ |
| | Effective rank lr | $\{1 \times 10^{-3}, 1 \times 10^{-4}, 1 \times 10^{-5}\}$ | $1 \times 10^{-7}$ |
| $L2$-ER | Learning rate | $\{1 \times 10^{-4}\}$ | $1 \times 10^{-4}$ |
| | Effective rank lr | $\{1 \times 10^{-5}, 1 \times 10^{-6}, 1 \times 10^{-7}\}$ | $1 \times 10^{-6}$ |
| | Weight decay | $\{1 \times 10^{-3}, 1 \times 10^{-4}, 1 \times 10^{-5}\}$ | $1 \times 10^{-3}$ |
| CBP+$L2$ | Learning rate | $\{1 \times 10^{-4}\}$ | $1 \times 10^{-4}$ |
| | Replacement rate | $\{1 \times 10^{-5}, 1 \times 10^{-6}, 1 \times 10^{-7}\}$ | $1 \times 10^{-7}$ |
| | Weight decay | $\{1 \times 10^{-3}, 1 \times 10^{-4}, 1 \times 10^{-5}\}$ | – |
| $L2$ | Learning rate | $\{1 \times 10^{-2}, 1 \times 10^{-3}, 1 \times 10^{-4}\}$ | $1 \times 10^{-4}$ |
| | Weight decay | $\{1 \times 10^{-3}, 1 \times 10^{-4}, 1 \times 10^{-5}\}$ | $1 \times 10^{-3}$ |
| LayerNorm-L2 | Learning rate | $\{1 \times 10^{-2}, 1 \times 10^{-3}, 1 \times 10^{-4}\}$ | $1 \times 10^{-4}$ |
| | Weight decay | $\{1 \times 10^{-3}, 1 \times 10^{-4}, 1 \times 10^{-5}\}$ | $1 \times 10^{-3}$ |
| Spectral | Learning rate | $\{1 \times 10^{-2}, 1 \times 10^{-3}, 1 \times 10^{-4}\}$ | $1 \times 10^{-4}$ |
| | Weight decay | $\{1 \times 10^{-3}, 1 \times 10^{-4}, 1 \times 10^{-5}\}$ | $1 \times 10^{-3}$ |

*Table 7.* Hyperparameter sweeps and selected best values for Slippery Ant across all algorithms.

| Hyperparam Name | Default | Description |
| --- | --- | --- |
| env | slippery_ant | environment name |
| num_envs | 1 | number of parallel environments |
| gamma | 0.99 | discount factor |
| num_steps | 2048 | steps per rollout |
| update_epochs | 10 | number of optimization epochs per update |
| num_minibatches | 128 | number of minibatches per epoch |
| activation | relu | activation function: {relu, tanh} |
| optimizer | adam | optimizer: {adam, sgd, muon} |
| lr | [2.5e-4] | learning rate(s) |
| lambda0 | [0.95] | GAE parameter $\lambda$ |
| vf_coeff | [1] | value function loss coefficient |
| weight_decay | 0.0 | $L2$ weight decay |
| beta_1 | 0.9 | Adam $\beta_1$ coefficient |
| beta_2 | 0.999 | Adam $\beta_2$ coefficient |
| **continual backpropagation** | | |
| cont_backprop | False | enable CBP |
| replacement_rate | $1 \times 10^{-4}$ | CBP replacement probability per step |
| decay_rate | 0.99 | exponential decay for CBP statistics |
| maturity_threshold | $1 \times 10^4$ | steps before a unit is considered mature |
| **effective rank** | | |
| er | False | enable ER regularization |
| er_lr | [0.01] | ER learning rate |
| er_batch | 128 | batch size for ER computation |
| er_step | 1 | ER update frequency |
| **hessian computation** | | |
| compute_hessian_init | False | compute Hessian at initialization |
| compute_hessian_end | False | compute Hessian at the end |
| compute_hessian_size | 2000 | number of samples for Hessian computation |
| compute_hessian_interval | 1 | interval (in tasks) between Hessian runs |
| **PPO** | | |
| hidden_size | 256 | size of hidden layers |
| total_steps | $5 \times 10^6$ | total number of training steps |
| entropy_coeff | 0.01 | entropy regularization coefficient |
| clip_eps | 0.2 | PPO clipping parameter |
| max_grad_norm | $1 \times 10^9$ | gradient clipping threshold |
| anneal_lr | False | anneal learning rate over training |
| **evaluation** | | |
| steps_log_freq | 4 | logging frequency in steps |
| update_log_freq | 8 | logging frequency in updates |
| save_checkpoints | False | save checkpoints during training |
| save_runner_state | False | save final runner state |
| seed | 2020 | random seed |
| n_seeds | 5 | number of random seeds |
| platform | gpu | {cpu, gpu} |
| debug | False | enable debug mode |
| show_discounted | False | show discounted returns in logs |
| study_name | batch_ppo_test | experiment name |
| change_every | $2 \times 10^6$ | steps between environment changes |
| friction_seed | 0 | seed for friction schedule |

*Table 8.* Non-stationary policy default hyperparameters

### K.3. Experiments

We swept the environment over 5 seeds for all hyperparameters in Table 7. Following the architecture of CBP in (Dohare et al., 2024), we use CBP together with $L2$ normalization instead of just CBP. With the additional sweep on weight decays, we fixed the learning rate of CBP constant. The best hyperparameters are reported in Table 7.

## L. Runtime

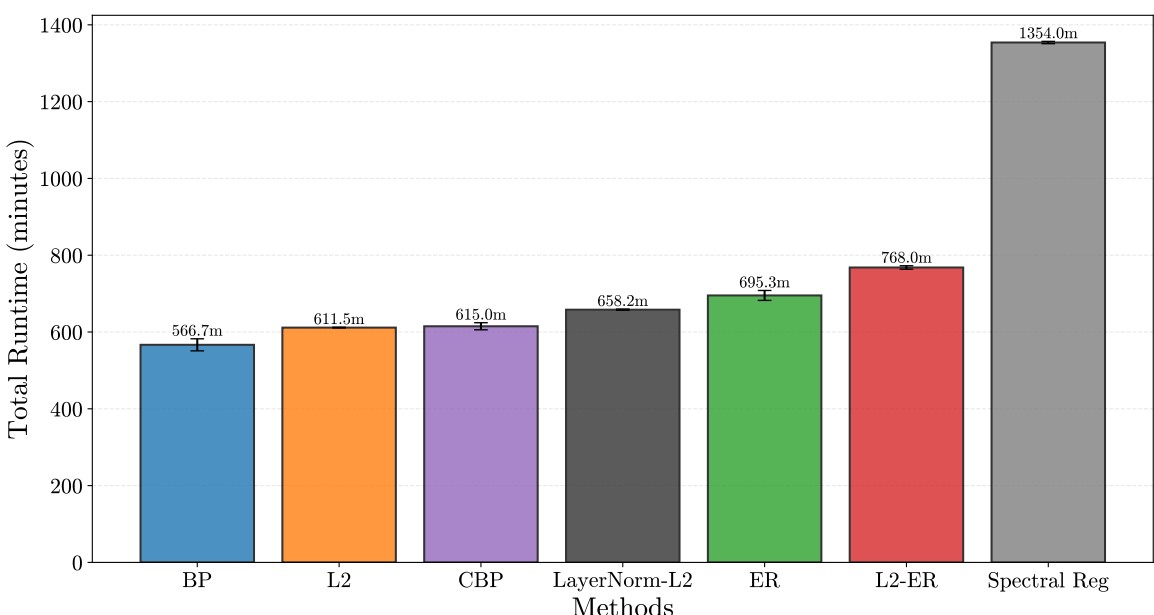

*Figure 13.* Runtime in Permuted MNIST Across all methods

## Runtime Comparison: ImageNet

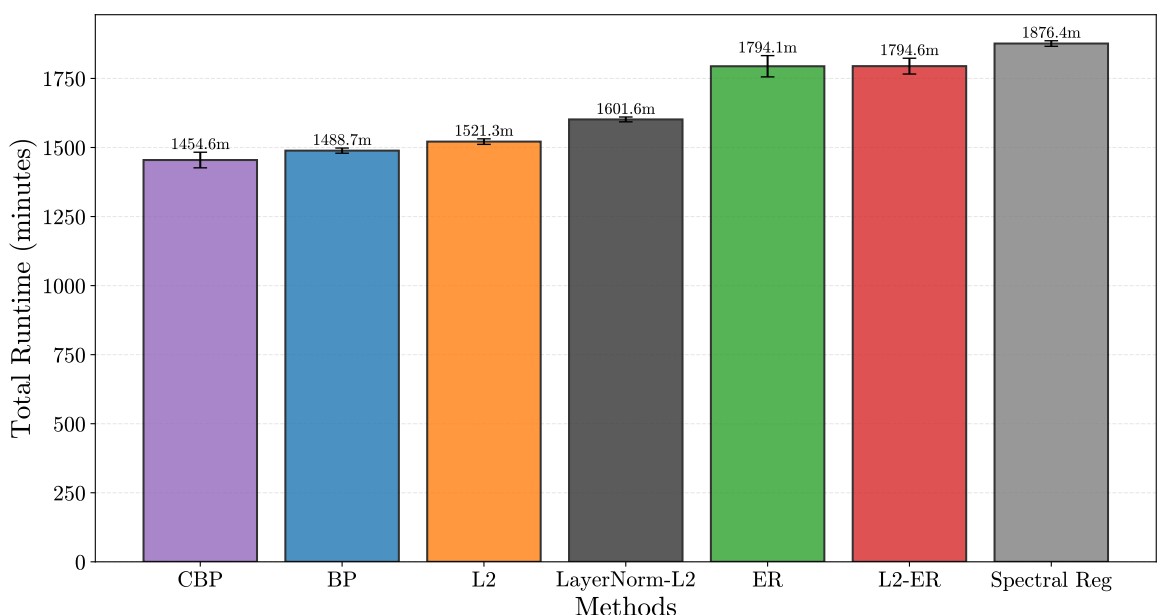

*Figure 14.* Runtime in ImageNet Across all methods

## Runtime Comparison: Incremental CIFAR

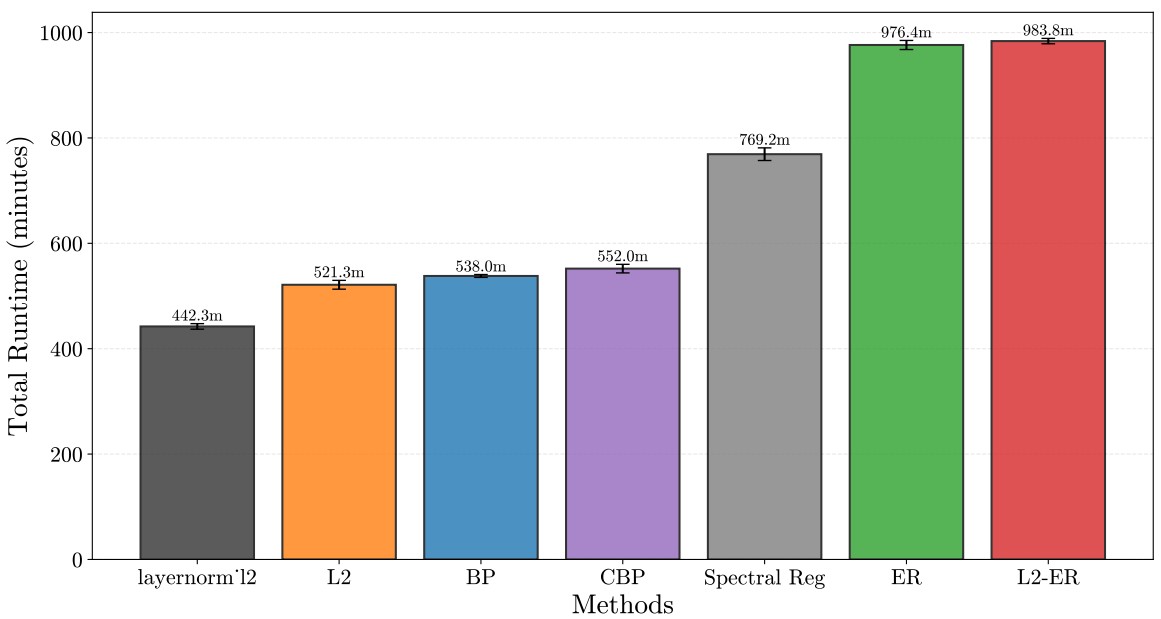

*Figure 15.* Runtime in Incremental CIFAR Across all methods

