# OpenReview forum: "Spectral Collapse Drives Loss of Plasticity in Deep Continual Learning"
_ICML.cc/2026/Conference — ICML 2026 regular_

### Official Review · Reviewer_Ecm1 · 2026-03-05

**Soundness:** 3
**Presentation:** 4
**Significance:** 3
**Originality:** 3
**Overall Recommendation:** 5
**Confidence:** 3

**Summary:**

The paper investigates the phenomenon of plasticity loss in deep continual learning. The authors propose that this failure is driven by a phenomenon they call Hessian spectral collapse, in which the eigenspectrum of the Hessian degenerates and most eigenvalues become very small. When this happens, the loss landscape becomes nearly flat in many directions, causing gradient-based optimisation to be ineffective and preventing the network from adapting to new tasks.

The paper studies training dynamics in a linearised ReLU network within the Neural Tangent Kernel (NTK) regime. The authors derive theoretical conditions for successful learning based on the ε-rank of the Gram matrix, showing that learning requires a sufficient number of eigenvalues above a threshold that depends on the learning rate and optimisation budget. When the number of such eigenvalues shrinks across tasks, the network loses the capacity to reduce training residuals for new tasks. The analysis also establishes a connection between the loss-weighted Gram matrix and the Generalized Gauss–Newton (GGN) approximation of the Hessian, linking function-space NTK analysis to parameter-space curvature.

The authors introduce a regularisation approach called L2-ER, which combines L2 weight regularisation with an effective-rank regularisation of layer activations. The method aims to maintain a high effective feature rank and prevent curvature degeneration, thereby preserving the dimensionality of the trainable subspace.

**Compliance With Llm Reviewing Policy:**

Affirmed.

**Key Questions For Authors:**

1- The theoretical analysis is developed under strong assumptions such as the NTK regime, linearised ReLU networks, and frozen Jacobians. Could the authors clarify to what extent the proposed spectral collapse mechanism is expected to hold in fully nonlinear networks trained with standard gradient descent? Additional empirical evidence demonstrating that the same spectral behaviour appears in realistic training settings would strengthen the claim that the theory explains plasticity loss in practical architectures. A convincing clarification or additional analysis could increase my confidence in the generality of the proposed explanation.

2- The paper shows a correlation between the ε-rank of the Hessian (or its approximations) and training performance across tasks. To what extent can the authors demonstrate a causal relationship between spectral collapse and plasticity loss rather than a diagnostic correlation? For example, are there controlled experiments where manipulating the spectrum alone directly changes plasticity outcomes? Evidence supporting a causal link would strengthen the central claim of the paper.

3- The proposed L2-ER regulariser combines effective-rank regularisation with L2 weight decay. Could the authors provide more detailed ablation studies quantifying the individual contribution of each component across the evaluated benchmarks? In particular, it would be helpful to understand whether the improvements are primarily driven by the effective-rank term, the L2 term, or their interaction. Clearer evidence here could strengthen the argument that the combined regulariser specifically targets spectral collapse.

**Limitations:**

The paper includes only a brief impact statement and does not meaningfully discuss the limitations of the proposed approach. In particular, the authors could better acknowledge that the theoretical analysis relies on strong assumptions (e.g., the NTK regime, linearised networks, and frozen Jacobians) that may not fully capture the behaviour of practical deep networks used in the experiments. It would also be helpful to discuss the potential limitations of relying on Hessian or Gauss–Newton approximations and KFAC-based estimates in large-scale models, including computational overhead and approximation accuracy.

**Strengths And Weaknesses:**

The paper proposes a theoretical explanation for plasticity loss in continual learning, attributing it to Hessian spectral collapse, where the eigenspectrum degenerates and only a small number of parameter directions remain useful for learning. The work addresses an important problem in continual learning and offers a perspective that plasticity loss can be understood through curvature properties of the optimisation landscape. The paper is generally well organised, with a clear progression from motivation to theory, method, and experiments. Figures illustrating the relationship between ε-rank and performance help communicate the main idea.
In terms of originality, the main contribution is framing spectral collapse as a unifying explanation for several previously observed mechanisms of plasticity loss. The submission’s significant contribution pertains to connecting continual learning plasticity to curvature properties of the optimisation landscape through ε-rank analysis. The authors strive to explore a broad context by relating prior observations within a unified spectral framework. The proposed L2-ER regulariser combines existing ideas (L2 regularisation and effective-rank regularisation), so the algorithmic novelty is moderate, but the theoretical interpretation and empirical study provide useful insights into continual learning dynamics.

However, the theoretical analysis relies on strong assumptions such as the NTK regime, linearised networks, and frozen Jacobians, which may not fully capture the behaviour of practical deep networks used in the experiments. While the authors motivate the extension to realistic networks through Gauss–Newton and KFAC approximations, the theoretical guarantees do not directly apply to the architectures evaluated empirically. Some results would benefit from additional ablations to better isolate the individual effects of the effective-rank and L2 components of the proposed regulariser. Also, While the empirical improvements over existing baselines are moderate, the conceptual framing may help guide future research on optimisation-aware continual learning methods. However, parts of the theoretical development are dense and introduce notation quickly, making the transition from the NTK-based analysis to the practical regularisation method somewhat difficult to follow. Clarifying this connection would improve readability.

However, the theoretical analysis relies on strong assumptions such as the NTK regime, linearised networks, and frozen Jacobians, which may not fully capture the behaviour of practical deep networks used in the experiments. While the authors motivate the extension to realistic networks through Gauss–Newton and KFAC approximations, the theoretical guarantees do not directly apply to the architectures evaluated. Some results would benefit from additional ablation studies to better isolate the individual effects of the effective-rank and L2 components of the proposed regulariser. While the experimental improvements over existing baselines are moderate, the conceptual framing may still help guide future research on optimisation-aware continual learning methods. In addition, parts of the theoretical development are dense and introduce notation rapidly, making the transition from the NTK-based analysis to the practical regularisation method somewhat difficult to follow. Clarifying this connection would improve readability.

---

> ### Author Rebuttal · Authors · 2026-03-30
>
> We thank the reviewer for their careful reading and for highlighting the boundaries of our theoretical framework.
>
> **2. Causality**
>
> Our goal is to start with a simple model that can be analyzed in order to find and control the reasons for loss of plasticity. Our claim has two levels. First, in the NTK setting, Lemma 4.7 shows that if too many eigenvalues of the kernel matrix, G, lie below the threshold $\epsilon\_{K, \\gamma}$, the subsequent task cannot be learned within K steps (with some exceptions). This means that the bulk of the eigenspectrum of the G must be wide.
>
> We further isolate this mechanism from task changes on Permuted MNIST in the NTK regime. Empirically, we demonstrate that the weighted kernel matrix collapses across tasks even though the kernel matrix at initialization $G^{(0)}$ is invariant to pixel permutations. Then we show that controlling the feature rank of this weighted matrix controls the rate of plasticity loss. Specifically, when we use a low-feature rank initialization which corresponds to a narrow kernel matrix bulk,  plasticity loss is worse than a full-feature rank initialization, shown in Figure 2.  We then empirically show that this mechanism remains operational in deep non-linear networks. Our L2ER regularizer increases the rank of the Hessian by increasing the feature rank. Our empirical results show that when we apply our intervention, the spectrum of the Hessian retains a wide bulk and maintains plasticity. When we remove the the intervention, the bulk collapses and so does plasticity. Moreover, we measure the spectrum at the initialization of each new task, showing that it is a precursor to the inability to learn.  In the revision, we will make this two-level claim explicit in the Introduction/Discussion and separate the exact linearized causality statement from the deep-network intervention evidence. We will also move Figure 5c to the main text to demonstrate that $G^{(0)}$ remains invariant throughout training.
>
> **1. Empirical Evidence**
>
> Specifically, our empirical results show that our L2ER regularize maintains a wide Hessian bulk with an MLP for permuted MNIST, a CNN for continual Imagenet, and a ResNet for continual CIFAR. Figure 3 in the main text and Figure 7 in the Appendix show the relationship between the eigenspectrum distribution at initialization of the task and accuracy (at the end of training) on these deep non-linear networks.  We are able to fit a linear relationship between the accuracy and normalized $\epsilon$-rank. That is, the fraction of eigenvalues above a chosen $\epsilon$. In general, plasticity preserving methods are to the top right of the chart, indicating high accuracy and a wide Hessian eigenspectrum bulk. Methods that do poorly have their eigenvalues concentrated below epsilon, which is associated with poor accuracy.  We also provide the actual eigenspectra plots in figures 9,11 and 13, our method as well as baselines.
>
> While it is difficult to control the eigenspectrum of the Hessian directly, we believe our model also serves to explain why other plasticity methods work. In our K-FAC analysis, the approximate Generalized Gauss-Newton matrix is $\\hat{H}\_{GGN}=\\mathbb{E}[aa^\\top]\\otimes\\mathbb{E}[gg^\\top]$, so $\\mathrm{rank}(\\hat{H}_{GGN})=\\mathrm{rank}(\\mathbb{E}[aa^\\top])\\times\\mathrm{rank}(\\mathbb{E}[gg^\\top])$. Thus, low feature rank collapses $\mathbb{E}[aa^\top]$, collinear gradients collapse $\mathbb{E}[gg^\top]$, and dead ReLUs also upper-bound Hessian rank [1]. This gives a common mathematical lens for several causes of plasticity loss, and also helps explain why methods like CBP work: resetting dormant neurons restores variance to $\mathbb{E}[aa^\top]$, thereby indirectly preserving Hessian rank. Figures 3 and 7 are consistent with this view. We will add this discussion.
>
> **3. Ablating L2 and ER**
>
> We find that **both** L2 (orange) and ER (green) are needed to preserve plasticity; each loses plasticity on its own but preserves plasticity when applied together (shown in Figure 4). We believe this finding is non-trivial. Both are needed to ensure that the increasing rank of the proxy $H\_{GGN}$ is guaranteed to increase the rank of the full Hessian since the additional residual matrix needs to be taken into account. We will be sure to highlight this fact more clearly in the discussion of our experimental results.
>
> **Additional limitations**
>
> Please see our comment to reviewer HRC3 for commentary on our limitations, which include the NTK to deep non-linear network gap, KFAC error, and runtime analysis, which we will add to the discussion of our method.
>
>
> We hope we have addressed all your concerns, and we hope you will consider raising the score to reflect that.
>
> [1] Singh, Sidak Pal, Gregor Bachmann, and Thomas Hofmann. “Analytic insights into structure and rank of neural network Hessian maps.” NeurIPS 34 (2021): 23914–23927.

---

> > ### Author Rebuttal · Reviewer_Ecm1 · 2026-04-02
> >
> > I thank authors for their comments. Their response is satisfactory and I am happy to keep my scores.

---

### Official Review · Reviewer_hA2p · 2026-03-10

**Soundness:** 3
**Presentation:** 2
**Significance:** 3
**Originality:** 3
**Overall Recommendation:** 3
**Confidence:** 3

**Summary:**

The paper investigates the phenomenon of plasticity loss in deep continual learning, proposing that this failure is driven by "Hessian spectral collapse". The authors provide theoretical grounding using a linearized ReLU network under the NTK regime. Building on this, they propose L2-ER, a regularization method combining an L2 penalty with an Effective Rank (ER) penalty. They evaluate their method across various datasets, showing that it successfully preserves plasticity.

**Compliance With Llm Reviewing Policy:**

Affirmed.

**Final Justification:**

While I recognize the empirical utility of the L2-ER regularization, I must recommend rejection. As demonstrated in my comment, the core issue is not simply a theory-practice gap, but a fundamental logical fallacy. The theoretical model provided can neither support the claim that Continual Learning leads to Spectral Collapse, nor that Spectral Collapse Drives Loss of Plasticity. The paper requires a substantial theoretical restructuring to accurately reflect the true mechanisms of plasticity loss.

**Key Questions For Authors:**

See weakness

**Limitations:**

Yes

**Strengths And Weaknesses:**

Strength: The paper explains why Hessian spectral collapse leads to the loss of plasticity. The derivation of the L2-ER regularizer from the the KFAC approximation is grounded. The authors provide clear visual and quantitative evidence that L2-ER regularizer preserves the plasticity by preserving the rank of the Hessian.

Weakness: 1. The connection between a low-rank Hessian (or related spectral properties) and the loss of plasticity is already established in the literature (e.g., Lewandowski et al., 2024a; 2024b). The paper needs to state more clearly how the claim of "Hessian spectral collapse" fundamentally differs from, or significantly advances, these existing observations.

2. The analysis in Section 4 relies heavily on the NTK regime, where the Hessian is approximately frozen. Consequently, Lemmas 4.2 through 4.7 merely prove that training is difficult if the initial Hessian has small eigenvalues; they do not inherently prove that continual learning causes these eigenvalues to shrink. To substantiate the causal claim that "Spectral Collapse Drives Loss of Plasticity," the theoretical framework should demonstrate how and why the Hessian becomes low-rank as training progresses across sequential tasks, and then how it negatively influences future tasks.

3. Section 5 suffers from a narrative that mixes theoretical derivations with empirical observations and influences the paper's logical progression. It is unclear whether Section 5 serves as a sequential theoretical continuation of Section 4 or merely a parallel empirical verification but with slightly different settings, and which part of Section 5 is empirical observations and which part is theoretical predictions.

4. While Figure 13 demonstrates that the empirical runtime of L2-ER is acceptable for the evaluated benchmarks, the paper lacks a formal discussion on the computational complexity of the effective rank penalty itself. Because computing the effective rank requires SVD on the feature matrices, its computational overhead is expected to scale cubically w.r.t. the feature dimensions. The authors should explicitly discuss this potential computational bottlenecks when applying L2-ER to much larger network architectures.

---

> ### Author Rebuttal · Authors · 2026-03-30
>
> We thank the reviewer for their questions and for pushing us to sharpen the causal framing and relation to existing work.
>
> **Related Work**
> We thank the reviewer for highlighting the foundational work by Lewandowski et al [1,2], which empirically links plasticity loss to the stable rank of the Hessian [1] and spectral properties of the layer-wise Jacobian [2] . Our paper advances these important observations by providing a rigorous mathematical mechanism: through our NTK analysis, we derive explicit epsilon-rank conditions that prove exactly why a shrinking eigenspectrum makes learning new tasks geometrically impossible. Crucially, our NTK framework proves that plasticity loss occurs even when the underlying kernel matrix remains mathematically invariant across tasks; therefore, loss of plasticity is entirely driven by an internal collapse of the weighted kernel matrix.
>
> Furthermore, our framework investigates the *eigenspectrum* of the Hessian, which is a much richer object of analysis than isolated, layer-wise Jacobians, or the rank of the Hessian. Crucially, our analysis reveals that maintaining plasticity requires preserving the _bulk_ of the eigenvalues above the threshold $epsilon\_{K_\gamma} is essential to prevent spectral collapse. While connected, this is a fundamentally different mechanism than constraining the maximum singular value [2], or maintaining a distribution close to initialization [1] . This bulk perspective directly motivates our unique L2-ER regularizer, which maximizes the effective rank of the Generalized Gauss-Newton approximation, offering an intervention that is entirely distinct from previous spectral or Wasserstein penalties. Moreover, we find that L2 and ER are not enough to preserve plasticity individually, but are sufficient when combined, which is a non-trivial finding.
>
> We will be sure to highlight these differences in related work.
>
> [1] Lewandowski, Alex, et al. "Directions of curvature as an explanation for loss of plasticity." arXiv preprint arXiv:2312.00246 (2023).
> [2] Lewandowski, Alex, et al. "Learning continually by spectral regularization." arXiv preprint arXiv:2406.06811 (2024).
>
> **NTK Analysis**
> We thank the reviewer for pushing us to sharpen our NTK analysis. In the NTK setting, lemma 4.7 shows that if too many eigenvalues of the kernel matrix lie below the threshold $\epsilon_{K, \gamma}, the subsequent task cannot be learned within K steps (with some exceptions). Meaning for continual learning, the bulk of eigenspectrum of the kernel matrix must be wide. We go beyond the mere observation that training is difficult with small eigenvalues to show how a wide bulk is necessary for continual learning.
>
> **Narrative**
> Having established the failure modes of the NTK, we begin empirical evaluation of these failures. Empirically, in Section 5, on permuted MNIST, we demonstrate that the weighted kernel matrix collapses across tasks, even though we prove that the kernel matrix at initialization $G^{(0)}$ is invariant to pixel permutations. Finally, we show that controlling the feature rank of this weighted matrix controls the rate of plasticity loss. Specifically, when we use a low-feature rank initialization which corresponds to a narrow- Kernel matrix bulk,  plasticity loss is worse than a full-feature rank initialization, shown in Figure 2.  Then, in Sections 6 and 7, we empirically show that this mechanism remains operational in deep non-linear networks. Our L2-ER regularizer increases the rank of the Hessian by increasing the feature rank. Our empirical results show that when we apply our intervention, the spectrum of the Hessian retains a wide bulk and maintains plasticity. When we remove the intervention, the bulk collapses, and so does plasticity. Moreover, we measure the spectrum at the initialization of each new task, showing that it is a precursor to the inability to learn.  In the revision, we will make this two-level claim explicit in the Introduction/Discussion and separate the exact linearized causality statement from the deep-network intervention evidence. We will also move Figure 5c to the main text to demonstrate that $G^{(0)} remains invariant throughout training. We will amend section 5 to be clearer that this is an empirical verification of our theoretical setting.
>
> **runtime**
> For layer $l$, we construct a stacked feature matrix $M_l\in\mathbb{R}^{m_l\times d_l}$, where $d_l$ is the layer width and $m_l$ is the number of buffered features used in one ER update. Computing $ER(M_l)$ requires an SVD with cost $O(m_l d_l \min\\{m_l,d_l\\})$; amortized over update interval $U$, the per-step overhead is $O \frac{1}{U} \sum_l (m_l d_l \min\\{m_l,d_l\\})$. In our experiments, $U=100$ for MNIST, $12$ for ImageNet, and $5$ for CIFAR. We should note that future work can investigate if regularzing every layer is even required to save computation.
>
> We hope we have addressed all your concerns, and we hope you will consider raising the score to reflect that.

---

> > ### Author Rebuttal · Reviewer_hA2p · 2026-03-31
> >
> > Thank you for your reply. However, my primary theoretical concern remains unresolved. The causal argument breaks down across the two regimes (at least for the MSE loss in Section 4):
> >
> > * Within the NTK Regime: Your mathematical framework successfully proves that a rank-collapsed Hessian leads to learning difficulties. However, under NTK assumptions, the Hessian is frozen and continual learning cannot cause rank collapse.
> >
> > * Outside the NTK Regime: Your empirical results demonstrate that continual learning does lead to rank collapse. However, your NTK-based theory is no longer valid in this non-stationary regime. Consequently, you cannot show that this rank collapse is the cause of plasticity loss outside the NTK setting, rather than just a correlated symptom.
> >
> > Thus my score will remain unchanged.
> >
> > **Reply to authors' further rebuttal**:
> >
> > Thank the authors for their detailed response. They claim that their causal claim relies on the cross-entropy extension where G_{CE} = H^{1/2} G^{(0)} H^{1/2}.
> >
> > ​However, examining the logit-Hessian H reveals a contradiction that invalidates the claim.
> >
> > ​For cross-entropy, the logit-Hessian is H = \text{diag}(p) - pp^T. Its scale depends on the predicted probabilities p.
> > ​At the end of Task A, the network is highly confident, p is saturated (near 0 or 1), and H \to 0. The spectrum collapses. This is expected.
> >
> > ​At the initialization of Task B (e.g., a newly permuted MNIST dataset) and under the NTK assumption (which your Section 4 theory relies on): Logits evaluated on a novel permuted dataset will roughly exhibit mean zero and O(1) variance. Consequently, the probabilities p will be close to uniform (i.e., the network makes random guesses on the new task). In this scenario, H is well-conditioned and full-rank. The matrix H, and consequently G_{CE}, would "bounce back" to full capacity. Spectral collapse at the start of a new task is impossible in the NTK limit.
> >
> > ​Thus, the theoretical framework in Section 4 does not—and mathematically cannot—model the cause of the rank collapse observed in your empirical results. It merely describes a consequence. While I still appreciate the practical utility of the L2-ER regularizer, the theoretical justification remains flawed.

---

> > > ### Author Response · Authors · 2026-04-03
> > >
> > > The reviewer's concern is valid for the MSE trainability lemmas taken in isolation. Under the linearized relu model, the kernel matrix $G=JJ^\top$ is fixed, which implies the MSE loss Hessian $J\top J$ is also fixed. Section 4 gives the necessary conditions for finite time trainability and specifically Lemma 4.7 establishes that if too many of the eigenvalues fall below $\epsilon_{K, \gamma}$, then learning the task is not possible (with some exceptions).
> > >
> > > However, our causal claim in section 5 relies on the cross-entropy extension where the relevant curvature matrix is the weighted gram matrix $G_{CE} = H^{1/2}G^{(0)}H^{1/2}$, where the kernel matrix $G^{(0)}$ is proved to be frozen (Lemma B.10). Even with a frozen kernel matrix, the weighted gram matrix can evolve because $H = \nabla^2_z l(z)$ depends on the logits $z$. Our paper explicitly states that $G_{CE}$ can evolve, and we empirically show that the $\epsilon$-rank of $G_{CE}$ collapses in Figure 2. Moreover, Figure 5C shows that the $\epsilon$-rank of $G^{(0)}$ remains flat over training. So while the frozen NTK objection applies to MSE loss, it does not apply to the kernel matrix induced by CE that underlies our result. We believe this is a useful finding as previous works have attributed loss of plasticity to the low-rank empirical NTK [1], while our finding shows that the reality is more subtle. We acknowledge that future work should establish a corollary to Lemma 4.7, which specifically adapts the failure condition to CE loss, which we will add to the discussion.  This would neatly separate out the contribution of the $\nabla^2_z l(z)$ term to spectral collapse while $G^{(0)}$ remains frozen.
> > >
> > > For the second concern, we believe it is subject to future research. It would be helpful to prove that continual learning causes spectral collapse in deep non-linear networks. For now, we have isolated a concrete mechanism of failure in continual learning. We validate this mechanism twice: first, we show that this failure is realized in the linearized relu setting where $G_{CE}$ collapses even though the $G^{(0)}$ matrix remains invariant. Then, in deep non-linear networks, we observe the same failure mode: Loss of plasticity is preceded by the collapse of the Hessian spectrum at the initialization of the task. Moreover,  our L2-ER optimizer, which preserves the bulk of this spectrum reliably maintains plasticity. This is the standard role of analyzable models: to identify the governing dynamics under assumptions, which can then be tested beyond where that model is exact.
> > >
> > > We thank the reviewer for pushing us to clarify our theoretical claims and how they relate to practice. We are more than happy to incorporate this into the discussion of our limitations.
> > >
> > > [1] Lyle, Clare, et al. "Disentangling the causes of plasticity loss in neural networks." arXiv preprint arXiv:2402.18762 (2024).

---

### Official Review · Reviewer_y3ZQ · 2026-03-12

**Soundness:** 3
**Presentation:** 3
**Significance:** 3
**Originality:** 3
**Overall Recommendation:** 4
**Confidence:** 1

**Summary:**

This paper studies the loss of plasticity in deep continual learning and argues that it is closely related to Hessian spectral collapse. The paper provides a theoretical perspective connecting trainability to the effective rank of curvature-related matrices, and proposes a practical regularization method, L2-ER, which combines L2 regularization with an effective-rank penalty. Experiments on several continual supervised learning and reinforcement learning benchmarks show that L2-ER improves plasticity compared with standard training and several prior methods.

**Compliance With Llm Reviewing Policy:**

Affirmed.

**Final Justification:**

My concern has been resolved, but reviewer hA2p’s concern has not. Therefore, I am raising my score while lowering my confidence.

**Key Questions For Authors:**

Can the authors provide stronger evidence that spectral collapse is a causal driver of plasticity loss, rather than mainly a correlated diagnostic?
How much of the final improvement comes from the effective-rank regularization itself, versus the L2 component?
How robust is the proposed explanation across settings where the Hessian-rank proxy may be less reliable?

**Limitations:**

This is an interesting and non-trivial paper with a valuable perspective on continual learning. I find the main idea meaningful and the experiments reasonably solid. However, the current version is more convincing as a strong mechanism hypothesis plus a useful regularizer than as a fully established explanation of plasticity loss.

**Strengths And Weaknesses:**

Strengths

The paper addresses an important problem that is highly relevant beyond standard forgetting-centric continual learning. The central perspective—linking plasticity loss to spectral collapse—is interesting and gives the work a stronger conceptual foundation than many purely heuristic papers. The paper also includes both theory and empirical results, and the experiments cover multiple environments rather than a single benchmark. Overall, the work is thoughtful and has clear research value.

Weaknesses

My main concern is that the paper makes a fairly strong claim about spectral collapse being the central mechanism behind plasticity loss, while the evidence is still more correlational than causal. The experiments support a meaningful connection between spectral properties and plasticity, but they do not yet fully establish that spectral collapse is the driving cause rather than an accompanying symptom.

A second concern is that the final method, L2-ER, is relatively simple compared with the ambition of the theoretical narrative. This is not necessarily a flaw, but it makes the paper rely heavily on the strength of its mechanism claim. If that claim is weakened, the methodological novelty becomes more moderate.

Finally, some benchmark-specific analyses are more convincing than others, so the generality of the proposed explanation is not yet fully established.

---

> ### Author Rebuttal · Authors · 2026-03-30
>
> We thank the reviewer for their questions and for pushing us to sharpen the causal framing.
>
> **Causality vs Correlation**
> Our claim has two levels. First, in the NTK setting, lemma 4.7 shows that if too many eigenvalues of the kernel matrix, G, lie below the threshold $\epsilon_{K, \gamma}$, the subsequent task cannot be learned within K steps (with some exception). This means that the bulk of the eigenspectrum of the G must be wide.
>
> We further isolate this mechanism from task changes on Permuted MNIST in the NTK regime. We demonstrate that the weighted kernel matrix collapses across tasks, even though the kernel matrix at initialization $G^{(0)}$ is invariant to pixel permutations. Empirically we show that controlling the feature rank of this weighted matrix controls the rate of plasticity loss. Specifically, when we use a low-feature rank initialization which corresponds to a narrow-bulk Kernel matrix,  plasticity loss is worse than a full-feature rank initialization, as shown in Figure 2.  We then empirically show that this mechanism remains operational in deep non-linear networks. Our L2-ER regularizer increases the rank of the Hessian by increasing the feature rank. Our empirical results show that when we apply our intervention, the spectrum of the Hessian retains a wide bulk and maintains plasticity. When we remove the intervention, the bulk collapses and so does plasticity. Moreover, we measure the spectrum at the initialization of each new task, showing that it is a precursor to the inability to learn.  In the revision, we will make this two-level claim explicit in the Introduction/Discussion and separate the exact linearized causality statement from the deep-network intervention evidence. We will also move Figure 5c to the main text to demonstrate that $G^{(0)}$ remains invariant throughout training
>
> **Simplicity of L2-ER**
> We believe the applicability of our analysis lies in the simplicity of the resulting regularizer. We find that **both** L2 and ER are needed to preserve plasticity; each loses plasticity on its own but preserves plasticity when applied together (shown in Figure 4). We believe this finding is non-trivial and it is of interest to the broader community when well-known methods interact in unexpected ways. We will be sure to highlight this fact more clearly in the discussion of our experimental results.
>
> **Benchmarks**
> We picked the full suite of experiments from Dohare et al. [1], and our method maintains plasticity on all tasks while either outperforming or matching baselines. As NTK analysis has been useful for the analysis of deep neural networks, we believe our results should also hold more generally.
>
> **Specific questions**
>
> On causality: Please see our above comment
>
> On ablating L2 and ER: Both are needed to ensure that increasing the rank of the proxy $H\_{GGN}$ is guaranteed to increase the rank of the full Hessian since the additional residual matrix needs to be taken into account. We will be sure to highlight this fact more clearly in the discussion of our experimental results.
>
> On robustness of the approximation: We believe our method is robust as it is able to maintain a wide spectrum bulk across MLPs, CNNs and ResNet. KFAC has also been adapted to RNNs [2] and vision transformers [3], so we believe our method can be adapted to these cases as well.
>
> We hope we have addressed all your concerns, and we hope you will consider raising the score to reflect that.
>
>
> [1] Dohare, Shibhansh, et al. "Loss of plasticity in deep continual learning." Nature 632.8026 (2024): 768-774.
> [2 ] Martens, James, Jimmy Ba, and Matt Johnson. "Kronecker-factored curvature approximations for recurrent neural networks." International Conference on Learning Representations. 2018.
> [3] Eschenhagen, Runa, et al. "Kronecker-factored approximate curvature for modern neural network architectures." Advances in Neural Information Processing Systems 36 (2023): 33624-33655.

---

> > ### Author Rebuttal · Reviewer_y3ZQ · 2026-04-03
> >
> > I am mostly convinced, but reviewer hA2p’s comments raise a new concern for me: the NTK issue seems to conflict with your algorithm when the Hessian matrix is assumed to be frozen. Please explain this.

---

> > > ### Author Response · Authors · 2026-04-03
> > >
> > > We thank the reviewer for the opportunity to expand on the discussion of the frozen kernel matrix.
> > >
> > > Reviewer hA2p's concern is valid for the MSE trainability lemmas taken in isolation. Under the linearized relu model, the kernel matrix $G=JJ^\top$ is fixed, which implies the MSE loss Hessian $J^\top J$ is also fixed. Section 4 gives the necessary conditions for finite time trainability and specifically Lemma 4.7 establishes that if too many of the eigenvalues fall below $\epsilon_{K, \gamma}$, then learning the task is not possible (with some exceptions).
> > >
> > > However, our causal claim in section 5 relies on the cross-entropy extension where the relevant curvature matrix is the weighted gram matrix $G\_{CE} = H^{1/2}\_zG^{(0)}H^{1/2}\_z$, where $G^{(0)}$ is the initial kernel matrix and $H\_z = \nabla^2\_z l(z)$ with logits $z$.  In Lemma B.10 we prove that $G^{(0)}$  remains frozen despite pixel permutations.
> > >
> > > Even with a frozen kernel matrix, the $C_{CE}$ can evolve as the logits $z$ change. We empirically show that the $\epsilon$-rank of $G\_{CE}$ collapses in Figure 2. Moreover, in Figure 5C, we show that the $\epsilon$-rank of $G^{(0)}$ remains flat over sequential tasks. So while the frozen NTK objection applies to MSE loss, it does not apply to the weighted kernel matrix induced by CE loss that underlies our result.
> > >
> > > Relating this back to the full parameter space Hessian and Generalized-Gauss Newton Matrix. Recall that  $G_{CE} = H^{1/2}\_zG^{(0)}H^{1/2}\_z = (H^{1/2}\_z J)(H^{1/2}\_z J)^\top$, while $H\_{GGN} = J^\top H\_z J =(H^{1/2}\_z J)^\top(H^{1/2}\_z J)$. Hence, $H_{GGN}$ and $G_{CE}$ have the same non-zero eigenspectrum. In our frozen linearized ReLU model, the full parameter-space Hessian is equivalent to the $H_{GGN}$ (because the residual Hessian term is zero since the model is linear in the trainable parameters).
> > >
> > > Thus, we isolate the fixed component $G^{(0)}$ and show that collapse under CE loss arises through the evolving logit-space curvature $H_z$, even though the initial kernel matrix remains frozen.  We hope this clarifies which component of the Hessian is fixed, and we will gladly include this extra discussion at the start of section 5 to clarify the issue in the camera-ready.
> > >
> > > We believe this is a useful finding as previous works have attributed loss of plasticity to the low-rank empirical NTK [1], while our finding shows that the reality is more subtle. We acknowledge that future work should establish a corollary to Lemma 4.7, which specifically adapts the failure condition to CE loss, which we will add to the discussion.  This would neatly separate out the contribution of the $\nabla^2_z l(z)$ term to spectral collapse while $G^{(0)}$ remains frozen.
> > >
> > > **Outside the NTK regime**
> > >
> > > For deep non-linear networks, we measure the Hessian spectrum at new-task initialisation and observe that collapse consistently precedes failures to learn, while methods that preserve a wide spectral bulk also preserve plasticity across our deep-network benchmarks. In this case, our evidence is intervention-based but motivated by our theoretical insight. In particular, connections to the implicit low-rank bias of SGD [2] and to singular learning theory are promising future directions to explain why deep nonlinear networks suffer from the spectral collapse pathology that we have identified in the current work.
> > >
> > > We gladly incorporate these clarifications into the camera-ready and highlight where future research can close the gaps between the theoretical model and practical non-linear networks.
> > >
> > > [1] Lyle, Clare, et al. "Disentangling the causes of plasticity loss in neural networks." arXiv preprint arXiv:2402.18762 (2024).
> > >
> > > [2] Galanti, Tomer, and Tomaso Poggio. "Sgd noise and implicit low-rank bias in deep neural networks." (2022).

---

### Official Review · Reviewer_HRC3 · 2026-03-12

**Soundness:** 4
**Presentation:** 3
**Significance:** 4
**Originality:** 3
**Overall Recommendation:** 6
**Confidence:** 4

**Summary:**

The authors argue that loss of plasticity can best be understood in terms of spectral collapse, where the Hessian eigenspectrum loses (epsilon-)rank. They provide empirical evidence across a wide range of continual learning benchmarks, theoretically characterize the conditions for loss of plasticity in a linearized ReLU network, and propose a new form of regularization which limits spectral collapse and improves continual learning outcomes.

**Compliance With Llm Reviewing Policy:**

Affirmed.

**Final Justification:**

There have been a lot of attempted explanations for the loss of plasticity in the literature. So much so that the leading explanation is the "Swiss Cheese Model" by Lyle et al. (2024), which posits that loss of plasticity isn't a single thing but that it's driven by a bunch of independent mechanisms. That seemed reasonable, but it leaves us without a unified mechanistic picture.

This paper provides, in my opinion, the first convincing counterargument. It argues that the different mechanisms are different facets of the same underlying Hessian spectral collapse. This isn't the first work to attempt at a unified explanation: Lewandowski et al. (2024) were earlier and, in fact, also identify spectral collapse (under a different name) as the relevant phenomenon. However, the current work is considerably more comprehensive and convincing. Unlike Lewandowski, this work shows that explicitly intervening to preserve the Hessian spectrum increases plasticity (the Wasserstein regularization in Lewandowski is from my understanding less grounded), and some derivations that show mechanistically how curvature collapse can prevent subsequent development.

To be transparent, the theoretical treatment isn't "complete." There's an unresolved theory-practice gap (see limitations discussed by various reviewers, esp. hA2p). Still, I believe this analysis meaningfully moves the discussion forward, and I do not think this limitation is fatal. Loss of plasticity is a major research program, and I do not expect any single paper to fully close the gap all at once.

Any future paper on loss of plasticity will need to engage with this work. For that reason, and given the authors' thorough responses during the rebuttal period, I am revising my score upward to strong accept.

**Key Questions For Authors:**

* **Q1 Do you have any results for language models (and larger models more generally)?** I won’t fault you for there not being good LLM baselines in the plasticity literature, but I think the paper would benefit from examining larger models.
* **Q2 Do you have any thoughts on how this relates to singular learning theory (SLT)?** SLT provides some reasoning for why we should expect increasing degeneracy (spectral collapse) to happen as a consequence of learning. I expect this to provide a useful perspective here. See [example](https://arxiv.org/abs/2510.12071).
* **Q3 How do the ER penalty compute costs scale?** The single snapshot in figure 13 is useful, but could you add some comments in appendix K on how the different techniques scale with model size and dataset size? Am I right to expect that the relative cost for (L2-)ER will get much larger for larger models?
* **Q4 How do you choose the other ER hyperparameters (batch size, steps, etc.)?** I am familiar with the pains of Hessian estimation, and the first place I expect hidden problems is in hyperparameter selection. I appreciate the appendices you included detailing this, and would be interested in hearing whether you have any recommendations for future practitioners.

Generally, I recommend the authors use their extra page of the camera-ready to expand a discussion section. I would also appreciate some commentary on how their results compare to e.g., the swiss-cheese model, some commentary on limitations (see below), and any other discussion the authors find suitable. With some of these changes implemented, I'm likely to recommend a strong accept.

Typos:
* 226 col 1: the K should be in the exponent.

**Limitations:**

I’d recommend the authors add some comments on the remaining gap between the theoretical treatment considered here and real-world models (e.g., taking the NTK limit, the linearization necessary to extend this analysis to cross-entropy objectives, the K-FAC approximation error, etc.). These assumptions are mentioned throughout but could be reiterated in the discussion. The NTK linearization in particular means this analysis holds best when a new task is introduced, but should get less accurate with episode length.

**Strengths And Weaknesses:**

* **Soundness**: I haven’t carefully checked the theory, but I can follow most of it since the theoretical ingredients are pretty straightforward, and the clever part is how the authors combine everything together. Those contributions seem very solid. The experiments are thorough (albeit at a quite small model scale). The only limitation is that the gap between theory and practice may be larger than presented. I think this can be mitigated by expanding the discussion on limitations (see below).
* **Presentation**: The paper is well-written.
* **Significance**: The central contribution of this paper is to “unify” many different explanations of loss of plasticity under one combined heading. I’m familiar with the plasticity literature and I expect that this paper will have an important effect.
* **Originality**: A lot of papers have investigated the role of loss landscape geometry and spectral properties of the Hessian as they relate to plasticity. That much isn’t new. But the synthesis is new and the epsilon-rank and L2-ER regularization contributions are new.

---

> ### Author Rebuttal · Authors · 2026-03-30
>
> We thank the reviewer for the thoughtful questions and suggestion to use the extra camera-ready page to expand the discussion. We will use that space to better position our work relative to prior explanations of plasticity loss, clarify limitations, and expand the practical discussion of L2-ER.
>
> **Q1: Larger models/language models.**
> We do not yet have LM results. Our main contribution is to isolate the mechanism in the NTK regime (Section 4.1), then show that the same analysis remains useful for understanding deep nonlinear models empirically. Extending to LMs is an exciting next step, and will likely require a selected-layer version of L2-ER and additional analysis of attention. We will add this to the limitations section.
>
> **Q2: Relation to SLT.**
> We thank the reviewer for raising this connection. We view SLT as complementary: NTK kernel and Fisher information are equivalent up to reversal and scaling [1], and SLT explains why overparameterized models move toward singular regions. Our work studies the continual-learning consequence of this phenomenon. We ask when eigenvalues become so small that gradient descent is practically ineffective for future tasks. We will add this to related work.
>
> **Q3/Q4: ER complexity and hyperparameters.**
> We will add this discussion to Section 6. For layer $l$, we construct a stacked feature matrix $M_l\in\mathbb{R}^{m_l\times d_l}$, where $d_l$ is the layer width and $m_l$ is the number of buffered features used in one ER update. Computing $ER(M_l)$ requires an SVD with cost $O(m_l d_l \min\\{m_l,d_l\\})$; amortized over update interval $U$, the per-step overhead is $ O  ( \frac{1}{U} \sum_l m_l d_l \min\\{m_l,d_l\\})$. In our experiments, $U=100$ for MNIST, $12$ for ImageNet, and $5$ for CIFAR. In our sweeps, the ER learning rate is the most sensitive hyperparameter, while the ER batch/window mainly stabilizes the singular-value estimate. In practice, we recommend tuning the task optimizer first, then tuning the ER learning rate below the task learning rate, using the smallest ER batch/window that gives a stable estimate, and applying ER as infrequently as possible while retaining benefit. We will also clarify that the eigenspectrum analysis is diagnostic and not directly used in the regularizer: L2-ER indirectly shapes the Hessian without explicitly materializing it.
>
> **Relation to swiss-cheese and other plasticity-preserving methods.**
> We appreciate this suggestion and will expand the discussion. We do not view Hessian spectral collapse as an alternative to the “swiss cheese” model of Lyle et al. [2], but as a unifying consequence of several of its pathologies. In our K-FAC analysis, the approximate Generalized Gauss-Newton matrix is $\\hat{H}\_{GGN}=\\mathbb{E}[aa^\\top]\\otimes\\mathbb{E}[gg^\\top]$, so $\\mathrm{rank}(\\hat{H}_{GGN})=\\mathrm{rank}(\\mathbb{E}[aa^\\top])\\times\\mathrm{rank}(\\mathbb{E}[gg^\\top])$. Thus, low feature rank collapses $\mathbb{E}[aa^\top]$, collinear gradients collapse $\mathbb{E}[gg^\top]$, and dead ReLUs also upper-bound Hessian rank [3]. This gives a common mathematical lens for several causes of plasticity loss, and also helps explain why methods like CBP work: resetting dormant neurons restores variance to $\mathbb{E}[aa^\top]$, thereby indirectly preserving Hessian rank. Figures 3 and 7 are consistent with this view. We will add this discussion.
>
>
> **Limitations and theory–practice gap.**
> We agree this should be stated more explicitly. Our theoretical treatment in Section 4.1 relies on a specifically constructed linearized ReLU network: by freezing the activation gates and output weights at initialization, the model becomes affine in the trainable first-layer weights. For deep nonlinear networks, however, the NTK approximation no longer holds, especially as the parameters leave the local neighbourhood as training progresses. We will update the discussion with this limitation. We will also make the gap between theory and practice clearer by promoting parts of Appendix C (KFAC error analysis) and Appendix B.3 (cross entropy objective) into the main paper.
>
> We also thank the reviewer for catching the typo on line 226.
>
> We hope we have addressed all your concerns, and we hope you consider raising the score to reflect that.
>
> [1] Karakida, Ryo, Shotaro Akaho, and Shun-ichi Amari. “Pathological spectra of the fisher information metric and its variants in deep neural networks.” arXiv preprint arXiv:1910.05992 (2019).
> [2] Lyle, Clare, et al. “Disentangling the causes of plasticity loss in neural networks.” arXiv preprint arXiv:2402.18762 (2024).
> [3] Singh, Sidak Pal, Gregor Bachmann, and Thomas Hofmann. “Analytic insights into structure and rank of neural network hessian maps.” NeurIPS 34 (2021): 23914–23927.
> [4] Ziyin, Liu. “Symmetry induces structure and constraint of learning.” arXiv preprint arXiv:2309.16932 (2023).

---

> > ### Author Rebuttal · Reviewer_HRC3 · 2026-03-31
> >
> > Thank you for the detailed and thoughtful responses to all reviewers. I'm largely satisfied with the proposed revisions, but have a few additional points.
> >
> > On Q2: I appreciate the response, but want to note that SLT is not limited to the overparametrized limit. It provides a general account for why deep neural networks may end up with an inductive bias towards low spectral rank (or more generally towards high degeneracy). It'd be an interesting direction for future research to attempt to relate this perspective to the one you've established here.
> >
> > On Limitations: Thank you, this goes a long way to addressing my concerns. I'd like to ask the authors to engage a bit more with Reviewer hA2p's specific concerns about the analysis breaking down:
> >
> > > - Within the NTK Regime: Your mathematical framework successfully proves that a rank-collapsed Hessian leads to learning difficulties. However, under NTK assumptions, the Hessian is frozen and continual learning cannot cause rank collapse.
> > > - Outside the NTK Regime: Your empirical results demonstrate that continual learning does lead to rank collapse. However, your NTK-based theory is no longer valid in this non-stationary regime. Consequently, you cannot show that this rank collapse is the cause of plasticity loss outside the NTK setting, rather than just a correlated symptom.
> >
> > I think it is appropriate to discuss this as an open question and direction for future research without needing to resolve it in the current work. But I would like to see it explicitly acknowledged in the discussion/limitations section. If the authors commit to incorporating this, I would consider revising my recommendation upward.

---

> > > ### Author Response · Authors · 2026-04-03
> > >
> > > We thank the reviewer for the opportunity to expand on our theoretical contributions and to demonstrate how they align with practice. In the NTK setting, Lemma 4.7 shows that if too many eigenvalues of the kernel matrix lie below the threshold $\epsilon_{K, \gamma}$, the subsequent task cannot be learned within K steps (with some exceptions). Then, in the deep non-linear network case, we observe that this failure mode manifests in the Hessian, and that methods which intervene on the spectrum to maintain a wide bulk preserve plasticity.
> > >
> > > We will gladly commit to including the limitations of our current work in the discussion of the camera-ready.
> > >
> > > We agree that extending the causal proof to the fully nonlinear deep neural network case remains an open challenge. Moreover, we agree that establishing exactly why the Hessian becomes low-rank as training progresses across sequential tasks is also an open question. We believe that insights from singular learning theory, and from the implicit low-rank bias of stochastic gradient descent [1] are promising directions for future work to understand these learning dynamics.
> > >
> > > For the specific comments from reviewer hA2p:
> > >
> > > **Within the NTK Regime**
> > > Our failure mode is indeed established for NTKs under the MSE loss, however our empirical result on the linearized network occurs with CE loss. While we empirically observe the collapse of $G_{CE}$, future work should establish a corollary to Lemma 4.7, which specifically adapts the failure condition to CE loss.  This would neatly separate out the contribution of the $\nabla^2_z l(z)$ term to spectral collapse while $G^{(0)}$ remains frozen.
> > >
> > > **Outside the NTK regime**
> > > We agree that our current theory does not provide a full end-to-end proof that sequential training in a fully nonlinear deep network must itself generate spectral collapse. Empirically, we measure the Hessian spectrum at new-task initialization and observe that collapse consistently precedes failures to learn, while methods that preserve a wide spectral bulk also preserve plasticity across our deep-network benchmarks.  Outside the exact NTK regime, our evidence is intervention-based but motivated by our theoretical insight.  In particular, connections to the implicit low-rank bias of SGD and to singular learning theory are promising future directions to explain why deep nonlinear networks suffer from the spectral collapse pathology that we have identified in the current work.
> > >
> > > We thank the reviewer for correcting our misunderstanding of SLT, and believe that integrating SLT into future work could genuinely help develop the theory of loss of plasticity.
> > >
> > > [1] Galanti, Tomer, and Tomaso Poggio. "Sgd noise and implicit low-rank bias in deep neural networks." (2022).

---

### Decision · Program_Chairs · 2026-04-30

**Decision:**

Accept (regular)

**Comment:**

This paper attributes the loss of plasticity (i.e., the ability to learn new tasks in continual learning) to the degeneration of the Hessian curvature in training. In particular, the Hessian eigenspectrum captures the meaningful directions that gradient descent can use to learn. When too many eigenvalue directions become small, the model loses the capacity to learn new tasks. The paper analyses a linearized ReLU/NTK regime and shows that successful learning requires the task signal to align with the fast eigenspace (the directions with large eigenvalues). They also provide a regularization method to prevent the Hessian spectrum from collapsing. They tested their results on continual supervised learning and reinforcement learning tasks and showed that their method can prevent the spectral collapse and preserve plasticity.

Main reasons for acceptance:
- Clear motivation and well-structured paper for an important problem in continual learning.
- Strong conceptual contribution that frames spectral collapse as a unifying mechanism.
- Insightful theoretical analysis and good empirical support for L2-ER. Theory and experiments are broadly consistent.

The reviewers also raised several useful comments, including the theory's assumptions about NTK linearization and that the contribution on spectral collapse does not seem entirely new. In particular, one reviewer has mentioned that under the NTK assumptions, the Hessian is frozen, and continual learning cannot cause rank collapse. So, there is some concern that the NTK-based theory has some gap to the actual training. About this point, the authors should be explicit in their revision. None of these seems to be a strong limitation for rejection, though.